# Higher-Order Learning with Graph Neural Networks via Hypergraph Encodings

**Raphael Pellegrin**\*
Independent Researcher
raphaelpellegrin@alumni.harvard.edu

**Lukas Fesser**\*
Harvard University
lukas_fesser@fas.harvard.edu

**Melanie Weber**
Harvard University
mweber@seas.harvard.edu

## Abstract

Higher-order information is crucial for relational learning in many domains where relationships extend beyond pairwise interactions. Hypergraphs provide a natural framework for modeling such relationships, which has motivated recent extensions of graph neural network (GNN) architectures to hypergraphs. Most of these architectures rely on message-passing to encode higher-order information. In this paper, we propose to instead use *hypergraph-level* encodings based on characteristics such as hypergraph Laplacians and discrete curvature notions. These encodings can be used on datasets that are naturally parametrized as hypergraphs and on graph-level datasets, which we reparametrize as hypergraphs to compute encodings. In both settings, performance increases significantly, on social networks by more than 10 percent. Our theoretical analysis shows that hypergraph-level encodings provably increase the representational power of message-passing graph neural networks beyond that of their graph-level counterparts. For complete reproducibility, we release our codebase: https://github.com/Weber-GeoML/Hypergraph_Encodings.

## 1 Introduction

Many datasets have inherent "multi-way" structure, where downstream tasks depend on relationships between groups of entities that ordinary graphs, whose edges are pairwise relationships, cannot represent (Bick et al., 2023; Benson et al., 2021; Schaub et al., 2021). Hypergraphs overcome this by allowing hyperedges that connect any number of vertices. *How can models effectively leverage such higher-order information for learning?*

The enhanced flexibility of hypergraphs has motivated a growing body of literature on hypergraph neural network architectures, including message-passing (Huang & Yang, 2021) and transformer-based models (Liu et al., 2024). Naïvely extending these paradigms from graph neural networks to hypergraphs often fails to capture crucial substructures. On the other hand, reparametrizing a hypergraph as a standard graph via clique expansion can introduce redundant pairwise edges that obscure higher-order structures. This suggests that architectural bias alone is insufficient.

While previous work has focused on such architectural extensions, we propose a different route. Instead of encoding higher-order structure as inductive biases directly, we propose a data augmentation approach. *Encodings* that augment the input with structural information (e.g., spectra (Dwivedi et al., 2023), motif counts (Zhao et al., 2021), discrete curvature (Fesser & Weber, 2024a)) have been shown to yield significant performance gains. However, encodings have been studied almost exclusively at the graph-level and fail to capture higher-order information specific to hypergraph representations.

---

[1]These authors\* contributed equally.

39th Conference on Neural Information Processing Systems (NeurIPS 2025).

In this work we design hypergraph-level encodings based on classical hypergraph characteristics, and prove that they capture higher-order structural information that cannot be represented by standard hypergraph message-passing architectures or existing graph-level encodings. Our experimental results benchmark the new encodings in combination with graph- and hypergraph-level message-passing as well as transformer-based architectures. While hypergraph-level encodings offer only marginal benefit for hypergraph-level architectures, they yield significant performance gains when paired with graph-level message-passing and transformers.

## 1.1 Related Work

Topological Deep Learning has emerged as the dominant framework for learning on topological domains, including hypergraphs, as well as simplicial, polyhedral and more general cell complexes (Hajij et al., 2022, 2024; Papillon et al., 2023). Many classical graph-learning architectures have been extended to these domains. In the case of hypergraphs, this includes message-passing (Huang & Yang, 2021; Wang et al., 2022, 2023) and transformer-based (Liu et al., 2024) hypergraph neural networks.

To the best of our knowledge, encodings have so far only been studied in the context of graph-level learning (Dwivedi et al., 2023). Popular encodings leverage structural and positional information captured by classical graph characteristics (Rampášek et al., 2022; Kreuzer et al., 2021; Cai & Wang, 2018; Zhao et al., 2021; Fesser & Weber, 2024a; Bouritsas et al., 2022).

## 1.2 Summary of Contributions

The main contributions of this paper are as follows:

(1) We introduce hypergraph-level encodings that augment a (hyper-)graph-structured input with higher-order positional and structural information captured in hypergraph characteristics.

(2) We show that hypergraph-level encodings are provably more expressive than their graph-level counterparts.

(3) We show that hypergraph-level encodings can significantly enhance the performance of graph neural networks applied to hypergraph expansions.

## 2 Background

We consider graphs $G = (V, X, E)$ with node attributes $X \in \mathbb{R}^{|V| \times m}$ and edges $E \subseteq V \times V$, representing pairwise relations between nodes in $V$. We further consider hypergraphs $H = (V, X, F)$ where hyperedges $F$ denote relations between groups of nodes. Hypergraphs can be reparametrized as graphs using clique expansions; for more details see Apx. A.1. A graph can be lifted to a hypergraph by adding a hyperedge for every set of vertices that are pairwise connected in the graph - effectively turning each clique into a single higher-order relation; see Apx. A.2. In a graph, we represent the set of all nodes $j$ that are adjacent to $i$ as $\mathcal{N}_i = \{j \in V \mid (i,j) \in E\}$. In a hypergraph, we similarly define the neighborhood of a node $i \in V$ as $\mathcal{N}_i = \{j \in V \setminus \{i\} \mid \exists e \in F \text{ such that } \{i,j\} \subseteq e\}$ i.e., the set of nodes $j$ that co-occur with $i$ in at least one hyperedge. The degree $d_i$ of a vertex $i$ of an undirected (hyper)graph $H = (V, X, F)$ is the number of (hyper)edges that contain $i$ (Klamt et al., 2009).

## 2.1 Architectures

**Message-passing GNN.** Message-Passing (MP) (Gori et al., 2005; Hamilton et al., 2017) is a prominent learning paradigm in relational learning, where a node's representation is iteratively updated based on the representations of its neighbors. Formally, let $x_v^l$ denote the representation of node $v$ at layer $l$. Message-passing implements the following update,

$$x_v^{l+1} = \phi_l \Big( \bigoplus_{p \in \mathcal{N}_v} \psi_l \left( x_p^l \right) \Big) ,$$

where $\psi_l$ denotes an aggregation function (e.g., averaging) acting on the 1-hop neighborhood $\mathcal{N}_v$ of $v$ (if self-loops are added, we consider $p \in \mathcal{N}_v \cup \{v\}$ instead), and $\phi_l$ an update function with trainable

parameters, such as an MLP. The number of MP iterations is commonly referred to as the *depth* of the network. Representations are initialized by the node attributes in the input.

**Transformer-based GNN.** The second major class of architectures for relational learning is transformer-based (GT). Networks consist of blocks of multi-head attention layers ($GlobalAttn(\cdot)$), followed by fully-connected feedforward networks. In the recent literature, hybrid architectures, which combine MP and attention layers, have been shown to exhibit strong performance on several state of the art benchmarks (Rampášek et al., 2022).

**Graph-level Architectures.** Our selection of graph-level architectures includes two message-passing GNNs (MPGNNs) and one hybrid architecture. GCN (Kipf & Welling, 2016) is one of the simplest and most popular MPGNNs, making it an important reference point. GIN (Xu et al., 2018) is designed to be a maximally expressive MPGNN. GraphGPS (Rampášek et al., 2022) is a widely used hybrid architecture that performs well across the benchmarks considered here. These models remain strong baselines and still outperform many newer models in both node- and graph-level tasks (Luo et al., 2024). As baselines, we evaluate simple instances of all three architectures without additional model interventions. An overview of the architectures can be found in Tab. 4; more detailed descriptions are deferred to Apx. B.1.

**Hypergraph-level Architectures.** The architectures[1] analyzed in this study implement message-passing, which on hypergraphs is implemented via a two-phase scheme: messages are passed from nodes to hyperedges and then back to nodes (Huang & Yang, 2021). Formally,

$$h_e^{l+1} = \phi_1 \left( \left\{ x_j^l \right\}_{j \in e} \right) , \qquad \tilde{x}_i^{l+1} = \phi_2 \left( x_i^l, \left\{ h_e^{l+1} \right\}_{e \in E_i} \right) .$$

Here, $x_j$ denotes the node features of node $j$, $h_e$ denotes the edge feature of edge $e$, $E_j$ is the set of all hyperedges containing $j$, and $\phi_1$ and $\phi_2$ are permutation-invariant functions for aggregating messages from vertices and hyperedges respectively. $\tilde{x}_i$ indicates the output of the message-passing layer before activation or normalization. Tab. 4 provides an overview of the hypergraph-level architectures considered here; more detailed description can be found in Apx. B.2.

## 2.2 Encodings

Structural (SE) and Positional (PE) encodings enhance MPGNNs by providing access to structural information that is crucial for downstream tasks, but that these networks cannot inherently learn (Dwivedi et al., 2023; Rampášek et al., 2022). Encodings can capture either local or global properties of the input graph. Local PEs supply nodes with information about their position within local clusters or substructures, such as their distance to the centroid of their community. In contrast, global PEs convey a node's overall position within the entire graph, often based on spectral properties like the eigenvectors of the Graph Laplacian (Kreuzer et al., 2021) or random-walk based node similarities (Dwivedi et al., 2021). Graph-level SEs capture structural information, such as pair-wise node distances, node degrees, or statistics regarding the distribution of neighbors' degrees (Cai & Wang, 2018), or discrete curvature (Fesser & Weber, 2024a). Empirical evidence demonstrates that incorporating these PEs and SEs significantly improves the performance of GNNs (Rampášek et al., 2022).

## 2.3 Representational Power

A key theoretical question in evaluating the effectiveness of different relational learning architectures is their *representational power* or *expressivity*: Which functions can and cannot be learned by the model? This question can be analyzed through the lens of a model's ability to distinguish graphs that are not topologically identical (isomorphic). The 1-Weisfeiler-Leman (1-WL) test (Weisfeiler & Leman, 1968) provides a heuristic for this question. Notably, Xu et al. (2018) showed that MPGNNs (specifically, GIN) are as expressive as the 1-WL test. While 1-WL (and, by extension, MPGNNs) is effective for many classes of graphs, it has notable limitations, such as in distinguishing regular graphs. Generalizations of this procedure, known as the $k$-WL test, establish a hierarchy of progressively more powerful tests. At the same time, several graph characteristics are known to be more expressive than the 1-WL test. Consequently, combining MPGNNs with encodings based on these characteristics can enhance their expressivity (Southern et al., 2023; Fesser & Weber, 2024a; Bouritsas et al., 2022). We conduct an expressivity analysis in Apx. E.

---

[1]With the exception of PhenomNN (Wang et al., 2023).

## 3 Hypergraph-level Encodings

### 3.1 Laplacian Eigenvectors

Let $D_v$ be the degree matrix, and $A$ be the adjacency matrix. The Graph Laplacian $\Delta = D_v - A$ is a classical graph characteristic that is often leveraged for the design of encodings. *Laplacian Eigenvector PE (LAPE)* are defined as

$$p_i^{\text{LapPE}} = (U_{i1}, U_{i2}, \ldots, U_{ik})^T \in \mathbb{R}^k \,, \tag{1}$$

where $\Delta = U^T \Lambda U$ is a spectral decomposition; $k$ is a hyperparameter. Note that the eigenvectors are only defined up to $\pm 1$; we follow the convention in (Dwivedi et al., 2021) and apply random sign flips.

In order to define a hypergraph-level extension of LAPE, we have to consider first the choice of Laplacian. We focus on the Hodge Laplacian here, but discuss other choices, specifically the normalized hypergraph Laplacian and random-walk Laplacian, in Apx. A.4. Our choice of the Hodge Laplacian is motivated by its desirable properties, including that it is symmetric.

**Definition 3.1. (Hodge Laplacian).** Let $B_1$ denote an incidence matrix whose entries indicate relations between nodes and hyperedges. If a node $i$ is on the boundary of a hyperedge $j$, the relation is expressed as $i \prec j$.

$$(B_1)_{i,j} = \begin{cases} 1 \text{ if } i \prec j \\ 0 \text{ otherwise} \end{cases} \in \mathbb{R}^{|V| \times |F|} \,. \tag{2}$$

The 0- and 1-Hodge Laplacian are given by $H_0 = B_1^T B_1$ and $H_1 = B_1 B_1^T$.

We define the *Hodge-Laplacian Positional Encoding (H-$k$-LAPE)* in analogy to Eq. 1 using the top $k$ eigenvectors of the Hodge Laplacian. We show below that the additional higher-order information captured by H-$k$-LAPE, but not by $k$-LAPE or standard message-passing, provably enhances the representational power of the architecture. The proofs in this and subsequent sections refer to graphs in the BREC dataset (Wang & Zhang, 2023), more information on which is provided in the Apx. E.1. We outline the computations of the different encodings for a specific pair of the BREC dataset in Apx. F.2.

**Theorem 3.2.** *(H-$k$-LAPE Expressivity).* *For any $k$, MPGNNs with H-$k$-LAPE are strictly more expressive than the 1-WL test and hence MPGNNs without encodings. Furthermore, there exist graphs which can be distinguished using H-$k$-LAPE, but not using $k$-LAPE.*

*Remark* 3.3. In the proof of our theorems, we follow a two-step strategy. We first note that the seminal work by Xu et al. (2018) has established that standard MPGNNs are at most as expressive as the 1-WL test, with GIN constructed specifically to be as expressive as the 1-WL test. It can be shown via an adaptation of Xu et al. (2018) that adding encodings does not decrease MPGNNs expressivity, i.e., GIN -and thus MPGNNs- with encodings are at least as expressive as the 1-WL test. To establish strictly better expressivity, it is sufficient to identify a set of non-isomorphic graphs that cannot be distinguished by the 1-WL test, but that differ in their encodings.

*Proof.* Pair 0 of the "Basic" category in BREC (Apx. E.1) is a pair of non-isomorphic, 1-WL indistinguishable graphs (see Fig. 1). The pair is 1-LAPE-indistinguishable, but can be distinguished with H-1-LAPE (see Apx. F.2). $\qquad\square$

*Remark* 3.4. **(H-LAPE Complexity).** Computing a full spectral decomposition of $\Delta$ has complexity $O(|V|^3)$, where $|V|$ is the number of nodes in the input hypergraph. However, by exploiting sparsity and the fact that we only require the top eigenvectors, Lanczos' algorithm can be used to compute H-$k$-LAPE in $O(|F|k)$.

### 3.2 Random Walk Transition Probabilities

Another widely used positional encoding, Random Walk PE (RWPE), is defined using the probability of a random walk revisiting node $i$ after $1, 2, \ldots, k$ steps, formally

$$p_i^{k\text{-RWPE}} = \left( RW_{ii}, RW_{ii}^2, \ldots, RW_{ii}^k \right)^T \in \mathbb{R}^k \,, \tag{3}$$

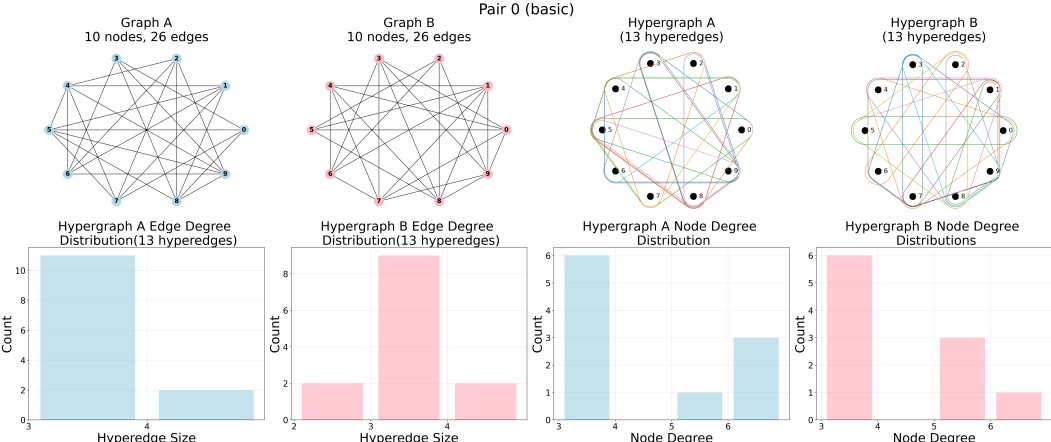

Figure 1: A pair of graph from the BREC "Basic" category (top left), the graphs' liftings (top right), the hyperedge sizes (bottom left) and node degrees (bottom right).

where $k$ is a hyperparameter. Since the return probabilities depend on the graph's topology, they capture crucial structural information. Notably, $k$-RWPE does not suffer from sign ambiguity like LAPE, instead providing a unique node representation whenever nodes have topologically distinct $k$-hop neighborhoods.

We define an analogous notion of PEs at the hypergraph level. We consider the following notion:

**Definition 3.5. (Random Walks on Hypergraphs (Coupette et al., 2022)).** We define Equal-Nodes Random Walks (EN) and Equal-Edges Random Walks (EE), which induce the following two measures:

$$\mu_i^{\text{EN}}(j) = \begin{cases} \frac{1}{|\mathcal{N}_i|}, \text{ if } j \in \mathcal{N}_i \\ 0 \text{ otherwise} \end{cases}, \qquad \mu_i^{\text{EE}}(j) = \begin{cases} \mathbb{P}^{\text{EE}}(i \to j), \text{ if } j \in \mathcal{N}_i \\ 0 \text{ otherwise} \end{cases}; \qquad (4)$$

where $\mathcal{N}_i$ are the neighbors of $i$ and transition probabilities are given by

$$\mathbb{P}^{\text{EE}}(i \to j) = \frac{1}{|\{e|i \in e, |e| \geq 2\}|} \sum_{\{e|\{i,j\} \subseteq e\}} \frac{1}{|e| - 1} .$$

For an EN random walk, considering a move from node $i$, we pick one of the neighbors of node $i$ at random. For the EE scheme, we first pick a hyperedge that $i$ belongs to at random and then pick one of the nodes in the hyperedge at random.

We can now define *Hypergraph Random Walk Positional Encodings* (H-$k$-RWPE) in analogy to $k$-RWPE. Again, we can show that H-$k$-RWPE provably enhances the representational power of an MPGNN, beyond those of $k$-RWPE.

**Theorem 3.6. (H-$k$-RWPE Expressivity).** *For $k \geq 2$, MPGNNs with H-$k$-RWPE are strictly more expressive than the 1-WL test and hence than MPGNNs without encodings. There exist graphs which can be distinguished using H-$k$-RWPE, but not using graph-level $k$-RWPE.*

*Proof.* Pair 0 of the "Basic" category in BREC is a pair of non-isomorphic graphs that is not 1-WL-distinguishable. The pair cannot be distinguished with graph-level 2-RWPE, but can be distinguished using the H-2-RWPE encodings computed at the hypergraph level (see Apx. F.2). □

*Remark* 3.7. Note that $k$-RWPE is less expressive that $(k + 1)$-RWPE and H-$k$-RWPE is less expressive than $(k + 1)$-H-RWPE.

*Remark* 3.8. **(H-$k$-RWPE Complexity).** Computing H-$k$-RWPE scales as $O(|V|d_{\max}^k)$, where $d_{\max}$ is the highest node degree in the input hypergraph.

### 3.3 Local Curvature Profiles

Recently it was shown that discrete Ricci curvature yields an effective structural encoding at the graph level (Fesser & Weber, 2024a). Ricci curvature is a classical tool from Differential Geometry that allows for characterizing local and global properties of geodesic spaces. Discrete analogues of Ricci curvature (Forman, 2003; Ollivier, 2007) have been studied extensively on graphs and, more recently, on hypergraphs (Leal et al., 2021; Coupette et al., 2022; Saucan & Weber, 2019). Here, we focus on defining hypergraph-level curvatures, we defer all details on graph-level notions to Apx. A.5.

We restrict ourselves to two notions of discrete Ricci curvature, originally introduced by Forman (Forman, 2003) and Ollivier (Ollivier, 2007), which have previously been considered for graph-level encodings. We begin with Forman's curvature:

**Definition 3.9. (Forman's Ricci Curvature on Hypergraphs (H-FRC) (Leal et al., 2021)).** The H-FRC of a hyperedge $e$ is defined as $F(e) = \sum_{k \in e}(2 - d_k)$.

Ollivier's Ricci curvature derives from a fundamental relationship between Ricci curvature and the behavior of random walks on geodesic spaces. To define an analogous notion on hypergraphs, we leverage again the previously introduced notions of random walks (Coupette et al., 2022).

**Definition 3.10. (Ollivier's Ricci Curvature on Hypergraphs (H-ORC) (Coupette et al., 2022)).** The H-ORC of a subset $s$ of nodes on a hypergraph is defined as:

$$\kappa(s) = 1 - \frac{AGG(s)}{d(s)} , \tag{5}$$

where $d(s) = \{\max d(i,j) | \{i,j\} \subseteq s\}$ and where $d(i,j)$ is the length of a shortest path connecting nodes $i, j$. We define for a hyperedge $e$

$$\kappa(e) = 1 - AGG(e) . \tag{6}$$

Here, $AGG(\cdot)$ denotes an aggregation function.

Different types of aggregations could be considered for the choice of $AGG(\cdot)$. Here, we choose $AGG(\cdot)$ to be the average of the Wasserstein distances between all pairs $\{i, j\}$ in a hyperedge $e$, i.e.,

$$AGG(e) = \frac{1}{\binom{|e|}{2}} \sum_{\{i,j\} \subseteq e} W_1(\mu_i, \mu_j) . \tag{7}$$

We can now define the actual encoding, extending Local Curvature Profiles (LCP) (Fesser & Weber, 2024a), computed at the graph level, to hypergraphs.

**Definition 3.11. (Hypergraph Curvature Profile (HCP)).** For $v \in V$ let $\text{CMS}(v)$ denote a *curvature multi-set* consisting of the curvatures of all hyperedges containing $v$, $\text{CMS}(v) = \{\kappa(e) : v \in e, e \in F\}$, where $\kappa$ may be chosen to denote either H-FRC or H-ORC. We define HCP as the following five summary statistics of $\text{CMS}(v)$:

$$\text{HCP}(v) = [\min(\text{CMS}(v)), \max(\text{CMS}(v)), \text{mean}(\text{CMS}(v)), \text{median}(\text{CMS}(v)), \text{std}(\text{CMS}(v))] . \tag{8}$$

As for the other proposed encodings, we investigate the expressivity of HCP. Note that ORC computed at the graph level is by itself very expressive, leading to LCP provably enhancing the expressivity of MPGNNs. In fact, there exist variants of ORC which can distinguish graphs that are not 3-WL distinguishable (Southern et al., 2023). However, the same is not true for graph-level FRC. This merits a closer analysis of HCP where $\kappa$ is chosen to be the H-FRC.

**Theorem 3.12.** *(HCP Expressivity). MPGNNs with HCP ($\kappa$ denoting H-FRC) are strictly more expressive than the 1-WL test and hence than MPGNNs without encodings. In contrast, leveraging LCP with standard FRC at the graph level does not enhance expressivity.*

*Proof.* Consider the 4 by 4 Rook and the Shrikhande graphs, which cannot be distinguished by the $k$-WL test for $k \leq 3$. All nodes in both graphs have identical LCP-FRC, namely $[-8, -8, -8, -8, 0]$. This is because all nodes have degree 6, consequently their FRC is $-8$. However, when computing HCP-FRC on the lifted hypergraphs the curvatures differ: In the Rook graph, all nodes have HCP-FRC

$[0, 0, 0, 0, 0]$, whereas in the Shrikhande graph all nodes have HCP-FRC $[-12, -12, -12, -12, 0]$ (see Apx. F.1). Furthermore, it is possible to find non-isomorphic graphs with the same LCP, but different HCP (even up to scaling): Pair 0 of the "Basic" category in BREC is an example where both graphs have the same LCP, but different HCP (even up to scaling) (see Apx. F.2 for additional details). □

*Remark* 3.13. **(HCP Complexity).** Computing the H-FRC and hence the HCP-FRC scales as $O(|F|e_{\max})$, where $e_{\max}$ denotes the size of the largest hyperedge. On the other hand, computing H-ORC incurs significant computational cost: The computation of the $W_1$-distance, which scales as $(|F|e_{\max}^3)$, introduces a significant bottleneck. Hence, HCP-FRC has significant scaling advantages over HCP-ORC.

### 3.4 Local Degree Profile

Lastly, we define a hypergraph-level notion of *Local Degree Profiles (LDP)* (Cai & Wang, 2018), which captures structural information encoded in the node degree distribution over a node's 1-hop neighborhood. We consider the multi-set of node degrees in the 1-hop neighborhood of a node $v$, i.e., $\mathrm{DN}(v) = \{d_u | u \in \mathcal{N}_v\}$ and define

$$\mathrm{LDP}(v) = [d_v, \min(\mathrm{DN}(v), \max(\mathrm{DN}(v)), \mathrm{mean}(\mathrm{DN}(v)), \mathrm{median}(\mathrm{DN}(v)), \mathrm{std}(\mathrm{DN}(v))] . \quad (9)$$

An analogous notion on the hypergraph level (H-LDP) can be defined by a simple extension. Again, H-LDP exhibits improved expressivity:

**Theorem 3.14.** *(H-LDP Expressivity). MPGNNs with H-LDP are strictly more expressive than the 1-WL test and hence than MPGNNs without encodings. There exist graphs which can be distinguished using H-LDP, but not using LDP.*

*Proof.* The 4 by 4 Rook graph and the Shrikhande graph cannot be distinguished by LDP, as all nodes the same degree, resulting in LDPs $[6, 6, 6, 6, 6, 0]$. However, they can be distinguished using H-LDP: The nodes in the Rook graph have H-LDP $[2, 2, 2, 2, 2, 0]$, the nodes in the Shrikhande graph $[6, 6, 6, 6, 6, 0]$ (see Apx. F.1). Furthermore, it is possible to find non-isomorphic graphs with the same LDP, but different H-LDP even up to scaling: Pair 0 of the "Basic" category in BREC is an example, where both graphs have identical LDPs, but different H-LDPs, even up to scaling. For more details, see Fig. 1 and Apx. F.2. □

*Remark* 3.15. **(H-LDP Complexity).** Computing H-LDP scales as $O(|F|e_{\max})$. We only need to count the degree for each node and save the statistics of 1-neighborhood for each node.

*Remark* 3.16. We observe that in the examples demonstrating the enhanced representational power of HCP and H-LDP, the respective profiles are scalar multiples of each other. It is common in hypergraph architectures to normalize node attributes during preprocessing, which would obscure the structural differences captured by the two encodings. However, we emphasize that no such preprocessing is applied in our experiments.

## 4 Experiments

### 4.1 Experimental setup

Throughout all of our experiments, we treat the computation of encodings as a preprocessing step, which is first applied to all graphs in the data sets considered. We then train a GNN on a part of the preprocessed graphs and evaluate its performance on a withheld set of test graphs (nodes in the case of node classification). Settings and optimization hyperparameters are held constant across baseline models for all encodings, so that hyperparameter tuning can be ruled out as a source of performance gain. We obtain the settings for the individual encoding types via hyperparameter tuning. For all preprocessing methods and hyperparameter choices, we record the test set performance of the settings with the best validation performance. As there is a certain stochasticity involved, especially when training neural networks, we accumulate experimental results across 50 random trials. We report the mean test accuracy, along with the 95% confidence interval for the node classification tasks on hypergraph datasets in Tab. 1 and for tasks on graph datasets in Tab. 2, 3, 10, 11, 12, and 13. For Peptides-func, we report average precision and for Peptides-struct the mean absolute error (MAE). Details on all datasets can be found in Apx. C.1.

| Model (Encodings) | citeseer-CC (↑) | cora-CA (↑) | cora-CC (↑) | pubmed-CC (↑) | DBLP (↑) |
|---|---|---|---|---|---|
| GCN (No Encoding) | 69.28 ± 0.28 | 76.51 ± 0.82 | 75.43 ± 0.26 | 84.66 ± 0.49 | 75.66 ± 0.81 |
| GCN (HCP-FRC) | 71.03 ± 0.51 | 78.43 ± 0.76 | 76.61 ± 0.31 | 84.78 ± 0.57 | 76.49 ± 0.90 |
| GCN (HCP-ORC) | 70.89 ± 0.54 | 79.25 ± 0.81 | 76.09 ± 0.70 | 85.12 ± 0.61 | 76.57 ± 0.85 |
| GCN (EE H-19-RWPE) | 69.63 ± 0.71 | 76.84 ± 0.69 | 75.92 ± 0.28 | 86.24 ± 0.63 | 76.18 ± 0.88 |
| GCN (EN H-19-RWPE) | 68.85 ± 0.91 | 77.19 ± 0.64 | 75.33 ± 0.35 | 86.53 ± 0.61 | 76.76 ± 0.84 |
| GCN (Hodge H-20-LAPE) | 69.61 ± 0.45 | 79.61 ± 0.85 | 75.62 ± 0.31 | 86.06 ± 0.52 | 77.48 ± 0.93 |
| GCN (Norm. H-20-LAPE) | 69.13 ± 0.77 | 78.13 ± 0.79 | 76.18 ± 0.29 | 85.78 ± 0.55 | 76.92 ± 0.88 |
| UniGCN (No Encoding) | 63.36 ± 1.76 | 75.72 ± 1.16 | 71.10 ± 1.37 | 75.32 ± 1.09 | 71.05 ± 1.40 |
| UniGCN (HCP-FRC) | 61.20 ± 1.83 | 74.64 ± 1.45 | 68.98 ± 1.59 | 67.37 ± 1.73 | 71.02 ± 1.43 |
| UniGCN (HCP-ORC) | 61.81 ± 1.70 | 75.03 ± 1.33 | 70.42 ± 1.17 | 71.64 ± 1.52 | 70.69 ± 1.62 |
| UniGCN (EE H-19-RWPEE) | 63.29 ± 1.52 | 75.34 ± 1.28 | 71.13 ± 1.24 | 74.61 ± 1.18 | 71.21 ± 1.53 |
| UniGCN (EN H-19-RWPEE) | 63.09 ± 1.62 | 75.30 ± 1.37 | 71.21 ± 1.34 | 74.61 ± 1.09 | 71.26 ± 1.47 |
| UniGCN (Hodge H-20-LAPE) | 63.46 ± 1.58 | 75.64 ± 1.37 | 71.31 ± 1.19 | 75.37 ± 1.01 | 70.71 ± 1.61 |
| UniGCN (Norm. H-20-LAPE) | 63.41 ± 1.61 | 75.55 ± 1.48 | 71.20 ± 1.24 | 75.30 ± 1.01 | 71.10 ± 1.33 |

Table 1: GCN and UniGCN performance on hypergraph datasets with different hypergraph encodings. We report mean accuracy and standard deviation over 50 runs. In our tables, the best value is in **bold** and the second best in blue. In case of overlap, we underline. For example, for citeseer-CC, GCN (HCP-ORC) is underlined because it is the tied best model-encoding pair, up to 0.51 accuracy (the accuracy of the best pair GCN (HFC-FRC)).

| Model (Encodings) | Collab (↑) | Imdb (↑) | Reddit (↑) | Peptides-f (↑) | Peptides-s (↓) |
|---|---|---|---|---|---|
| GCN (No Encoding) | 61.94 ± 1.27 | 48.10 ± 1.02 | 67.87 ± 1.38 | 0.532 ± 0.005 | 0.266 ± 0.002 |
| GCN (LCP-FRC) | 68.36 ± 1.13 | 63.42 ± 1.47 | 79.53 ± 1.62 | 0.537 ± 0.006 | 0.261 ± 0.003 |
| GCN (LCP-ORC) | 70.48 ± 0.97 | 67.93 ± 1.55 | 80.75 ± 1.54 | 0.561 ± 0.005 | 0.252 ± 0.004 |
| GCN (19-RWPE) | 49.63 ± 2.38 | 50.41 ± 1.26 | 78.93 ± 1.60 | 0.538 ± 0.007 | 0.265 ± 0.003 |
| GCN (20-LAPE) | 58.33 ± 1.64 | 48.82 ± 1.31 | 77.26 ± 1.58 | 0.534 ± 0.006 | 0.258 ± 0.003 |
| GCN (HCP-FRC) | 72.03 ± 0.51 | 64.64 ± 0.88 | 82.09 ± 0.58 | 0.559 ± 0.004 | 0.255 ± 0.004 |
| GCN (HCP-ORC) | 70.82 ± 0.68 | 66.16 ± 0.75 | 80.35 ± 0.72 | 0.559 ± 0.004 | 0.258 ± 0.003 |
| GCN (EE H-19-RWPE) | 69.63 ± 0.71 | 73.96 ± 0.65 | 82.79 ± 0.62 | 0.546 ± 0.006 | 0.263 ± 0.003 |
| GCN (EN H-19-RWPE) | 68.85 ± 0.91 | 73.84 ± 0.48 | 83.30 ± 0.79 | 0.549 ± 0.005 | 0.263 ± 0.003 |
| GCN (Hodge H-20-LAPE) | 69.61 ± 0.45 | 71.38 ± 0.75 | 79.46 ± 0.82 | 0.557 ± 0.005 | 0.254 ± 0.003 |
| GCN (Norm. H-20-LAPE) | 69.13 ± 0.77 | 71.05 ± 0.82 | 80.08 ± 0.67 | 0.557 ± 0.006 | 0.253 ± 0.003 |

Table 2: GCN performance with graph level encodings (top) and hypergraph level encodings (bottom). We report mean and standard deviation across 50 runs.

## 4.2 Comparison of Hypergraph- and Graph-level Architectures

We begin by comparing the utility of our encodings for message-passing architectures that operate at the graph or at the hypergraph level. Hypergraph neural networks are predominantly used for node classification in hypergraphs. In fact, we are not aware of hypergraph classification datasets analogous to the graph datasets used in the previous subsection. As such, we choose five common hypergraph node classification datasets: Cora-CA, Cora-CC, Citeseer, DBLP, and Pubmed. We use clique expansion to convert these hypergraphs into graphs (empirically, we found this to be the best performing expansion - see Apx. A.1) and train GCN on them with either no encoding or one of our hypergraph encodings. As a hypergraph-level message-passing architecture, we use UniGCN (Huang & Yang, 2021). Additional experiments with UniGIN, UniGAT, and other architectures are presented in Apx. G, along with a detailed explanation of the clique expansions we use.

**Graph-level message-passing Benefits from Hypergraph-level Encodings.** Our results are presented in Tab. 1. Somewhat surprisingly, we note that even on these datasets, which are originally hypergraphs, GCN with no encodings outperforms UniGCN. Perhaps even more surprising, UniGCN does not seem to benefit from any of the encodings provided. Apx. G shows that the same holds true for UniGIN and UniGAT. GCN on the other hand clearly benefits from (most) hypergraph-level encodings, although admittedly less so than when used for graph classification. Previous work has reported similar differences in the utility of encodings for graph and node classification tasks. Overall, we take our results in this subsection and in Apx. G as evidence that our proposed hypergraph-level encodings present a strong alternative to established message-passing architectures at the hypergraph level.

## 4.3 Hypergraph-level Encodings Capture Higher-order Information Effectively

We now evaluate to what extent our hypergraph encodings can be used for datasets that are originally graph-structured. We lift these graphs to the hypergraph level (see. Apx. A.2 for details) and compare

| Model (Encodings) | Reddit ($\uparrow$) | Enzymes ($\uparrow$) | Proteins ($\uparrow$) | Peptides-f ($\uparrow$) | Peptides-s ($\downarrow$) |
|---|---|---|---|---|---|
| GPS (No Encoding) | $80.94 \pm 1.42$ | $46.83 \pm 1.14$ | $\mathbf{74.10 \pm 0.98}$ | $0.593 \pm 0.009$ | $0.262 \pm 0.003$ |
| GPS (LCP-FRC) | $80.53 \pm 1.55$ | $43.75 \pm 1.39$ | $73.38 \pm 1.07$ | $0.598 \pm 0.010$ | $0.257 \pm 0.003$ |
| GPS (LCP-ORC) | $82.83 \pm 1.47$ | $48.51 \pm 1.58$ | $\mathbf{74.88 \pm 1.20}$ | $\mathbf{0.613 \pm 0.010}$ | $\mathbf{0.252 \pm 0.003}$ |
| GPS (19-RWPE) | $81.92 \pm 1.31$ | $\underline{\mathbf{51.09 \pm 1.64}}$ | $71.92 \pm 1.18$ | $0.594 \pm 0.011$ | $0.257 \pm 0.003$ |
| GPS (20-LAPE) | $82.05 \pm 1.29$ | $\underline{42.90 \pm 1.35}$ | $71.46 \pm 1.25$ | $0.599 \pm 0.011$ | $\mathbf{0.253 \pm 0.003}$ |
| GPS (HCP-FRC) | $81.68 \pm 1.16$ | $47.66 \pm 0.92$ | $74.50 \pm 1.13$ | $0.604 \pm 0.010$ | $\mathbf{0.254 \pm 0.003}$ |
| GPS (HCP-ORC) | $83.07 \pm 1.24$ | $48.19 \pm 1.31$ | $\mathbf{74.52 \pm 1.20}$ | $\mathbf{0.609 \pm 0.010}$ | $0.254 \pm 0.004$ |
| GPS (EE H-19-RWPE) | $\underline{84.04 \pm 1.07}$ | $51.83 \pm 1.07$ | $75.08 \pm 1.14$ | $\underline{0.615 \pm 0.009}$ | $0.251 \pm 0.003$ |
| GPS (EN H-19-RWPE) | $\mathbf{84.25 \pm 1.13}$ | $\underline{51.28 \pm 1.12}$ | $74.82 \pm 1.11$ | $\mathbf{0.617 \pm 0.010}$ | $0.252 \pm 0.003$ |
| GPS (Hodge H-20-LAPE) | $\underline{83.97 \pm 1.21}$ | $47.44 \pm 1.16$ | $\mathbf{73.95 \pm 1.08}$ | $0.602 \pm 0.010$ | $0.252 \pm 0.003$ |
| GPS (Norm. H-20-LAPE) | $\underline{83.85 \pm 1.18}$ | $47.78 \pm 0.98$ | $\mathbf{74.03 \pm 1.10}$ | $0.604 \pm 0.010$ | $\mathbf{0.254 \pm 0.002}$ |

Table 3: GPS performance with graph level encodings (top) and hypergraph level encodings (bottom). We report mean and standard deviation across 50 runs.

against encodings computed at the graph level. Tables 2, 10 and 3, 11 report results for GCN and GPS, respectively; additional results with GIN can be found in Apx. D.

**Performance Gains with Hypergraph-level Encodings.** We note several things: 1) adding encodings is beneficial in nearly all scenarios, 2) encodings computed at the hypergraph level are always at least as beneficial as their cousins computed at the graph level (e.g. H-RWPE is at least as useful as RWPE), and 3) on social network datasets (Collab, Imdb, and Reddit), hypergraph encodings provide the largest performance boosts, often by a wide margin. This aligns with our intuition, as social networks can often naturally be thought of as hypergraphs.

**Positional vs Structural Encodings.** Our results with GPS confirm our observations with GCN. Hypergraph-level encodings significantly boost performance on almost all datasets (only Proteins is not statistically significant) and are generally more useful than their graph-level analogues. Further, while GCN usually performed best with local structural encodings such as the Local Curvature Profile, GraphGPS seems to benefit more from global positional encodings such as (Hypergraph-) Random Walk Positional Encodings. This aligns with previous findings in the literature using graph-level encodings (Fesser & Weber, 2024a).

**Utility beyond Weisfeiler-Leman.** Our previous results on the BREC dataset indicate that much of the utility of our hypergraph-level encodings can perhaps be attributed to improving the expressivity of GCN and GPS. To better quantify this, we run an additional suite of experiments on the Collab, Imdb, and Reddit datasets using the GIN. As noted previously, GIN is provably as powerful as the 1-WL test and therefore more expressive than GCN. Our results in Apx. D show that GIN has indeed a higher baseline accuracy (without encodings) than GCN, and benefits significantly less from encodings than both GCN and GPS. Nevertheless, our hypergraph-level encodings significantly boost performance. We take this as evidence that providing information from domains other than the computational domain (graphs in our setting) provides benefits beyond increased expressivity.

# 5 Discussion

In this study, we proposed hypergraph-level encodings as an approach for leveraging higher-order relational information. Our results show that combining hypergraph encodings with graph-level architectures leads to enhanced representational power and significant performance gains in "multi-way" relational learning tasks.

**Lessons for Model Design.** Our study is motivated by the observation that graph-level architectures applied to a hypergraphs' clique expansions frequently outperform hypergraph-level architectures, even when the inputs are naturally parametrized as hypergraphs. While hypergraph-level encodings do not significantly enhance the performance of hypergraph architectures, they can lead to substantial performance gains when used in graph architectures. Notably, random walk-based (H-$k$-RWPE) and curvature-based encodings (HCP) are particularly effective across data sets. These findings suggest a graph-level architecture augmented with hypergraph-level encodings as a suitable model choice for higher-order learning tasks.

**Limitations.** A key limitation of this study, and many of the related works, is a lack of benchmarks consisting of *true* hypergraph-structured data. Many of the existing data sets consists of graphs that are reparametrized ("lifted") to hypergraphs, or hypergraphs that can be easily reparametrized

as graphs. This suggests the establishment of better benchmark as a key direction for future work. Given the promise of topological deep learning for scientific machine learning, we envision future benchmarks that are based on scientific data, where multi-way interactions are known to arise naturally (Garcia-Chung et al., 2023; Gjorgjieva et al., 2011). Another limitation of this study arises in the choice of hypergraph architectures. While our selection was guided by top-performing models in recent benchmarks (Huang & Yang, 2021; Telyatnikov et al., 2024), a more comprehensive analysis could further strengthen the validity of the reported observations.

**Other Future Directions.** Despite the aforementioned caveats regarding datasets and the breadth of architectures included in this study, our observations raise questions about the effectiveness of existing message-passing schemes on hypergraphs. We believe that a thorough analysis of these architectures' ability to effectively encode higher-order information into learned representations is an important direction for future work. A possible lens for such an investigation could be graph reasoning tasks, as previously suggested in (Luo et al., 2023).

Additionally, the negative results observed regarding hypergraph-level encodings paired with hypergraph-level architectures warrant further exploration. Specifically, understanding how to effectively augment hypergraph inputs with structural and positional information that can be leveraged by hypergraph-level architectures is a promising direction for further study.

Lastly, while this study primarily focused on hypergraph learning, there are several other topological domains of interest, including simplicial complexes, polyhedral complexes, and more general CW complexes. Extending the present study to these domains represents another interesting avenue for further investigation.

# 6 Broader Impacts

This paper presents work whose goal is to advance our theoretical understanding of Machine Learning. There are many potential societal consequences of our work, none of which we feel must be specifically highlighted here.

# 7 Acknowledgments

LF was supported by a Kempner Graduate Fellowship. MW was partially supported by NSF award CBET-2112085 and DMS-2406905, and an Alfred P. Sloan Research Fellowship in Mathematics. Some of the computations in this paper were run on the FASRC Cannon cluster supported by the FAS Division of Science Research Computing Group at Harvard University.

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

# A  Extended Background

## A.1  Hypergraph Expansions

There exist several expansion techniques for reparametrizing hypergraphs as graphs. Here, we focus on clique expansion, which we empirically found to be the best performing expansion. For more details see, e.g., (Sun et al., 2008).

To reparametrize a hypergraph $H = (V, X, F)$ as a graph via *clique expansion*, we define $G = (V, X, E)$ where $E = \{\{u, v\} | \{u, v\} \subseteq e, e \in F\}$. An example is given in Fig. 2.

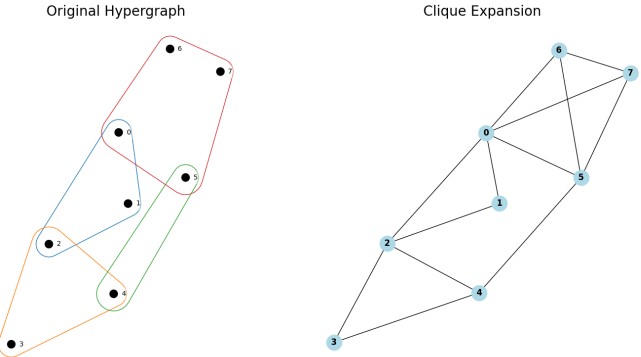

Figure 2: Example of a clique expansion of a hypergraph to a graph. The plots are created using NetworkX (Hagberg et al., 2008) and HyperNetX (Praggastis et al., 2023).

## A.2  Lifting graphs to hypergraphs

The term "lifting" refers generally to the reparametrization of one topological domain to another, usually one that captures richer higher-order information. In our setting we lift graphs to hypergraphs by adding hyperedges to groups of nodes that are pairwise interconnected. An example of a lift of a graph to a hypergraph is shown in Fig. 3.

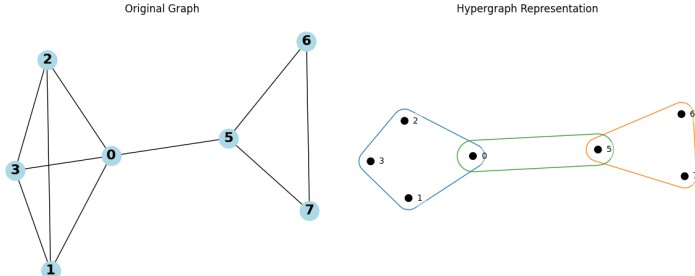

Figure 3: Lifting of a graph to a hypergraph.

## A.3  Weighted-Edges (WE) Hypergraph Random walks

We define Weighted-Edges Random Walks (WE), which induce the following measure

$$\mu_i^{\text{WE}}(j) = \begin{cases} \mathbb{P}^{\text{WE}}(i \rightarrow j), \text{ if } j \in \mathcal{N}_i \\ 0 \text{ otherwise} \end{cases}, \tag{10}$$

where $\mathcal{N}_i$ are the neighbors of $i$ and transition probabilities are given by

$$\mathbb{P}^{\text{WE}}(i \rightarrow j) = \frac{1}{\sum_{\{f | i \in f\}} (|f| - 1)} \sum_{\{e | \{i,j\} \subseteq e\}} 1 . \tag{11}$$

The probability of picking a hyperedge is directly proportional to the number of nodes in the hyperedge minus 1.

## A.4 Laplacians

Several notions of Laplacians have been studied on hypergraphs. In this work, we consider two types of Laplacians on graphs for implementing H-LAPE, the Hodge-Laplacian, with we defined in the main text, and the normalized Laplacian, which we discuss below. Additionally, we comment on random walks hypergraphs Laplacians. However, since they need not be symmetric, there are not suitable for use in H-LAPE. Nonetheless, their spectrum provides an additional means for defining structural encodings.

### A.4.1 Normalized graph and hypergraph Laplacian

For graphs, the *symmetrically normalized graph Laplacian* is defined as

$$I - D_v^{-1/2}AD_v^{-1/2} = D_v^{-1/2}LD_v^{-1/2} \; , \tag{12}$$

where $L = D_v - A$ is the standard graph Laplacian.

The *normalized hypergraph Laplacian* (Zhou et al., 2006; Feng et al., 2019) is defined as

$$\Delta = I - D_v^{-1/2}B_1 D_e^{-1} B_1^T D_v^{-1/2} = D_v^{-1/2}(D_v - B_1 D_e^{-1} B_1^T)D_v^{-1/2} \; , \tag{13}$$

where $D_v$ and $D_e$ are the diagonal node and edge degree matrices. The Dirichlet energy $E(f)$ of a scalar function on a hypergraph is defined as

$$E(f) = \frac{1}{2} \sum_{e \in E} \sum_{\{u,v\} \subseteq e} \frac{1}{|e|} \left( \frac{f(u)}{\sqrt{d(u)}} - \frac{f(v)}{\sqrt{d(v)}} \right)^2 \; . \tag{14}$$

The normalized hypergraph Laplacian satisfies

$$E(f) = f^T \Delta f \; , \tag{15}$$

which establishes that the normalized hypergraph Laplacian is positive semi-definite (Zhou et al., 2006). The smallest eigenvalue of $\Delta$ is 0.

### A.4.2 Random walks hypergraph Laplacians

For a graph, the *random walk Laplacian* is defined as $L = I - D_v^{-1}A$, where, as usual, $D_v$ denotes the degree matrix and $A$ the adjacency matrix. The probability of a random walk transitioning from node $i$ to $j$ is given by $-L_{ij} = \frac{A_{ij}}{d_i}$. Mulas et al. (2022) introduce a generalized random-walk Laplacians on hypergraphs: For any random walk on a hypergraph, they define in analogy to the graph case

$$L_{ij} = \begin{cases} 1 \text{ if } i = j \\ -\mathbb{P}(i \to j) \end{cases} \; . \tag{16}$$

This random walk notion is equivalent to the EE scheme in Coupette et al. (2022), defined in the main text: Starting at $v$, choose one of the hyperedges containing $v$ with equal probability, then select any of the vertices of the chosen hyperedge (other than $v$) with equal probability. Formally, we write

$$\mathbb{P}(i \to j) = \frac{\mathcal{A}_{ij}}{\mathcal{D}_{ii}} \; . \tag{17}$$

A similar notion was previously studied in (Banerjee, 2021).

Note that the random-walk Laplacian need not be symmetric. As a result, it is not suitable for defining H-LAPE. However, in some recent works, the spectrum of the graph Laplacian, rather than its eigenvectors, have been used as SE (Kreuzer et al., 2021). An analogous notion can be defined at the hypergraph level, which we term *Hypergraph Laplacian Structural Encoding (H-LASE)*. We analyze the expressivity of such SEs, establishing that they a provably more expressive than the 1-WL test/ MPGNNs.

**Theorem A.1.** *(H-LASE Expressivity). MPGNNs with H-LASE are strictly more expressive than the 1-WL test and hence than MPGNNs without encodings. Further, there exist graphs, which can be distinguished using H-LASE, but not using standard, graph-level LASE.*

*Proof.* Consider the 4 by 4 Rook and Shrikhande graphs: the two graphs are isospectral using the Normalized, Random Walk and Hodge Laplacians. But the two graphs's liftings to hypergraphs are not isospectral for the Normalized Laplacian. $\square$.

## A.5 Discrete Curvature

**Forman's curvature.** Forman (2003) proposed a curvature definition on CW complexes, which derives from a fundamental relation between Ricci curvature and Laplacians (Bochner-Weizenböck identity). For a simple, undirected, and unweighted graph $G = (V, X, E)$, the Forman-Ricci curvature (FRC) of an edge $e = \{u, v\} \in E$ is given by:

$$\mathcal{FR}(u, v) = 4 - \deg(u) - \deg(v) \ .$$

The edge-based Forman curvature definition can be extended to capture curvature contributions from higher-order structures. Incorporating cycle counts up to order $k$ (denoted as $\mathcal{AF}_k$) has been shown to enrich the utility of the notion. Setting $k = 3$ and $k = 4$, the *Augmented Forman-Ricci curvature* is given by

$$\mathcal{AF}_3(u, v) = 4 - \deg(u) - \deg(v) + 3\triangle(u, v)$$
$$\mathcal{AF}_4(u, v) = 4 - \deg(u) - \deg(v) + 3\triangle(u, v) + 2\square(u, v) \ ,$$

where $\triangle(u, v)$ and $\square(u, v)$ represent the number of triangles and quadrangles containing the edge $\{u, v\}$. Prior work in the graph machine learning literature has demonstrated the effectiveness of these notions, e.g., (Fesser & Weber, 2024b; Fesser et al., 2024). To the best of our knowledge, characterizations of such higher-order information via hypergraph curvatures have not been previously studied in this literature.

**Ollivier's curvature.** We also consider the Ollivier-Ricci Curvature (ORC), a notion of curvature on metric spaces equipped with a probability measure (Ollivier, 2007). On graphs endowed with the shortest path distance $d(\cdot, \cdot)$, the ORC of an edge $\{i, j\}$ is defined as

$$\kappa(i, j) = 1 - \frac{W_1(\mu_i, \mu_j)}{d(i, j)} \ , \tag{18}$$

where $W_1$ denotes the Wasserstein distance. Recall that, in general, $W_1(\cdot, \cdot)$ between two probability distributions $\mu_1, \mu_2$ is defined as

$$W_1(\mu_1, \mu_2) = \inf_{\mu \in \Gamma(\mu_1, \mu_2)} \int d(x, y)\mu(x, y) \, dx \, dy \ , \tag{19}$$

where $\Gamma(\mu_1, \mu_2)$ is the set of measures with marginals $\mu_1, \mu_2$. In our case, the measures are defined by a uniform distribution over the 1-hop neighborhoods of the nodes $i$ and $j$.

*Remark* A.2. **(ORC in a general setting).** As noted in (Southern et al., 2023), the ORC can be defined in a more general setting on graphs, where the metric $d$ does not have to be the shortest-path distance. Furthermore, the probability measures need not be uniform probability measures in the 1-hop neighborhood of the node. This is shown to be beneficial in distinguishing 3-WL indistinguishable graphs using the ORC computed with respect to measures induced by $m$-hop random walks where $m > 1$.

# B  Architectures

| Architecture | Type | Level | Update Function |
|---|---|---|---|
| GCN (Kipf & Welling, 2016) | MP | graph | $X^{l+1} = \sigma\left(\tilde{D}^{-1/2}\tilde{A}\tilde{D}^{-1/2}X^l W^l\right)$ 
 $\tilde{A} = A + I_N$ 
 $\tilde{D}_{ii} = \sum_j \tilde{A}_{ij}$ |
| GIN (Xu et al., 2018) | MP | graph | $X^{l+1} = \text{MLP}^l\left((1+\epsilon)X^l + AX^l\right)$ |
| GPS (Rampášek et al., 2022) | hybrid (MP, GT) | graph | $X^{l+1}, E^{l+1} = \text{GPS}^l(X^l, E^l, A)$ 
 $X_M^{l+1}, E^{l+1} = \text{MPNN}_e^l(X^l, E^l, A)$ 
 $X_T^{l+1} = \text{GlobalAttn}^l(X^l)$ 
 $X^{l+1} = \text{MLP}(X_M^{l+1} + X_T^{l+1})$ |
| UniGCN (Huang & Yang, 2021) | MP | hypergraph | $\tilde{x}_i^{l+1} = \frac{1}{\sqrt{d_i+1}}\sum_{e\in\tilde{E}_i}\frac{1}{\sqrt{d_e}}W^l h_e^{l+1}$ |
| UniGIN (Huang & Yang, 2021) | MP | hypergraph | $\tilde{x}_i^{l+1} = W^l\left((1+\varepsilon)x_i^l + \sum_{e\in E_i}h_e^{l+1}\right)$ |
| UniGAT (Huang & Yang, 2021) | MP | hypergraph | $\alpha_{ie}^{l+1} = \sigma\left(a^T\left[W^l h_{\{i\}}^{l+1}; W^l h_e^{l+1}\right]\right),$ 
 $\tilde{\alpha}_{ie}^{l+1} = \frac{\exp(\alpha_{ie}^{l+1})}{\sum_{e'\in\tilde{E}_i}\exp(\alpha_{ie'}^{l+1})},$ 
 $\tilde{x}_i^{l+1} = \sum_{e\in\tilde{E}_i}\hat{\alpha}_{ie}^{l+1}W^l h_e^{l+1}$ |
| UniSAGE (Huang & Yang, 2021) | MP | hypergraph | $\tilde{x}_i^{l+1} = W^l(x_i^l + \text{AGGREGATE}(\{h_e^{l+1}\}_{e\in E_i}))$ |
| UniGCNII (Huang & Yang, 2021) | MP | hypergraph | $\hat{x}_i^{l+1} = \sqrt{\frac{1}{d_i+1}}\sum_{e\in\tilde{E}_i}\sqrt{\frac{1}{d_e}}h_e^{l+1}$ 
 $\tilde{x}_i^{l+1} = \left((1-\beta)I + \beta W^l\right)\left((1-\alpha)\hat{x}_i^{l+1} + \alpha x_i^0\right)$ 
 where $\alpha$ and $\beta$ are hyperparameters |
| HGNN (Feng et al., 2019) | MP | hypergraph | $X^{l+1} = \sigma\left(D_v^{-1/2}B_1 D_e^{-1}B_1^\top D_v^{-1/2}X^l W^l\right)$ |
| PhenomNN (Wang et al., 2023) | MP | hypergraph | $Y^{l+1} = \text{ReLU}\left((1-\alpha)Y^l + \alpha\tilde{D}^{-1}\tilde{Y}^l\right)$ 
 where $\tilde{Y}^l = f(X;W) + \lambda_0\tilde{Y}_C^l + \lambda_1(\bar{L}_S Y^l + \tilde{Y}_S^l)$ |

Table 4: Overview of Architectures. $W^l$ represents a trainable weight matrix for layer $l$. $\epsilon$ represents a learnable parameter. We use matrix notation except for Huang & Yang (2021) where we use vector notation.

## B.1  GNN architectures

### B.1.1  GCN

GCN extends convolutional neural networks to graph-structured data. It derives a shared representation by integrating node features and graph connectivity through message-passing. Mathematically, a GCN layer is expressed as

$$X^{l+1} = \sigma\left(\tilde{D}^{-1/2}\tilde{A}\tilde{D}^{-1/2}X^l W^l\right) ,$$

where $W^l$ is the learnable weight matrix at layer $l$, and $\tilde{D}^{-1/2}\tilde{A}\tilde{D}^{-1/2}$ is the normalized adjacency matrix of the original graph with added self-loops. This graph has adjacency matrix $\tilde{A} = A + I_N$ and node degree matrix $\tilde{D}$. The activation function $\sigma$ is typically chosen as ReLU or a sigmoid function.

### B.1.2  GIN

GIN is a message-passing graph neural network (MPGNN) designed for maximum expressiveness, meaning it can learn a broader range of structural patterns compared to other MPGNNs like GCN. GIN is inspired by the Weisfeiler-Leman (WL) graph isomorphism test. Formally, the GIN layer is given by

$$x_i^{l+1} = \text{MLP}^l\left((1+\epsilon)\cdot x_i^l + \sum_{j\in\mathcal{N}_i}x_j^l\right) , \tag{20}$$

where $x_i^l$ denotes the feature of node $i$ at layer $l$, $\mathcal{N}_i$ represents the neighbors of node $i$, and $\epsilon$ is a learnable parameter. The update step is carried out using a multi-layer perceptron MLP$(\cdot)$, which is a fully connected neural network.

### B.1.3 GraphGPS

GraphGPS is a hybrid graph transformer (GT) model that integrates MPGNNs with transformer layers to effectively capture both local and global patterns in graph learning. It enhances standard GNNs by incorporating positional encodings (which provide node location information) and structural encodings (which capture graph-theoretic properties of nodes). By alternating between GNN layers (for local aggregation) and transformer layers (for global attention), GraphGPS can efficiently model both short-range and long-range dependencies in graphs. It employs multi-head attention, residual connections, and layer normalization to maintain stability and improve learning performance. Mathematically, GraphGPS updates the node and edge features as follows:

$$X^{l+1}, E^{l+1} = \text{GPS}^l(X^l, E^l, A) \,,$$

computed as:

$$X_M^{l+1}, E^{l+1} = \text{MPNN}_e^l(X^l, E^l, A) \,,$$

$$X_T^{l+1} = \text{GlobalAttn}^l(X^l) \,,$$

$$X^{l+1} = \text{MLP}(X_M^{l+1} + X_T^{l+1}) \,.$$

where MLP$(\cdot)$ is a 2-layer Multi-Layer Perceptron (MLP) block. Note that we omit the batch normalization in this exposition.

## B.2 HNN architectures

Models on the hypergaph-level domain are approaches that preserve the hypergraph structure during learning (Kim et al., 2024). Huang & Yang (2021) proposes UniGCN, UniGIN, UniGAT, UniSAGE and UniGCNII, which directly generalize the classic GCN, GIN, GAT (Veličković et al., 2017),and GraphSAGE (Hamilton et al., 2017) and GNCII (Chen et al., 2020). We also study PhenomNN Wang et al. (2023) and ED-HNN Wang et al. (2022).

### B.2.1 UniGCN

UniGCN follows the two-phase scheme (described in 2.1) and sets the second aggregation function $\phi_2$ to be

$$\tilde{x}_i^{l+1} = \frac{1}{\sqrt{d_i + 1}} \sum_{e \in \tilde{E}_i} \frac{1}{\sqrt{\overline{d}_e}} W^l h_e^{l+1} \,, \tag{21}$$

where $W^l$ is a trainable weight matrix, $\overline{d}_e = \frac{1}{|e|} \sum_{i \in e} (d_i + 1)$ is the average degree of an hyperedge (after adding self-loops to the original hypergraph), and where $\tilde{\mathcal{N}}_i$ and $\tilde{E}(i)$ are the neighborhood of vertex $i$ and the incident hyperedges to $i$ after adding self loops.

### B.2.2 UniGIN

UniGIN also follows the two-phase scheme and sets the second aggregation function $\phi_2$ to be

$$\tilde{x}_i^{l+1} = W^l \left( (1 + \varepsilon) x_i^l + \sum_{e \in E_i} h_e^{l+1} \right) \,. \tag{22}$$

### B.2.3 UniGAT

UniGAT adopts an attention mechanism to assign importance score to each of the center node's neighbors (Huang & Yang, 2021). The attention mechanism is formulated as

$$\alpha_{ie}^{l+1} = \sigma \left( a^T \left[ W^l h_{\{i\}}^{l+1}; W^l h_e^{l+1} \right] \right) , \tag{23}$$

$$\tilde{\alpha}_{ie}^{l+1} = \frac{\exp(\alpha_{ie}^{l+1})}{\sum_{e' \in \tilde{E}_i} \exp(\alpha_{ie'}^{l+1})} , \tag{24}$$

$$\tilde{x}_i^{l+1} = \sum_{e \in \tilde{E}_i} \tilde{\alpha}_{ie}^{l+1} W^l h_e^{l+1} . \tag{25}$$

### B.2.4 UniSAGE

UniSAGE follows the two-phase scheme and sets the second aggregation function $\phi_2$ to be

$$\tilde{x}_i^{l+1} = W^l(x_i^l + \text{AGGREGATE}(\{h_e^{l+1}\}_{e \in E_i} \cdot)) \tag{26}$$

### B.2.5 UniGCNII

UniGCNII updates node features using:

$$\hat{x}_i^{l+1} = \sqrt{\frac{1}{d_i + 1}} \sum_{e \in \tilde{E}_i} \sqrt{\frac{1}{\bar{d}_e}} h_e^{l+1} , \tag{27}$$

$$\tilde{x}_i^{l+1} = \left((1 - \beta)I + \beta W^l\right) \left((1 - \alpha)\hat{x}_i^{l+1} + \alpha x_i^0\right) , \tag{28}$$

where $\alpha$ and $\beta$ are hyperparameters.

### B.2.6 HGNN

HGNN (Feng et al., 2019) utilizes the normalized hypergraph Laplacian 13, which expands the hypergraph to a graph via clique expansion. HGNN also follows the two-phase scheme (described in 2.1). Using non-linear activation $\sigma$, the HGNN layer update is:

$$X^{(+1} = \sigma \left( D_v^{-1/2} B_1 D_e^{-1} B_1^\top D_v^{-1/2} X^l W^l \right) \tag{29}$$

where $X^0 = X$.

### B.2.7 PhenomNN

PhenomNN Wang et al. (2023) is defined on a hypergraph $H = (V, X, F)$ with incidence matrix $B_1 \in \{0, 1\}^{|V| \times |F|}$, node features $X \in \mathbb{R}^{|V| \times d}$ and base predictor $f(X; W) \in \mathbb{R}^{|V| \times d}$, and produces embeddings $Y^l \in \mathbb{R}^{|V| \times d}$ via $L$ shared-parameter layers. Writing $D_e \in \mathbb{R}^{|F| \times |F|}$ for the hyperedge degree matrix, define the clique-expansion adjacency

$$A_C = B_1 B_1^\top, \quad D_C = \text{diag}(A_C \mathbf{1}), \quad \bar{A}_S = B_1 D_e^{-1} B_1^\top, \quad \bar{D}_S = \text{diag}(\bar{A}_S \mathbf{1}). \tag{30}$$

At iteration $l$, let

$$\tilde{Y}_C^l = A_C Y^l (H_0 + H_0^\top) - D_C Y^l H_0 H_0^\top, \tag{31}$$

$$\tilde{Y}_S^l = \bar{A}_S Y^l (H_1 + H_1^\top) - \bar{D}_S Y^l H_1 H_1^\top, \tag{32}$$

where $H_0, H_1 \in \mathbb{R}^{d \times d}$ are learned compatibility matrices, and set $\tilde{D} = D_C \lambda_0 + \bar{D}_S \lambda_1 + I$. Then

$$Y^{l+1} = \text{ReLU}\left((1 - \alpha) Y^l + \alpha \tilde{D}^{-1}\left(f(X; W) + \lambda_0 \tilde{Y}_C^l + \lambda_1 (\bar{L}_S Y^l + \tilde{Y}_S^l)\right)\right), \tag{33}$$

with step-size $\alpha$, Laplacian $\bar{L}_S = \bar{D}_S - \bar{A}_S$, and nonlinearity $\text{ReLU}(Z) = \max(0, Z)$. Tying $\{H_0, H_1, W, \lambda_0, \lambda_1, \alpha\}$ across layers yields an unrolled proximal-gradient descent that provably converges to the minimizer of a hypergraph energy while remaining implementable via standard message-passing primitives. We can re-express PhenomNN in the node-wise MP form:

$$y_i^{l+1} = \sigma \left( \sum_{j \in \mathcal{N}_i^C} y_j^l W_{ij}^l + y_i^l W_i^l + \alpha \tilde{D}_{ii}^{-1} f(x_i; W) \right) \tag{34}$$

where $\mathcal{N}_i^C$ are the neighbors in the clique expansion graph, and:

$$W_{ij}^l = \alpha \tilde{D}_{ii}^{-1} \left[ \lambda_0 [A_C]_{i,j} (H_0 + H_0^\top) + \lambda_1 [\bar{A}_S]_{i,j} (H_1 + H_1^\top - I) \right], \tag{35}$$

$$W_i^l = (1 - \alpha)I - \alpha \tilde{D}_{ii}^{-1} \left[ \lambda_0 [D_C]_{i,i} H_0 H_0^\top + \lambda_1 [\bar{D}_S]_{i,i} (H_1 H_1^\top - I) \right]. \tag{36}$$

## C Experimental details

### C.1 Datasets

We consider multiple datasets commonly used for benchmarking in the literature, including social networks, chemical reaction networks, and citation networks.

#### C.1.1 Graph Datasets

Collab, Imdb and Reddit are proposed in (Yanardag & Vishwanathan, 2015). Collab is a collection of ego-networks where nodes are researchers. The labels correspond to the fields of research of the authors. Imdb is also a collection of ego-networks. Nodes are actors and an edge between two nodes is present if the actors played together. The labels correspond to the genre of movies used to construct the networks. Reddit is a collection of graphs corresponding to online discussion threads on reddit. Nodes correspond to users, who are connected if they replied to each other comments. The task consists in determining if the community is a discussion-community or a question answering community.

Mutag is a collection of graphs corresponding to nitroaromatic compounds (Debnath et al., 1991). The goal is to predict their mutagenicity in the Ames test (Ames et al., 1973) using S. typhimurium TA98.

Proteins and Enzymes are introduced in (Borgwardt et al., 2005). These datasets use the 3D structure of the folded proteins to build a graph of amino acids.

Peptides is a chemical data set introduced in (Dwivedi et al., 2022). The graphs are derived from peptides, short chains of amino acid, such that the nodes correspond to the heavy (non-hydrogen) atoms while the edges represent the bonds between them. Peptides-func is a graph classification task, with a total of 10 classes based on the peptide function (Antibacterial, Antiviral, etc). Peptides-struct is a graph regression task.

We outline basic characteristics of these datasets in Tab. 5.

|  | Collab | Imdb | Reddit | Mutag | Enzymes | Proteins | Peptides-func | Peptides-struct |
|---|---|---|---|---|---|---|---|---|
| # graphs | 5000 | 1000 | 2000 | 188 | 600 | 1113 | 15,535 | 15,535 |
| avg. # node per graph | 74.49 | 19.77 | 425.57 | 17.93 | 31.86 | 37.40 | 150.94 | 150.94 |
| # classes | 3 | 2 | 2 | 2 | 6 | 2 | 10 | - |

Table 5: Dataset Statistics for Collab, Imdb, Reddit, Mutag, Enzymes, Proteins and Peptides.

#### C.1.2 Hypergraph Datasets

We use five datasets that are naturally parametrized as hypergraphs: pubmed, Cora co-authorship (Cora-CA), cora co-citation (Cora-CC), Citeseer (Sen et al., 2008) and DBLP (Rossi & Ahmed, 2015). We use the same pre-processed hypergraphs as in Yadati et al. (2019), which are taken from Huang & Yang (2021). The hypergraphs are created with each vertex representing a document. The Cora data set, for example, contains machine learning papers divided into one of seven classes. In a given graph of the co-authorship datasets Cora-CA and DBLP, all documents co-authored by one author form one hyperedge. In pubmed, citeseer and Cora-CC, all documents cited by an author from one hyperedge. We outline basic characteristics of these datasets in Tab. 6.

|  | **Pubmed** | **Cora-CA** | **Cora-CC** | **Citeseer** | **DBLP** |
|---|---|---|---|---|---|
| # hypernodes, $V$ | 19717 | 2708 | 2708 | 3312 | 43413 |
| # hyperedges, $E$ | 7963 | 1072 | 1579 | 1079 | 22535 |
| # features, $d$ | 500 | 1433 | 1433 | 3703 | 1425 |
| # classes, $q$ | 3 | 7 | 7 | 6 | 6 |

Table 6: Dataset statistics.

We use these datasets in semi-supervised hypernode classification tasks, where the objective is to predict labels for test nodes given the hypergraph structure, node features, and a limited number of training labels.

## C.2 Hyperparameters

**For GNNs.**

We outline the hyperparameter used for Tab. 1, Tab. 2, Tab. 10, and Tab. 13 in Tab. 7, Tab. 8, Tab. 9.

| **Features** | **citeseer-CC** | **Cora-CA** | **Cora-CC** | **Pubmed-CC** | **DBLP** |
|---|---|---|---|---|---|
| Num. Layers | 3 | 3 | 3 | 3 | 3 |
| Hidden Dim. | 128 | 128 | 128 | 128 | 128 |
| Learning Rate | 0.001 | 0.001 | 0.001 | 0.001 | 0.001 |
| Dropout | 0.2 | 0.2 | 0.2 | 0.2 | 0.2 |
| Batch Size | 50 | 50 | 50 | 50 | 50 |
| Epochs | 300 | 300 | 300 | 300 | 300 |

Table 7: Hyperparameter settings for Tab. 1 (for GCN).

| **Features** | **Collab** | **Imdb** | **Reddit** | **Mutag** | **Enzymes** | **Proteins** | **Peptides-f** | **Peptides-s** |
|---|---|---|---|---|---|---|---|---|
| Num. Layers | 4 | 4 | 4 | 4 | 4 | 4 | 8 | 8 |
| Hidden Dim. | 64 | 64 | 64 | 64 | 64 | 64 | 235 | 235 |
| Learning Rate | 0.001 | 0.001 | 0.001 | 0.001 | 0.001 | 0.001 | 0.001 | 0.001 |
| Dropout | 0.5 | 0.5 | 0.5 | 0.5 | 0.5 | 0.5 | 0.1 | 0.1 |
| Batch Size | 50 | 50 | 50 | 50 | 50 | 50 | 50 | 50 |
| Epochs | 300 | 300 | 300 | 300 | 300 | 300 | 300 | 300 |

Table 8: Hyperparameter settings used for GCN for Tab. 2 and 10.

| **Features** | **Collab** | **Imdb** | **Reddit** |
|---|---|---|---|
| MP-Layer | GIN | GIN | GIN |
| Num. Layers | 4 | 4 | 4 |
| Hidden Dim. | 64 | 64 | 64 |
| Learning Rate | 0.001 | 0.001 | 0.001 |
| Dropout | 0.2 | 0.2 | 0.2 |
| Batch Size | 50 | 50 | 50 |
| Epochs | 300 | 300 | 300 |

Table 9: Hyperparameter settings for GIN for Tab. 13.

**For HNNs.**

For the hypergraph architectures (Huang & Yang (2021)), we use a weight decay of 0.0005, a patience of 200 and 200 epochs. The learning rate is set to 0.01, the dropout to 0.6. The activation layer between UniConvs is relu. We add self loops to the (hyper-)graphs. The number of convolution heads is 8. The number of hidden layers is 2. The parameters for PhenomNN are presented in Tab. 27 and the parameters for HGNN are presented in Tab. 29

# D  Additional GNN results

We include additional results with graph-level and hypergraph-level encodings with GCN (Tab. 10) and GPS (Tab. 11).

| Model (Encodings) | Enzymes (↑) | Proteins (↑) |
|---|---|---|
| GCN (No Encoding) | $28.03 \pm 1.15$ | $71.48 \pm 0.90$ |
| GCN (LCP-FRC) | $27.66 \pm 1.48$ | $70.89 \pm 1.16$ |
| GCN (LCP-ORC) | $\mathbf{33.17 \pm 1.43}$ | $\mathbf{74.22 \pm 1.77}$ |
| GCN (19-RWPE) | $30.66 \pm 1.78$ | $71.94 \pm 1.58$ |
| GCN (20-LAPE) | $28.52 \pm 1.16$ | $71.46 \pm 1.52$ |
| GCN (HCP-FRC) | $30.87 \pm 1.38$ | $71.27 \pm 1.20$ |
| GCN (HCP-ORC) | $\mathbf{32.83 \pm 1.36}$ | $\mathbf{73.78 \pm 1.25}$ |
| GCN (EE H-19-RWPE) | $\underline{31.74 \pm 1.30}$ | $\underline{73.83 \pm 1.08}$ |
| GCN (EN H-19-RWPE) | $30.93 \pm 1.27$ | $\underline{74.05 \pm 1.13}$ |
| GCN (Hodge H-20-LAPE) | $29.46 \pm 1.14$ | $\underline{72.89 \pm 1.31}$ |
| GCN (Norm. H-20-LAPE) | $29.60 \pm 1.21$ | $\underline{73.12 \pm 1.36}$ |

Table 10: GCN performance with graph level encodings (top) and hypergraph level encodings (bottom). We report mean and standard deviation across 50 runs.

| Model (Encodings) | Collab (↑) | Imdb (↑) |
|---|---|---|
| GPS (No Encoding) | $74.17 \pm 1.33$ | $70.93 \pm 1.21$ |
| GPS (LCP-FRC) | $74.22 \pm 1.27$ | $71.46 \pm 1.77$ |
| GPS (LCP-ORC) | $74.52 \pm 1.18$ | $71.84 \pm 1.26$ |
| GPS (19-RWPE) | $74.29 \pm 1.42$ | $66.40 \pm 1.53$ |
| GPS (20-LAPE) | $74.74 \pm 1.23$ | $70.67 \pm 1.18$ |
| GPS (HCP-FRC) | $73.37 \pm 1.59$ | $71.48 \pm 1.03$ |
| GPS (HCP-ORC) | $74.18 \pm 1.22$ | $72.05 \pm 1.15$ |
| GPS (EE H-19-RWPE) | $\mathbf{76.19 \pm 1.29}$ | $\mathbf{73.19 \pm 1.07}$ |
| GPS (EN H-19-RWPE) | $\mathbf{75.92 \pm 1.33}$ | $\underline{73.08 \pm 1.24}$ |
| GPS (Hodge H-20-LAPE) | $\underline{76.10 \pm 1.16}$ | $\underline{73.15 \pm 1.02}$ |
| GPS (Norm. H-20-LAPE) | $\underline{75.81 \pm 1.21}$ | $72.94 \pm 1.18$ |

Table 11: GPS performance with graph level encodings (top) and hypergraph level encodings (bottom). We report mean and standard deviation across 50 runs.

We include additional results with graph-level and hypergraph-level encodings on the Mutag dataset with GCN and GPS (Tab. 12) and on the social networks Collab, Imdb, and Reddit using GIN (Tab. 13).

| Encodings | GCN (↑) | GPS (↑) |
|---|---|---|
| No Encoding | $65.96 \pm 1.76$ | $80.40 \pm 1.53$ |
| LCP-FRC | $67.04 \pm 1.49$ | $83.94 \pm 2.06$ |
| LCP-ORC | $83.09 \pm 1.71$ | $84.93 \pm 1.82$ |
| 19-RWPE | $71.75 \pm 2.08$ | $80.13 \pm 1.65$ |
| 20-LAPE | $73.30 \pm 1.95$ | $82.27 \pm 1.57$ |
| HCP-FRC | $80.85 \pm 1.77$ | $\mathbf{89.36 \pm 1.68}$ |
| EE H-19-RWPE | $\mathbf{85.32 \pm 1.63}$ | $\underline{88.65 \pm 2.24}$ |
| EN H-19-RWPE | ${\color{blue}82.34 \pm 2.68}$ | $\underline{88.49 \pm 2.12}$ |
| Hodge H-20-LAPE | ${\color{blue}83.66 \pm 1.90}$ | $86.72 \pm 1.96$ |
| Norm. H-20-LAPE | $81.68 \pm 1.79$ | $86.90 \pm 1.81$ |

Table 12: GCN and GPS performance on the Mutag dataset with various graph and hypergraph encodings. We report mean and standard deviation across 50 runs.

| Model (Encodings) | Collab (↑) | Imdb (↑) | Reddit (↑) |
|---|---|---|---|
| GIN (No Encoding) | $67.44 \pm 1.13$ | $67.12 \pm 1.36$ | $75.38 \pm 1.27$ |
| GIN (LCP-FRC) | $\mathbf{71.96 \pm 1.30}$ | $\mathbf{70.18 \pm 1.44}$ | $69.66 \pm 1.62$ |
| GIN (LCP-ORC) | $\mathbf{72.60 \pm 1.28}$ | $\mathbf{70.64 \pm 1.32}$ | $\mathbf{87.19 \pm 1.56}$ |
| GIN (19-RWPE) | $\mathbf{71.76 \pm 1.34}$ | $\mathbf{69.35 \pm 2.24}$ | $74.40 \pm 1.68$ |
| GIN (20-LAPE) | $\underline{71.52 \pm 1.26}$ | $68.16 \pm 2.83$ | $75.84 \pm 1.65$ |
| GIN (HCP-FRC) | $\mathbf{71.44 \pm 1.46}$ | $\mathbf{70.40 \pm 1.52}$ | $70.53 \pm 1.48$ |
| GIN (HCP-ORC) | $\mathbf{72.18 \pm 1.37}$ | $\mathbf{69.92 \pm 1.50}$ | $84.82 \pm 1.62$ |
| GIN (EE H-19-RWPE) | $\mathbf{72.08 \pm 1.40}$ | $\mathbf{70.23 \pm 1.78}$ | $77.87 \pm 1.49$ |
| GIN (EN H-19-RWPE) | $\mathbf{72.32 \pm 1.42}$ | $\mathbf{70.53 \pm 1.80}$ | $77.46 \pm 1.53$ |
| GIN (Hodge H-20-LAPE) | $\mathbf{72.16 \pm 1.39}$ | $\mathbf{69.37 \pm 1.65}$ | $79.94 \pm 1.81$ |
| GIN (Norm. H-20-LAPE) | $\mathbf{71.95 \pm 1.35}$ | $\mathbf{69.48 \pm 1.71}$ | $79.15 \pm 1.54$ |

Table 13: GIN performance with selected graph-level encodings (top) and hypergraph level encodings (bottom). We report mean and standard deviation across 50 runs for the Collab, Imdb, and Reddit datasets. In our tables, the best value is in **bold**, the second best in blue and the third best in green.

# E    Empirical Expressivity Analysis

## E.1    BREC Dataset

The BREC dataset is an expressiveness dataset containing 1-WL-indistinguishable graphs in 4 categories: **B**asic, **R**egular, **E**xtension, and **C**FI graphs (Wang & Zhang, 2024). The 140 pairs of regular graphs are further sub-categorized into simple regular graphs (50 pairs), strongly regular graphs (50 pairs), 4-vertex condition graphs (20 pairs) and distance regular graphs (20 pairs). Note that we remove pairs that include non-connected graphs from the original 400 pairs to arrive at a total of 390 pairs. Graphs in the Basic category (60 pairs, of which we remove 4) are non-regular. Some of the CFI graphs are 4-WL-indistinguishable. We provide a plot of the number of nodes and edges in each categories' graphs in Fig. 4 and Fig. 5.

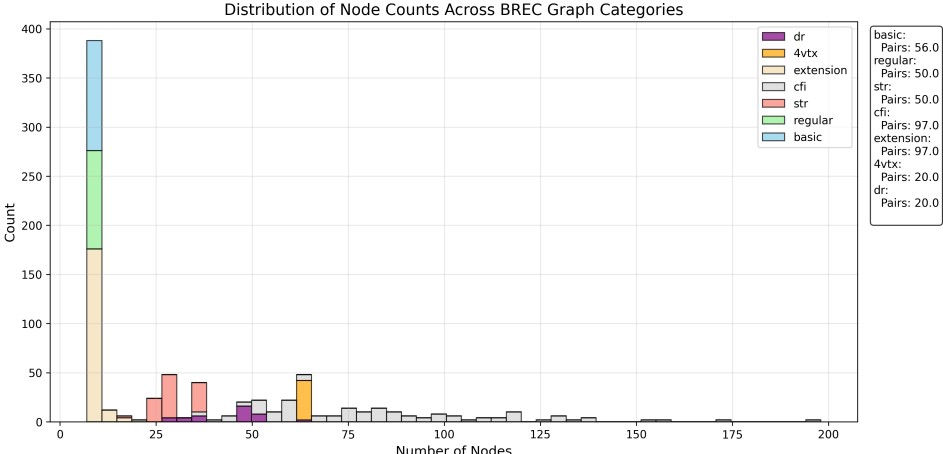

Figure 4: Histogram of the number of nodes in the graphs in BREC. The bars are stacked. We exclude pairs containing non-connected graphs from the original 800 graphs to arrive at 780 graphs. Best seen in color.

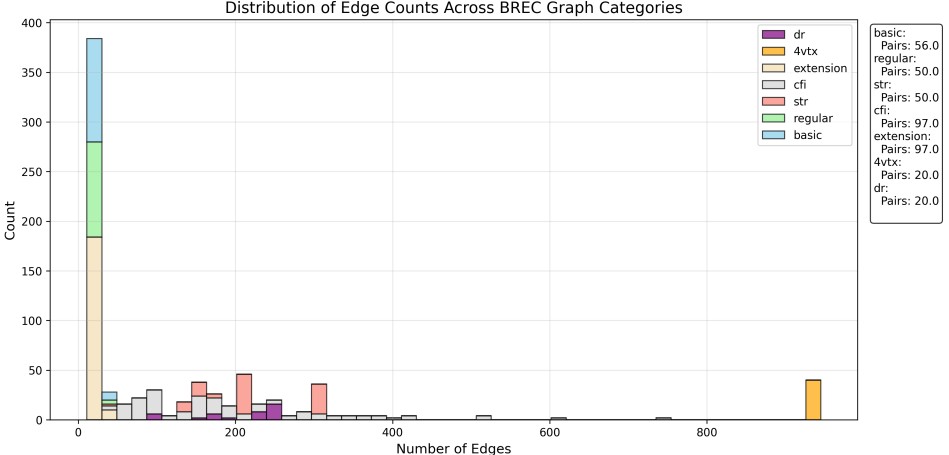

Figure 5: Histogram of the number of edges in the graphs in BREC. The bars are stacked. We exclude pairs containing non-connected graphs from the original 800 graphs to arrive at 780 graphs. Best seen in color.

## E.2 Empirical comparaisons of encodings

We provide a comparison of encodings computed at the graph level and the hypergraph level in Tab. 14. We report the percentage of pairs in BREC that can be distinguished using the encodings, up to row permutation. The results in this table further illustrate theorems A.1, 3.2, 3.6, 3.12 and 3.14. We note that hypergraph-level encodings, with the exception of Hodge H-LAPE, are unable to distinguish pairs in the "Distance Regular" category. The "CFI" category is also notoriously difficult: Some pairs are 4-WL-indistinguishable.

| Level (Encodings) | BASIC | Regular | str | Extension | CFI | 4-Vertex-Condition | Distance Regular |
|---|---|---|---|---|---|---|---|
| Graph: 1-WL/GIN | 0 | 0 | 0 | 0 | 0 | 0 | 0 |
| Graph (LDP) | 0 | 0 | 0 | 0 | 0 | 0 | 0 |
| Hyperaph (LDP) | 91.07 | 94.0 | 100 | 25.77 | 0 | 100 | 0 |
| Graph (LCP-FRC) | 0 | 0 | 0 | 0 | 0 | 0 | 0 |
| Hypergaph (HCP-FRC) | 91.07 | 96.0 | 100 | 26.8 | 0 | 100 | 0 |
| Graph (LCP-ORC) | 100 | 100 | 100 | 100 | 55.67 | 100 | 0 |
| Hypergaph (HCP-ORC) | 100 | 100 | 100 | 94.85 | 100 | 100 | 0 |
| Graph (EE 2-RWPE) | 0 | 0 | 0 | 0 | 0 | 0 | 0 |
| Hypergraph (EE H-2-RWPE) | 91.07 | 82 | 98 | 50.52 | 0 | 100 | 0 |
| Graph (EE 3-RWPE) | 85.71 | 92 | 0 | 6.19 | 0 | 0 | 0 |
| Hypergraph (EE H-3-RWPE) | 98.21 | 98 | 98 | 59.79 | 0 | 100 | 0 |
| Graph (EE 4-RWPE) | 100 | 96 | 0 | 83.51 | 0 | 0 | 0 |
| Hypergraph (EE H-4-RWPE) | 100 | 100 | 98 | 92.78 | 0 | 100 | 0 |
| Graph (EE 5-RWPE) | 100 | 100 | 0 | 95.88 | 0 | 0 | 0 |
| Hypergraph (EE H-5-RWPE) | 100 | 100 | 98 | 95.88 | 0 | 100 | 0 |
| Graph (EE 20-RWPE) | 100 | 100 | 0 | 100 | 3.09 | 0 | 0 |
| Hypergraph (EE H-20-RWPE) | 100 | 100 | 98 | 100 | 3.09 | 100 | 0 |
| Graph (Normalized 1-LAPE) | 0.0 | 0.0 | 0 | 0 | 0 | 0 | 0 |
| Hypergraph (Normalized 1-LAPE) | 91.07 | 90 | 96 | 25.77 | 0 | 100 | 0 |
| Graph (Hodge 1-LAPE) | 48.21 | 100 | 100 | 71.13 | 7.22 | 100 | 5 |
| Hypergraph (Hodge 1-LAPE) | 98.21 | 98 | 100 | 74.23 | 7.22 | 100 | 10 |

Table 14: Difference in encodings on the BREC dataset (390 pairs). We report the percentage of pairs with different encoding up to row permutation, at different level (graph or hypergraph). For the ORC Computations, we use the code from (Coupette et al., 2022) applied to hypergraphs and graphs.

## F  Detailed Comparaison of encodings

### F.1  Rook and Shrikhande graphs

The Rook and Shrikhande graphs are examples of strongly regular graphs with parameters srg(16,6,2,2), meaning that they have 16 nodes, all of degree 6, and that any ajacent vertices share 2 common neighbors, while any non-adjacent vertices also share 2 common neighbors. We illustrate these graphs in 6.

We first compute 2-RWPE. The first entry is $0$, as a random walk from node $i$ with 1-hop does not return to node $i$. The second entry is:

$$\frac{1}{6} \sum_{j \in \mathcal{N}_i} \left( \frac{1}{6} \right) = \frac{1}{6} \; .$$

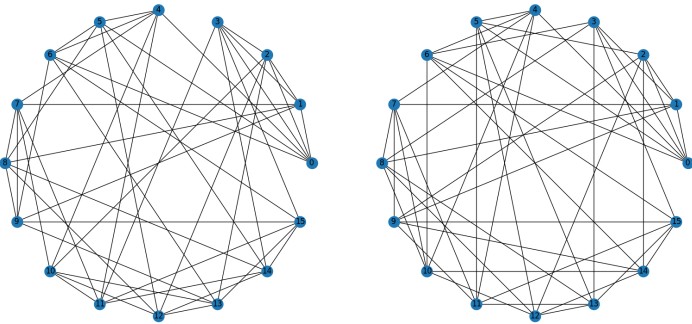

Figure 6: The Rook graph and the Shrikhande graph. Those two non-isomorphic graphs can be hard to distinguish: they are both srg(16,6,2,2) and are isospectral.

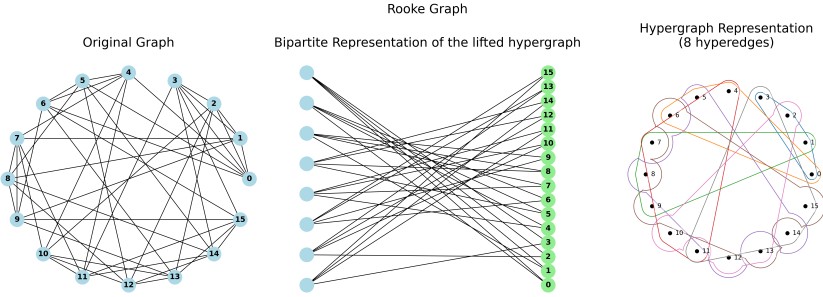

Figure 7: The Rook graph (left), its lifting (right) and its lifting's bipartite representation (center).

For the Rook graph's lifting to a hypergraph (which we shall call the Rook hypergraph), the edges and vertices degree matrices are $D_e = 4I_8$ and $D_v = 2I_{16}$: every hyperedge has 4 nodes, and every node is in two hyperedge (see 7). For the Shrikhande graph's lifting to a hypergraph (which we call the Shrikhande hypergraph), these matrices are $D_e = 3I_8$ and $D_v = 6I_16$: every hyperedge has 3 nodes, and every node is in 6 hyperedge. Using Def. 3.9, we see that for the Rook graph, the FRC of any edge is $4(2-2) = 0$, while for Shrikhande graph, the FRC of any edge is $3(2-6) = -12$.

## F.2 Pair 0 of the "Basic" Category of BREC

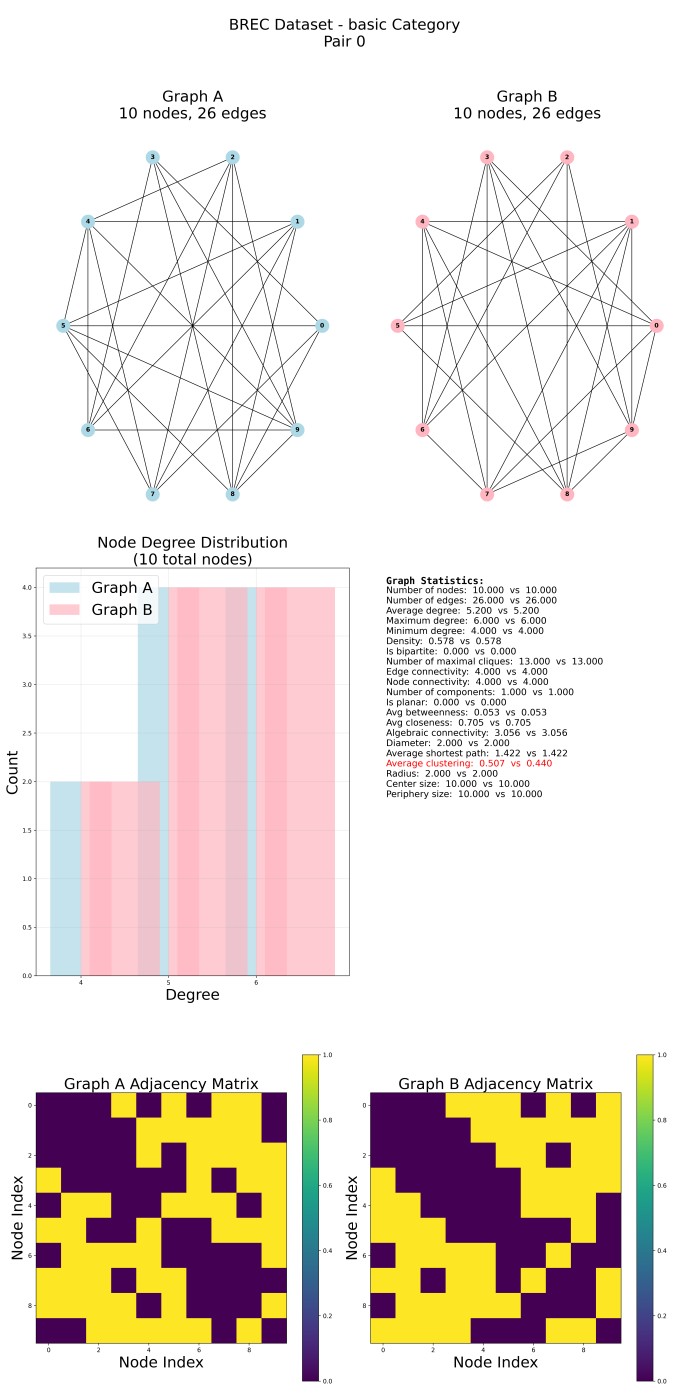

Figure 8: The pair 0 of the "Basic" category in BREC. Top: the two graphs in the pair. Second row: the (node) degree distributions and some statistics. Bottom: the adjacency matrices of the graphs.

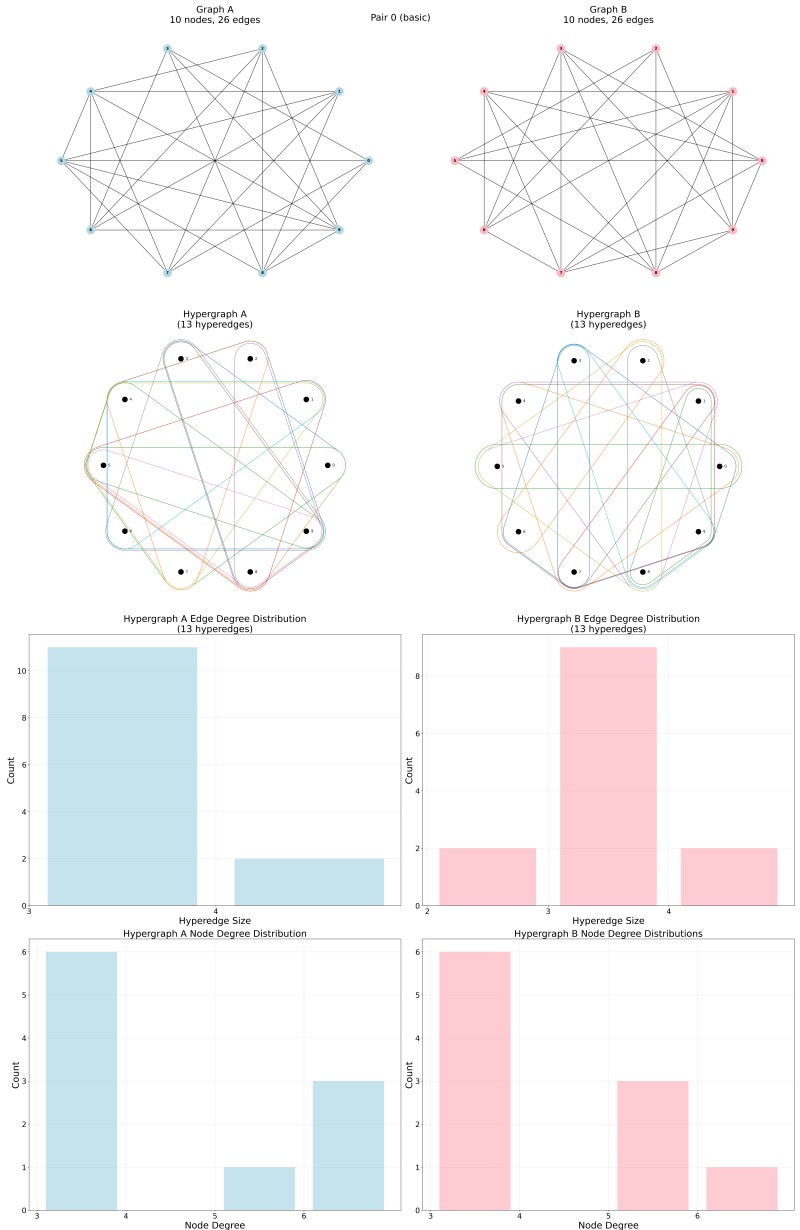

Figure 9: The pair 0 of the "Basic" category in BREC. Top: the two graphs in the pair. Second row: the graphs' liftings. Third row: the sizes of the (hyper)edges. Bottom: the node degrees.

We compute various encodings on this pair.

### F.2.1 RWPE

The 1st entry of the RWPE encoding is 0. We now compute one of the 2nd entries at the graph level. Start with node 0, a node of degree 4, on pair A (see 8). A random-walker can go to nodes of degree 4 (the node 3), 5 (the node 7) or 6 (the nodes 5 and 8). Thus, the probability of coming back to node 0 after 2 hops is $\frac{1}{4} \times \frac{1}{4} + \frac{1}{4} \times \frac{1}{5} + \frac{2}{4} \times \frac{1}{6} = 0.1958\overline{3}$.

We report the full encodings for 2-RWPE in 15. We can actually see they are the same (up to row permutation.

At the hypergraph level, H-2-RWPE are different because the maximum absolute value of the last (second) column of the encoding for hypergraph A is 0.2503052503052503 while it is

| 0.0 | $0.1958\overline{3}$ | 0.0 | $0.19\overline{6}$ |
|---|---|---|---|
| 0.0 | $0.191\overline{6}$ | 0.0 | $0.19\overline{6}$ |
| 0.0 | 0.20555556 | 0.0 | $0.191\overline{6}$ |
| 0.0 | 0.18 | 0.0 | $0.191\overline{6}$ |
| 0.0 | $0.19\overline{6}$ | 0.0 | $0.1958\overline{3}$ |
| 0.0 | 0.18 | 0.0 | 0.18 |
| 0.0 | $0.19\overline{6}$ | 0.0 | $0.1\overline{8}$ |
| 0.0 | $0.1958\overline{3}$ | 0.0 | 0.18 |
| 0.0 | $0.1\overline{8}$ | 0.0 | $0.20\overline{5}$ |
| 0.0 | $0.191\overline{6}$ | 0.0 | $0.1958\overline{3}$ |

Table 15: Pair A (left) and Pair B (right) 2-RWPE encodings. They match if we reorder the rows of pair A as follow: (4, 6, 1, 9, 0, 3, 8, 5, 2, 7).

0.2935064935064935 for graph B. The full encodings can be found in 16. It is straightforward to check that the two encodings cannot be made the same even up to scaling and row permutation.

| 0.0 | 0.15842491 | 0.0 | 0.15165945 |
|---|---|---|---|
| 0.0 | 0.24619611 | 0.0 | 0.15818182 |
| 0.0 | 0.15620094 | 0.0 | 0.21682409 |
| 0.0 | 0.14429618 | 0.0 | 0.15909091 |
| 0.0 | 0.24619611 | 0.0 | 0.15909091 |
| 0.0 | 0.15842491 | 0.0 | 0.29350649 |
| 0.0 | 0.25030525 | 0.0 | 0.26695527 |
| 0.0 | 0.24175824 | 0.0 | 0.21682409 |
| 0.0 | 0.14429618 | 0.0 | 0.15165945 |
| 0.0 | 0.15620094 | 0.0 | 0.15818182 |

Table 16: Pair A (left) and Pair B (right) H-2-RWPE encodings.

### F.2.2   FRC

We now turn our attention to the FRC-LCP and FRC-HCP. The FRC-LCP of both pairs is presented in 17. The encoding match with the following ordering for pair A: (6, 5, 0, 1, 2, 3, 7, 4, 8, 9).

| -6.00 | -4.00 | -5.25 | -5.50 | 0.8291562 | -7.00 | -5.00 | -6.20 | -6.00 | 0.74833148 |
|---|---|---|---|---|---|---|---|---|---|
| -7.00 | -6.00 | -6.60 | -7.00 | 0.48989795 | -8.00 | -6.00 | $-7.3\overline{3}$ | -7.50 | 0.74535599 |
| -7.00 | -6.00 | -6.60 | -7.00 | 0.48989795 | -6.00 | -4.00 | -5.25 | -5.50 | 0.8291562 |
| -6.00 | -4.00 | -5.25 | -5.50 | 0.8291562 | -7.00 | -6.00 | -6.60 | -7.00 | 0.48989795 |
| -8.00 | -7.00 | $-7.3\overline{3}$ | -7.00 | 0.47140452 | -7.00 | -6.00 | -6.60 | -7.00 | 0.48989795 |
| -8.00 | -6.00 | $-7.3\overline{3}$ | -7.50 | 0.74535599 | -6.00 | -4.00 | -5.25 | -5.50 | 0.8291562 |
| -7.00 | -5.00 | -6.20 | -6.00 | 0.74833148 | -7.00 | -5.00 | -6.20 | -6.00 | 0.74833148 |
| -7.00 | -5.00 | -6.20 | -6.00 | 0.74833148 | -8.00 | -7.00 | $-7.3\overline{3}$ | -7.00 | 0.47140452 |
| -8.00 | -6.00 | -7.00 | -7.00 | 0.81649658 | -8.00 | -6.00 | -7.00 | -7.00 | 0.81649658 |
| -8.00 | -6.00 | $-7.3\overline{3}$ | -7.50 | 0.74535599 | -8.00 | -6.00 | $-7.3\overline{3}$ | -7.50 | 0.74535599 |

Table 17: Pair A (left) and Pair B (right) FRC-LCP encodings. They match with the following permuation: (6, 5, 0, 1, 2, 3, 7, 4, 8, 9)

At the hypergraph level, they are different because the maximum absolute value of encoding graph A is 12.0, the maximum absolute value of encoding graph B is 10.0. The full encodings are presented in 18. It is straightforward to check that the matrices cannot be scaled and row permuted to match.

| -9 | -6 | -7 | -6 | 1.41421356 | -8 | -2 | -5 | -5 | 2.44948974 |
|---|---|---|---|---|---|---|---|---|---|
| -9 | -5 | $-7.\bar{6}$ | -9 | 1.88561808 | -10 | -8 | -8.6 | -8 | 0.8 |
| -9 | -5 | $-7.\bar{6}$ | -9 | 1.88561808 | -8 | -2 | $-5.\bar{3}$ | -6 | 2.49443826 |
| -9 | -6 | -7 | -6 | 1.41421356 | -8 | -5 | -7 | -8 | 1.41421356 |
| -11 | -5 | -7.8 | -9 | 2.4 | -8 | -5 | -7 | -8 | 1.41421356 |
| -12 | -6 | $-9.\bar{3}$ | -9 | 1.88561808 | -8 | -2 | $-5.\bar{3}$ | -6 | 2.49443826 |
| -9 | -5 | $-6.\bar{6}$ | -6 | 1.69967317 | -8 | -2 | -5 | -5 | 2.44948974 |
| -9 | -5 | $-6.\bar{6}$ | -6 | 1.69967317 | -9 | -5 | -7 | -8 | 1.67332005 |
| -12 | -6 | -9 | -9 | 1.73205081 | -10 | -6 | -8 | -8 | 1.15470054 |
| -12 | -6 | $-9.\bar{3}$ | -9 | 1.88561808 | -10 | -8 | -8.6 | -8 | 0.8 |

Table 18: Pair A (left) and Pair B (right) FRC-HCP encodings.

### F.2.3 1-LAPE

Using the Normalized Laplacian, the pair is 1-LAPE indistinguishable (up to row permuation, and sign flip, as the eingenvectors are defined up to $\pm 1$). The 1-LAPE encodings are presented in 19.

| | |
|---|---|
| 0.30348849 | 0.30348849 |
| 0.3441236 | 0.3441236 |
| 0.3441236 | 0.30348849 |
| 0.30348849 | 0.3441236 |
| 0.28097574 | 0.28097574 |
| 0.28097574 | 0.28097574 |
| 0.30348849 | 0.30348849 |
| 0.30348849 | 0.30348849 |
| 0.3441236 | 0.3441236 |
| 0.3441236 | 0.3441236 |

Table 19: Pair A (left) and Pair B (right) Normalized 1-LAPE encodings. They match with the following ordering for pair A: (0, 1, 3, 2, 4, 5, 6, 7, 8, 9).

At the hypergraph level, H-1-LAPE are different because the maximum absolute value is 0.408248290463863 for pair A and 0.39223227027636787 for pair B. The H-1-LAPE can be found in 20.

| | |
|---|---|
| 0.25 | 0.2773501 |
| 0.40824829 | 0.39223227 |
| 0.40824829 | 0.2773501 |
| 0.25 | 0.39223227 |
| 0.40824829 | 0.39223227 |
| 0.40824829 | 0.39223227 |
| 0.25 | 0.2773501 |
| 0.25 | 0.2773501 |
| 0.20412415 | 0.19611614 |
| 0.20412415 | 0.19611614 |

Table 20: Pair A (left) and Pair B (right) Normalized H-1-LAPE encodings.

## F.3 Additional Plots

We present the pairs 0 of the regular category, alongside with the graphs' liftings, in Fig. 10. We present the pairs 0 of the strongly regular category in Fig. 11.

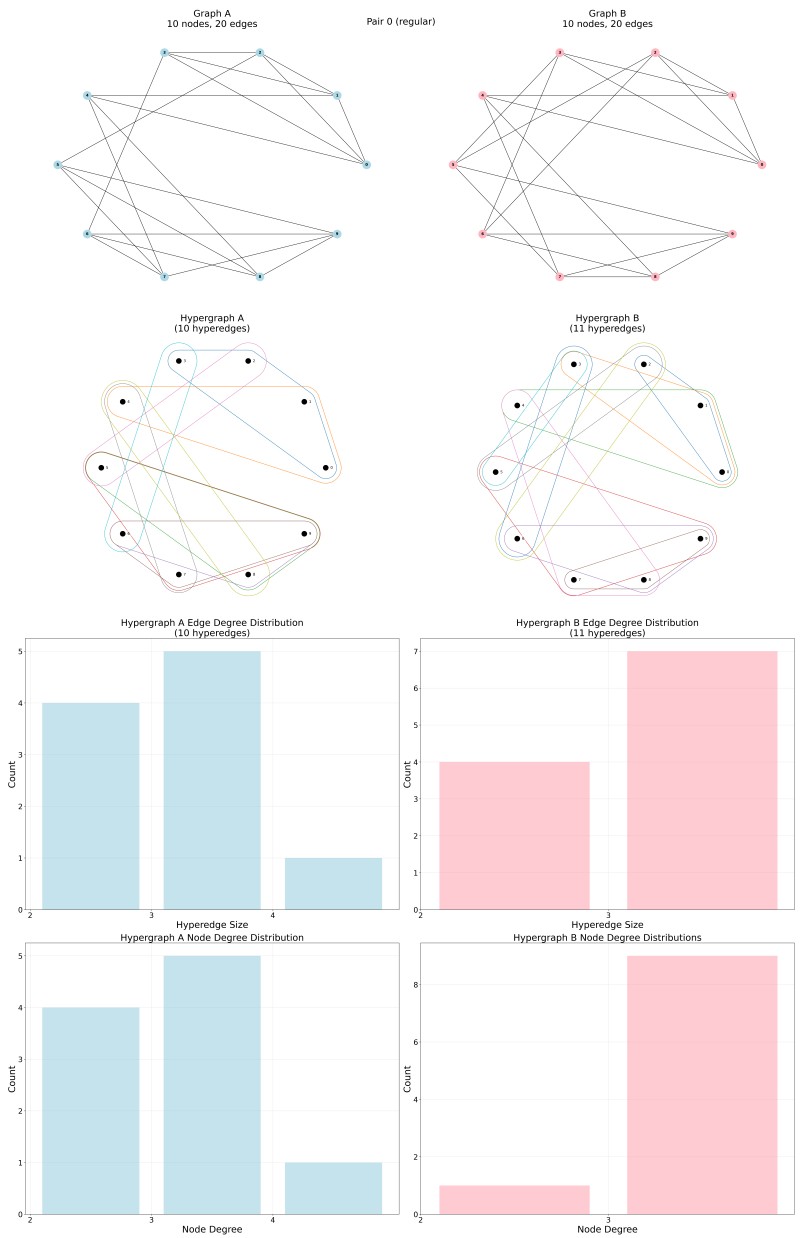

Figure 10: The pair 0 of the regular category in BREC. Top: the two graphs in the pair. Second row: the graphs' liftings. Third row: the sizes of the (hyper)edges. Bottom: the node degrees.

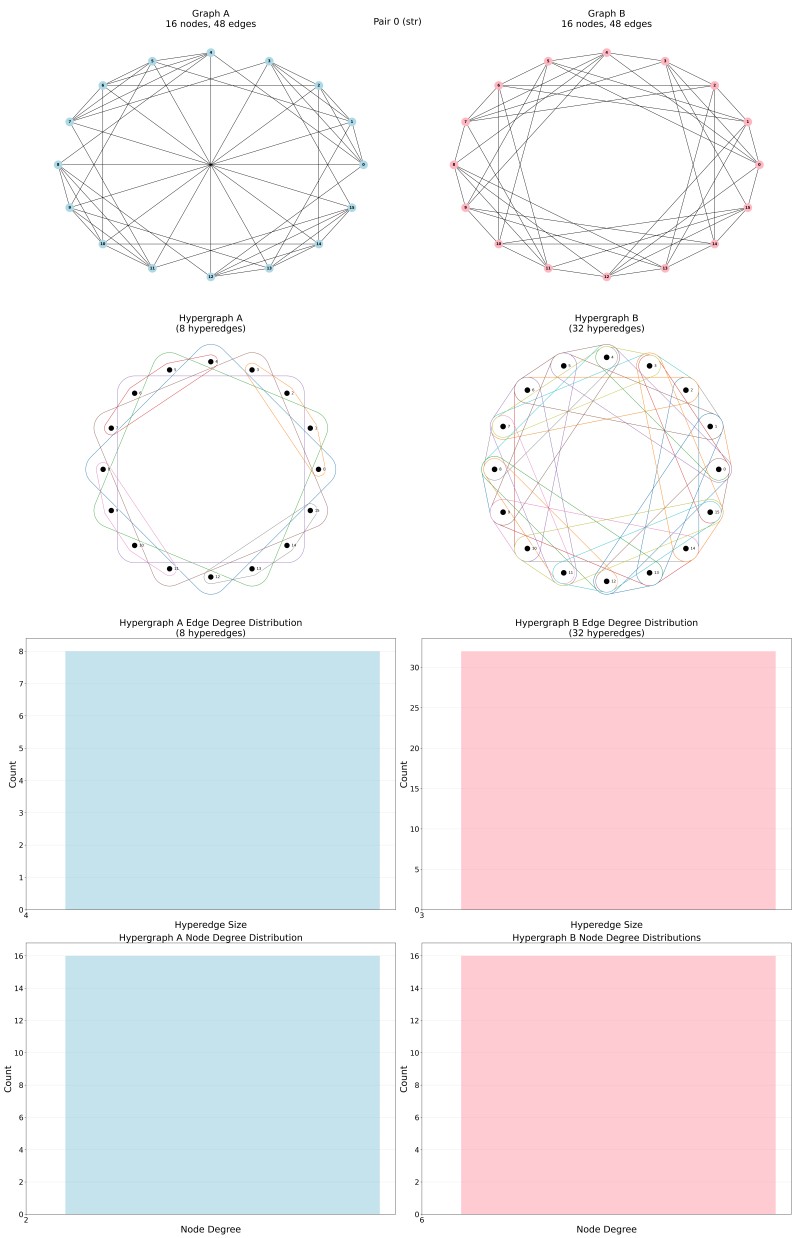

Figure 11: The pair 0 of the strongly regular category in BREC. Top: the two graphs in the pair. Second row: the graphs' liftings. Third row: the sizes of the (hyper)edges. Bottom: the node degrees.

# G   Detailed ablation study for HNNs

## G.1   UniGNN

We run node-level classification tasks using HNNs (Huang & Yang, 2021). We present results using UniGAT in Tab. 22, UniGCN in Tab. 23, UniGIN in Tab. 24, UniSAGE in Tab. 25 and UniGCNII in Tab. 26. For these experiments, we repeat experiments over 10 data splits (Yadati et al., 2019) with 8 different random seeds (80 total experiments). We train for 200 epochs and report the mean and std of the testing accuracies across 80 runs. We use the Adam optimizer with a learning rate of $0.01$ and a weight decay of $0.0005$. We use the relu activation function. The patience is set to 200 epochs, and the dropout probability for the input layer is 0.6. The number of hidden features is set to 8, and the number of layers is 2.

For this semi-supervised hypernode classification task, each dataset is split so that a small fraction of labeled nodes is used for training (with label rates ranging from 0.8% to 5.2%, depending on the dataset - see Tab. 21), and the rest are used for testing and validation.

| Dataset | Total Samples | Train | Test/Val | Classes |
|---------|--------------|-------|----------|---------|
| Cora-CA | 2,708 | 140 (5.2%) | 2,568 (94.8%) | 7 |
| Citeseer | 3,312 | 138 (4.2%) | 3,174 (95.8%) | 6 |
| Cora-CC | 2,708 | 140 (5.2%) | 2,568 (94.8%) | 7 |
| Pubmed | 19,717 | 78 (0.4%) | 19,639 (99.6%) | 3 |

Table 21: Train/Test split proportions in the hypergraph datasets.

| Model | citeseer-CC (↑) | cora-CA (↑) | cora-CC (↑) | pubmed-CC (↑) |
|-------|-----------------|-------------|-------------|---------------|
| UniGAT (HCP-FRC) | 61.08 ± 1.85 | **74.85 ± 1.66** | 65.95 ± 3.24 | 66.34 ± 1.79 |
| UniGAT (LDP) | **62.07 ± 1.68** | **75.47 ± 1.47** | 69.31 ± 2.23 | 68.41 ± 1.79 |
| UniGAT (Hodge H-20-LAPE) | **63.21 ± 1.53** | **75.80 ± 1.23** | **71.22 ± 1.60** | **75.77 ± 1.05** |
| UniGAT (Norm. H-20-LAPE) | **63.15 ± 1.63** | **75.65 ± 1.50** | **71.23 ± 1.87** | **75.77 ± 1.02** |
| UniGAT (H-19-RWPEE EE) | **62.97 ± 1.51** | **75.65 ± 1.40** | **70.78 ± 1.85** | **74.78 ± 1.18** |
| UniGAT (H-19-RWPEE EN) | **62.88 ± 1.53** | **75.76 ± 1.37** | **70.74 ± 1.86** | **74.75 ± 1.18** |
| UniGAT (H-19-RWPEE WE) | **62.97 ± 1.45** | **75.53 ± 1.53** | **70.86 ± 1.93** | **74.82 ± 1.12** |
| UniGAT (no encodings) | **63.25 ± 1.48** | **75.68 ± 1.45** | **71.16 ± 1.55** | **75.62 ± 1.09** |

Table 22: Node level classification for hypergraph using hypegraph encodings for UniGAT. The depth is 2.

| Model | citeseer-CC (↑) | cora-CA (↑) | cora-CC (↑) | pubmed-CC (↑) |
|-------|-----------------|-------------|-------------|---------------|
| UniGCN (HCP-FRC) | 61.20 ± 1.83 | **74.64 ± 1.45** | 68.98 ± 1.59 | 67.37 ± 1.73 |
| UniGCN (LDP) | 61.67 ± 1.90 | **75.17 ± 1.54** | 69.17 ± 1.58 | 69.33 ± 1.57 |
| UniGCN (Hodge H-20-LAPE) | **63.46 ± 1.58** | **75.64 ± 1.37** | **71.31 ± 1.19** | **75.37 ± 1.01** |
| UniGCN (Norm. H-20-LAPE) | **63.41 ± 1.61** | **75.55 ± 1.48** | **71.20 ± 1.24** | **75.30 ± 1.01** |
| UniGCN (H-19-RWPEE EE) | **63.29 ± 1.52** | **75.34 ± 1.28** | **71.13 ± 1.24** | **74.61 ± 1.18** |
| UniGCN (H-19-RWPEE EN) | **63.09 ± 1.62** | **75.30 ± 1.37** | **71.21 ± 1.34** | **74.61 ± 1.09** |
| UniGCN (H-19-RWPEE WE) | **63.04 ± 1.74** | **75.53 ± 1.43** | **71.40 ± 1.25** | **74.59 ± 1.11** |
| UniGCN (no encodings) | **63.36 ± 1.76** | **75.72 ± 1.16** | **71.10 ± 1.37** | **75.32 ± 1.09** |

Table 23: Node level classification for hypergraph using hypegraph encodings for UniGCN. The depth is 2.

| Model | citeseer-CC (↑) | cora-CA (↑) | cora-CC (↑) | pubmed-CC (↑) |
|-------|-----------------|-------------|-------------|---------------|
| UniGIN (HCP-FRC) | **59.10 ± 1.84** | **72.91 ± 1.88** | 57.77 ± 3.10 | 65.55 ± 2.40 |
| UniGIN (LDP) | **59.88 ± 2.37** | **73.83 ± 1.59** | 62.96 ± 2.89 | 67.41 ± 3.16 |
| UniGIN (Hodge H-20-LAPE) | **60.67 ± 2.23** | **74.29 ± 1.62** | **67.67 ± 2.50** | **75.11 ± 1.47** |
| UniGIN (Norm. H-20-LAPE) | **60.18 ± 2.37** | **74.12 ± 1.46** | **67.92 ± 2.23** | **75.09 ± 1.45** |
| UniGIN (H-19-RWPEE EE) | **60.33 ± 2.04** | **74.03 ± 1.51** | **67.70 ± 2.15** | **74.44 ± 1.44** |
| UniGIN (H-19-RWPEE EN) | **60.13 ± 2.31** | **74.04 ± 1.58** | **67.62 ± 2.46** | **74.34 ± 1.37** |
| UniGIN (H-19-RWPEE WE) | **60.23 ± 2.19** | **73.97 ± 1.61** | **67.50 ± 2.46** | **74.44 ± 1.32** |
| UniGIN (no encodings) | **60.56 ± 2.31** | **73.97 ± 1.56** | **67.70 ± 2.33** | **75.02 ± 1.39** |

Table 24: Node level classification for hypergraph using hypegraph encodings for UniGIN. The depth is 2.

| Model | citeseer-CC (↑) | cora-CA (↑) | cora-CC (↑) | pubmed-CC (↑) |
|---|---|---|---|---|
| UniSAGE (HCP-FRC) | **59.10 ± 2.29** | 72.57 ± 1.96 | 57.35 ± 3.15 | 65.71 ± 2.58 |
| UniSAGE (LDP) | **59.97 ± 2.27** | **73.88 ± 1.71** | 63.08 ± 2.68 | 67.53 ± 3.09 |
| UniSAGE (Hodge H-20-LAPE) | **60.55 ± 2.02** | **74.13 ± 1.57** | **67.80 ± 2.27** | **75.03 ± 1.42** |
| UniSAGE (Norm. H-20-LAPE) | **60.54 ± 2.19** | **74.10 ± 1.52** | 67.89 ± 2.37 | **75.07 ± 1.44** |
| UniSAGE (H-19-RWPEE EE) | **60.29 ± 2.17** | **73.99 ± 1.59** | 67.76 ± 1.91 | **74.41 ± 1.43** |
| UniSAGE (H-19-RWPEE EN) | **60.30 ± 2.33** | **74.00 ± 1.57** | 67.86 ± 2.15 | **74.29 ± 1.36** |
| UniSAGE (H-19-RWPEE WE) | **60.22 ± 2.22** | **73.97 ± 1.35** | 67.82 ± 2.18 | **74.37 ± 1.33** |
| UniSAGE (no encodings) | **60.56 ± 2.10** | **74.16 ± 1.50** | **67.80 ± 2.33** | **75.02 ± 1.44** |

Table 25: Node level classification for hypergraph using hypegraph encodings for UniSAGE. The depth is 2.

| Model | citeseer-CC (↑) | cora-CA (↑) | cora-CC (↑) | pubmed-CC (↑) |
|---|---|---|---|---|
| UniGCNII (HCP-FRC depth 2) | 61.19 ± 1.65 | 75.50 ± 1.41 | 66.83 ± 1.88 | 65.00 ± 2.18 |
| UniGCNII (LDP depth 2) | 62.34 ± 1.62 | **76.39 ± 1.58** | 68.65 ± 1.59 | 67.40 ± 1.93 |
| UniGCNII (Hodge H-20-LAPE depth 2) | 63.90 ± 1.87 | **76.68 ± 1.44** | 71.09 ± 1.20 | **75.51 ± 1.13** |
| UniGCNII (Norm. H-20-LAPE depth 2) | **64.09 ± 1.80** | **76.79 ± 1.31** | 71.06 ± 1.28 | **75.44 ± 1.09** |
| UniGCNII (H-19-RWPEE EE depth 2) | 63.72 ± 1.55 | **76.59 ± 1.39** | 70.64 ± 1.28 | **75.05 ± 0.99** |
| UniGCNII (H-19-RWPEE EN depth 2) | 63.67 ± 1.47 | **76.56 ± 1.48** | 70.87 ± 1.31 | **75.01 ± 0.98** |
| UniGCNII (H-19-RWPEE WE depth 2) | 63.71 ± 1.53 | **76.74 ± 1.36** | 70.68 ± 1.30 | **75.03 ± 0.96** |
| UniGCNII (no encodings depth 2) | **64.13 ± 1.68** | **76.70 ± 1.43** | 70.68 ± 1.53 | **75.40 ± 1.18** |
| UniGCNII (HCP-FRC depth 8) | 62.05 ± 1.47 | **75.97 ± 1.37** | 66.45 ± 1.88 | 64.27 ± 2.66 |
| UniGCNII (LDP depth 8) | 62.90 ± 1.40 | **76.98 ± 1.28** | 69.06 ± 1.67 | 66.78 ± 2.23 |
| UniGCNII (Hodge H-20-LAPE depth 8) | **65.18 ± 1.41** | **77.06 ± 1.22** | **71.93 ± 1.15** | **75.29 ± 1.33** |
| UniGCNII (Norm. H-20-LAPE depth 8) | **65.01 ± 1.60** | **77.00 ± 1.37** | **71.91 ± 1.26** | **75.30 ± 1.35** |
| UniGCNII (H-19-RWPEE EE depth 8) | **64.77 ± 1.49** | **76.95 ± 1.31** | **71.57 ± 1.22** | **74.59 ± 1.40** |
| UniGCNII (H-19-RWPEE EN depth 8) | **64.66 ± 1.43** | **76.92 ± 1.20** | **71.79 ± 1.14** | **74.61 ± 1.35** |
| UniGCNII (H-19-RWPEE WE depth 8) | **64.75 ± 1.46** | **76.85 ± 1.23** | **71.60 ± 1.28** | **74.60 ± 1.37** |
| UniGCNII (no encodings depth 8) | **64.72 ± 1.58** | **77.17 ± 1.34** | **71.57 ± 1.32** | **75.24 ± 1.30** |
| UniGCNII (HCP-FRC depth 64) | 62.93 ± 1.45 | 75.01 ± 1.40 | 65.54 ± 1.93 | 64.44 ± 2.82 |
| UniGCNII (LDP depth 64) | 63.60 ± 1.61 | **75.89 ± 1.41** | 69.85 ± 1.65 | 66.80 ± 2.07 |
| UniGCNII (Hodge H-20-LAPE depth 64) | **65.38 ± 1.53** | **76.35 ± 1.08** | **72.52 ± 1.38** | **75.36 ± 1.29** |
| UniGCNII (Norm. H-20-LAPE depth 64) | **65.25 ± 1.60** | **76.24 ± 1.16** | **72.68 ± 1.36** | **75.36 ± 1.29** |
| UniGCNII (H-19-RWPEE EE depth 64) | **65.40 ± 1.49** | **76.25 ± 1.17** | **72.62 ± 1.24** | **74.54 ± 1.43** |
| UniGCNII (H-19-RWPEE EN depth 64) | **65.38 ± 1.56** | **76.31 ± 1.15** | **72.59 ± 1.25** | **74.63 ± 1.36** |
| UniGCNII (H-19-RWPEE WE depth 64) | **65.20 ± 1.52** | **76.16 ± 1.20** | **72.65 ± 1.27** | **74.60 ± 1.33** |
| UniGCNII (no encodings depth 64) | **65.24 ± 1.67** | **76.34 ± 1.12** | **72.64 ± 1.06** | **75.34 ± 1.24** |

Table 26: Node level classification for hypergraph using hypegraph encodings for UniGCNII (depth: 2, 8, 64).

### G.2 PhenomNN

We conduct experiments using PhenomNN with the hyperparameters from Wang et al. (2023). The model employs a hidden dimension of $64$ units with a single hidden layer architecture. For the PhenomNN-specific parameters, we set $\lambda_0 = 20$ and $\lambda_1 = 80$ to control the weighting between different adjacency matrices, $\alpha = 0.1$ for residual connections, and 16 propagation steps. Training is performed with a learning rate of $0.01$, high dropout rate of $0.7$ for regularization, and up to 1000 epochs with early stopping patience of 100. L2 regularization is applied with weight decay of $5 \times 10^{-4}$. The normalization type is set to "full". The full set of parameters is presented in Tab. 27. The results are presented in Tab. 28.

| Parameter Category | Parameter | Value |
|---|---|---|
| **Core Model** | Hidden dimension | 64 |
| | Number of hidden layers | 1 |
| | Base block layers | 1 |
| | Batch normalization | False |
| **PhenomNN-Specific** | $\lambda_0$ | 20 |
| | $\lambda_1$ | 80 |
| | $\alpha$ (residual) | 0.1 |
| | Propagation steps | 16 |
| | Normalization type | Full |
| **Training** | Learning rate | 0.01 |
| | Dropout rate | 0.7 |
| | Epochs | 1000 |
| | Weight decay | $5 \times 10^{-4}$ |
| | Early stopping patience | 100 |
| **Additional** | K (APPNP) | 10 |
| | $\alpha$ (GCNII) | 0.1 |
| | $\lambda$ (GCNII) | 0.5 |
| | $\lambda_4$ | 0 |

Table 27: Hyperparameters used for the PhenomNN experiments.

| Model | citeseer-CC ($\uparrow$) | cora-CA ($\uparrow$) | cora-CC ($\uparrow$) | pubmed-CC ($\uparrow$) |
|---|---|---|---|---|
| PhenomNN (HCP-FRC) | 58.87 ± 1.03 | 74.03 ± 0.97 | 54.02 ± 2.06 | 68.88 ± 01.44 |
| PhenomNN (HCP-ORC) | **64.89 ± 0.45** | 75.93 ± 0.41 | **72.21 ± 0.31** | 77.37 ± 0.23 |
| PhenomNN (LDP) | 62.54 ± 0.52 | 75.67 ± 0.50 | 62.28 ± 3.33 | 70.63 ± 0.87 |
| PhenomNN (Hodge H-20-LAPE) | **64.98 ± 0.58** | **76.76 ± 0.71** | **72.18 ± 0.33** | **78.36 ± 0.23** |
| PhenomNN (Norm. H-20-LAPE) | **64.75 ± 0.52** | **76.64 ± 0.49** | **72.06 ± 0.68** | **78.27 ± 0.25** |
| PhenomNN (H-20-RWPEE EE) | 64.23 ± 0.40 | 76.24 ± 0.57 | **72.18 ± 0.52** | 77.97 ± 0.29 |
| PhenomNN (H-20-RWPEE EN) | **64.47 ± 0.62** | 76.39 ± 0.57 | **71.57 ± 0.73** | 78.10 ± 0.20 |
| PhenomNN (H-20-RWPEE WE) | **64.52 ± 0.69** | 76.54 ± 0.57 | **72.30 ± 0.79** | 78.06 ± 0.24 |
| PhenomNN (no encodings) | **64.96 ± 0.58** | 76.98 ± 0.41 | **72.27 ± 0.41** | 78.09 ± 0.24 |

Table 28: Node level classification for hypergraph using hypegraph encodings for PhenomNN. Mean accuracy (%) ± standard deviation results over 10 train-test splits.

## G.3 HGNN

We conduct experiments using HGNN with the hyperparameters presented in Tab. 29. The results are presented in Tab. 30.

| Parameter | Value |
|---|---|
| hidden_dims | 128.0 |
| dropout_rate | 0.5 |
| learning_rate | 0.01 |
| weight_decay | 0.0005 |
| epochs | 500.0 |
| patience | 50.0 |
| val_ratio | 0.2 |
| normalize_features | False |
| normalize_encodings | False |

Table 29: HGNN configuration parameters. Model depth: 2 layers.

| Model | citeseer-CC (↑) | cora-CA (↑) | cora-CC (↑) | pubmed-CC (↑) |
|---|---|---|---|---|
| HGNN (HCP-ORC) | **58.48 ± 1.24** | **59.41 ± 1.53** | **59.93 ± 1.40** | **71.69 ± 1.41** |
| HGNN (HCP-FRC) | 56.04 ± 1.54 | 58.36 ± 1.93 | 56.37 ± 2.47 | 65.83 ± 1.70 |
| HGNN (LDP) | 57.07 ± 1.39 | **59.85 ± 1.71** | 58.06 ± 2.12 | 67.12 ± 1.30 |
| HGNN (Hodge H-20-LAPE) | **58.82 ± 1.24** | **60.10 ± 1.64** | **60.15 ± 1.52** | **72.47 ± 1.37** |
| HGNN (Norm. H-20-LAPE) | **58.68 ± 1.40** | **60.20 ± 1.55** | **60.16 ± 1.55** | **72.47 ± 1.36** |
| HGNN (H-19-RWPEE EE) | **57.79 ± 1.33** | **59.14 ± 2.15** | **59.76 ± 1.53** | **72.42 ± 1.28** |
| HGNN (H-19-RWPEE EN) | **57.73 ± 1.28** | **59.28 ± 1.97** | **59.79 ± 1.63** | **72.39 ± 1.28** |
| HGNN (H-19-RWPEE WE) | **57.69 ± 1.22** | **59.29 ± 1.91** | **59.69 ± 1.57** | **72.38 ± 1.31** |
| HGNN (no encodings) | **58.24 ± 1.20** | **60.35 ± 1.48** | **59.48 ± 1.80** | **72.60 ± 1.39** |

Table 30: Node level classification for hypergraph using hypegraph encodings for HGNN. We report mean accuracy and standard deviation over 80 runs (8 seed with ten runs each). We use the splits present in Tab 21.

# H    Rankings of encodings

We rank the hypergraph encodings by raw accuracies on all (graph model, datasets) pairs. We then average the rankings by model or by task in Tab. 31, 32, 33, 34, 35, 36. For example, we average the rankings over GPS, GCN and GIN for graph classification tasks in Tab. 31.

| Encoding | Average Rank (↓) |
|---|---|
| EE H-19-RWPE | 2.24 |
| EN H-19-RWPE | 2.53 |
| HCP-ORC | 3.13 |
| Hodge H-20-LAPE | 4.12 |
| Norm. H-20-LAPE | 4.29 |
| HCP-FRC | 4.41 |
| No Encoding | 6.71 |

Table 31: Average ranking of each hypergraph encoding across all dataset evaluations on graph classification tasks using the models GPS, CGN, and GIN. Lower is better. A total of 17 evaluations were considered.

| Encoding | Average Rank (↓) |
|---|---|
| EE H-19-RWPE | 1.43 |
| EN H-19-RWPE | 2.14 |
| HCP-ORC | 4.00 |
| Hodge H-20-LAPE | 4.28 |
| HCP-FRC | 4.71 |
| Norm. H-20-LAPE | 5.71 |
| No Encoding | 6.43 |

Table 32: Average ranking of each hypergraph encoding across all dataset evaluations on graph classification tasks using GPS. Lower is better. A total of 7 evaluations were considered.

| Encoding | Average Rank (↓) |
|---|---|
| EE H-19-RWPE | 2.43 |
| HCP-ORC | 2.67 |
| EN H-19-RWPE | 3.00 |
| HCP-FRC | 3.86 |
| Hodge H-20-LAPE | 4.14 |
| Norm. H-20-LAPE | 4.29 |
| No Encoding | 6.71 |

Table 33: Average ranking of each hypergraph encoding across all dataset evaluations on graph classification tasks using GCN. Lower is better. A total of 7 evaluations were considered.

| Encoding | Average Rank (↓) |
|---|---|
| EE H-19-RWPE | 3.67 |
| EN H-19-RWPE | 2.33 |
| HCP-ORC | 2.33 |
| Hodge H-20-LAPE | 3.67 |
| Norm. H-20-LAPE | 4.33 |
| HCP-FRC | 5.00 |
| No Encoding | 6.67 |

Table 34: Average ranking of each hypergraph encoding across all dataset evaluations on graph classification tasks using GIN. Lower is better. A total of 3 evaluations were considered. Note that the rows are not ordered by average rank here.

| Encoding | Average Rank (↓) |
|---|---|
| EE H-19-RWPE | 1.38 |
| EN H-19-RWPE | 2.13 |
| HCP-ORC | 4.0 |
| Hodge H-20-LAPE | 4.0 |
| Norm. H-20-LAPE | 5.5 |
| HCP-FRC | 4.63 |
| No Encoding | 6.5 |

Table 35: Average ranking of each hypergraph encoding across all dataset evaluations on graph classification and regressions tasks using GPS. Lower is better. A total of 8 evaluations were considered. Note that the rows are not ordered by average rank here.

| Encoding | Average Rank (↓) |
|---|---|
| EE H-19-RWPE | 2.75 |
| EN H-19-RWPE | 3.25 |
| HCP-ORC | 2.86 |
| Hodge H-20-LAPE | 3.88 |
| Norm. H-20-LAPE | 3.88 |
| HCP-FRC | 3.75 |
| No Encoding | 6.75 |

Table 36: Average ranking of each hypergraph encoding across all dataset evaluations on graph classification and regressions tasks using GCN. Lower is better. A total of 8 evaluations were considered. Note that the rows are not ordered by average rank here.

# I  Runtimes

We present the runtimes necessary to compute the graph and hypergraph encodings on Imdb in Tab. 37 and on Mutag in Tab. 38.

## I.1  On Imdb

| Encoding Method | Encodings on original graphs | Encodings on lifted hypergraphs |
|---|---|---|
| Random Walk EN | 0.0007 ± 0.0009 | 0.0006 ± 0.0008 |
| Curvature FRC | 0.0009 ± 0.0005 | 0.0007 ± 0.0003 |
| Degree | 0.0010 ± 0.0005 | 0.0010 ± 0.0005 |
| Laplacian Hodge | 0.0014 ± 0.0022 | 0.0003 ± 0.0002 |
| Laplacian Normalized | 0.0075 ± 0.0113 | 0.0007 ± 0.0003 |
| Random Walk EE | 0.0182 ± 0.0397 | 0.0022 ± 0.0031 |
| Curvature ORC | 5.1573 ± 0.1235 | 5.1990 ± 0.0994 |

Table 37: Encoding runtimes (mean ± std in seconds) on original graphs and clique-lifted hypergraphs for the first 50 graphs of Imdb.

## I.2  On Mutag

| Encoding Method | Encodings on original graphs | Encodings on lifted hypergraphs |
|---|---|---|
| Laplacian Hodge | 0.0003 ± 0.0001 | 0.0003 ± 0.0001 |
| Random Walk EN | 0.0004 ± 0.0002 | 0.0004 ± 0.0001 |
| Curvature FRC | 0.0007 ± 0.0002 | 0.0007 ± 0.0002 |
| Degree | 0.0007 ± 0.0002 | 0.0007 ± 0.0002 |
| Laplacian Normalized | 0.0012 ± 0.0006 | 0.0007 ± 0.0002 |
| Random Walk EE | 0.0024 ± 0.0012 | 0.0017 ± 0.0007 |
| Curvature ORC | 5.3687 ± 0.2250 | 5.3564 ± 0.1863 |

Table 38: Encoding runtimes (mean ± std in seconds) on original graphs and clique-lifted hypergraphs for the first 50 graphs in Mutag.

# J  Hardware specifications and libraries

All experiments in this paper were implemented in Python using PyTorch, Numpy PyTorch Geometric, and Python Optimal Transport.

Our experiments were conducted on a local server with the specifications presented in Tab. J.

| Components | Specifications |
|---|---|
| Architecture | X86_64 |
| OS | UBUNTU 20.04.5 LTS x86_64 |
| CPU | AMD EPYC 7742 64-CORE |
| GPU | NVIDIA A100 TENSOR CORE |
| RAM | 40GB |

Table 39: Server specifications.

