# OpenReview forum: "Higher-Order Learning with Graph Neural Networks via Hypergraph Encodings"
_NeurIPS.cc/2025/Conference — NeurIPS 2025 poster_

### Official Review · Reviewer_GZBz · 2025-06-03

**Clarity:** 3
**Significance:** 2
**Originality:** 3
**Rating:** 4
**Confidence:** 4

**Summary:**

This paper proposes a data augmentation approach for graph neural networks (GNNs) by introducing hypergraph-level encodings from hypergraph Laplacians and discrete curvature to capture higher-order structural information.

The authors theoretically demonstrate that these encodings enhance the representational power of GNNs.

Experiments demonstrate improvements on various datasets, particularly when used with graph-level architectures applied to hypergraphs or lifted graphs.

**Questions:**

1. Is it possible that current hypergraph message-passing schemes (e.g., the two-stage node-to-hyperedge-to-node employed by UniGCN) inherently struggle to integrate such explicit, pre-computed node-level features effectively compared to how standard GNNs integrate them?
2. Could the normalization or aggregation steps within these HNNs be inadvertently diminishing the unique information provided by the unnormalized encodings (as hinted at in Remark 3.14 regarding normalization obscuring differences)?
3. What was the approximate pre-processing time for the encodings on representative datasets?
4. How much did the size of the graph (nodes/edges) increase when lifted to a hypergraph (nodes/hyperedges/total incidence entries), and how did this impact subsequent GNN training time/memory compared to using only graph-level encodings or no encodings?

**Ethical Concerns:**

["NO or VERY MINOR ethics concerns only"]

**Final Justification:**

After considering the rebuttal and discussions, I acknowledge that several concerns were addressed by the authors. The theoretical analysis is well-supported, preprocessing overhead is minor, and clearly quantified. Extensive experiments in the rebuttal demonstrate thoughtful and transparent empirical validation.

**Limitations:**

yes

**Paper Formatting Concerns:**

No major issues.

**Quality:**

2

**Strengths And Weaknesses:**

### Strengths

1. The paper provides theoretical analysis in the form of Theorems 3.2, 3.5, 3.11, 3.13, demonstrating that the proposed hypergraph-level encodings provably increase the representational power of GNNs.
2. The authors provide an anonymized codebase and extensive appendices detailing architectures (Appendix B), datasets (Appendix C), hyperparameters (Appendix C.2), and further experimental results (Appendix D, G) and expressivity analysis (Appendix E, F).

$~$

### Weaknesses

1. The experiments primarily focus on performance improvements over old models (e.g., GCN, UniGCN, GPS) and lack a comparison against state-of-the-art baselines, see surveys [1, 2]. Moreover, it is unclear from the paper how the demonstrated improvements compare against the current state-of-the art results on the respective datasets.
2.   The observation that dedicated hypergraph neural networks (like UniGCN) do not benefit significantly from these hypergraph-level encodings is intriguing and counter-intuitive (lines 276-278, Appendix G). While the paper suggests it as an area for future work (lines 340-343), a more elaborated discussion or hypothesis in the main text about why this might be the case (e.g., are HNNs already implicitly capturing this, or are current message-passing schemes in HNNs not well-suited to integrate such explicit features?) would add depth.
3. A more in-depth discussion of the practical computational overhead, especially for very large graphs leading to dense hypergraph expansions would be beneficial. An accuracy - run time trade-off analysis, including the pre-processing times, would significantly strengthen the paper's empirical contributions.

$~$

References
1. A Survey on Hypergraph Neural Networks: An In-Depth and Step-By-Step Guide, In KDD 2024.
2. A Survey of Graph Transformers: Architectures, Theories and Applications, In arXiv 2502.16533.

---

> ### Author Rebuttal · Authors · 2025-07-31
>
> ## General response
>
> Due to the character limit, please see reviewer [UZHC] for our general response. We apologize for the inconvenience.
>
> **We now address each of the reviewer's comments and questions individually:**
>
> > The experiments primarily focus on performance improvements over old models (e.g., GCN, UniGCN, GPS) and lack a comparison against state-of-the-art baselines [...] it is unclear from the paper how the demonstrated improvements compare against the current state-of-the art results on the respective datasets.
>
> - Thank you for voicing this concern. As mentioned in the general response, the GNNs we consider such as GCN and GIN have recently been shown to still outperform many newer models, including GTs, in both node- and graph-level tasks [1], [2], [3]. Using the hyperparameters provided in [3] or, when those are unavailable, using a more extensive hyperparameter search, we significantly increase the performance of GCN on all graph-level datasets and come within range of more advanced models such as GRIT [4] and GraphViT [5] on peptides-func and peptides-struct.
>
> |GCN+Encodings|Collab (↑)|Imdb (↑)|Reddit (↑)|Enzymes (↑)|Proteins (↑)|Peptides-f (↑)|Peptides-s (↓)|
> |---|---:|---:|---:|---:|---:|---:|---:|
> |(No Encoding)|62.69 ± 2.02|50.04 ± 2.23|68.45 ± 2.36|29.93 ± 2.64|72.00 ± 2.85|0.662 ± 0.012|0.259 ± 0.007|
> |(LCP-FRC)|70.26 ± 2.57|65.08 ± 2.65|80.26 ± 2.12|29.59 ± 2.68|72.22 ± 1.67|0.660 ± 0.011|0.258 ± 0.006|
> |(LCP-ORC)|71.94 ± 2.07|68.35 ± 1.90|81.66 ± 1.87|34.79 ± 2.52|74.84 ± 1.60|0.677 ± 0.010|0.252 ± 0.004|
> |(19-RWPE)|50.83 ± 2.53|50.77 ± 2.98|80.50 ± 2.06|31.27 ± 1.63|72.98 ± 1.83|0.674 ± 0.011|0.249 ± 0.004|
> |(20-LAPE)|58.64 ± 2.19|49.19 ± 2.14|77.66 ± 2.35|28.72 ± 1.67|72.55 ± 1.81|0.669 ± 0.009|0.250 ± 0.005|
> |(HCP-FRC)|72.34 ± 1.61|65.25 ± 1.78|83.12 ± 1.83|32.24 ± 2.36|71.64 ± 1.79|0.665 ± 0.012|0.248 ± 0.006|
> |(HCP-ORC)|70.94 ± 2.15|67.21 ± 2.37|81.53 ± 2.28|33.71 ± 2.30|75.72 ± 1.95|0.681 ± 0.007|0.246 ± 0.007|
> |(EE H-19-RWPE)|71.36 ± 2.25|74.82 ± 2.06|82.88 ± 1.96|31.98 ± 2.11|75.38 ± 2.42|0.683 ± 0.008|0.245 ± 0.004|
> |(EN H-19-RWPE)|70.05 ± 1.88|74.42 ± 1.52|84.52 ± 1.38|31.92 ± 2.02|75.93 ± 2.57|0.680 ± 0.007|0.246 ± 0.005|
> |(Hodge H-20-LAPE)|71.03 ± 1.42|72.60 ± 2.37|79.80 ± 1.59|29.53 ± 1.60|74.68 ± 2.14|0.678 ± 0.007|0.247 ± 0.004|
> |(Norm. H-20-LAPE)|69.17 ± 1.43|71.33 ± 1.59|80.21 ± 2.12|31.42 ± 2.83|74.32 ± 1.30|0.684 ± 0.009|0.245 ± 0.006|
>
> - We would be happy to provide additional results using GIN, GPS, or another model the reviewer might be particularly interested in during the discussion period.
>
> - At the hypergraph-level, we provide additional results using PhenomNN [6], a HGNN that is closer to the state-of-the-art. However, we still find that the model does not benefit from any of our hypergraph-level encodings, which we believe might indicate that this is a more general phenomenon affecting HGNNs more broadly (as the reviewer also seemed to suspect). Below is the hypernode-level accuracy for hypergraphs using hypergraph encodings with PhenomNN. Mean accuracy (%) ± standard deviation over 10 train-test splits.
>
> |**PhenomNN + Encoding**|**citeseer-CC**|**cora-CA**|**cora-CC**|**pubmed-CC**|
> |---|---:|---:|---:|---:|
> |No encoding|64.96 ± 0.58|76.98 ± 0.41|72.27 ± 0.41|78.09 ± 0.24|
> |HCP-FRC|58.87 ± 1.03|74.03 ± 0.97|54.02 ± 2.06|68.88 ± 1.44|
> |HCP-ORC|64.89 ± 0.45|75.93 ± 0.41|72.21 ± 0.31|77.37 ± 0.23|
> |LDP|62.54 ± 0.52|75.67 ± 0.50|62.28 ± 3.33|70.63 ± 0.87|
> |Hodge H-20-LAPE|64.98 ± 0.58|76.76 ± 0.71|72.18 ± 0.33|78.36 ± 0.23|
> |Norm. H-20-LAPE|64.75 ± 0.52|76.64 ± 0.49|72.06 ± 0.68|78.27 ± 0.25|
> |H-20-RWPEE EE|64.23 ± 0.40|76.24 ± 0.57|72.18 ± 0.52|77.97 ± 0.29|
> |H-20-RWPEE EN|64.47 ± 0.62|76.39 ± 0.57|71.57 ± 0.73|78.10 ± 0.20|
> |H-20-RWPEE WE|64.52 ± 0.69|76.54 ± 0.57|72.30 ± 0.79|78.06 ± 0.24|
>
> > The observation that dedicated hypergraph neural networks (like UniGCN) do not benefit significantly from these hypergraph-level encodings is intriguing [...] a more elaborated discussion or hypothesis in the main text about why this might be the case [...] would add depth.
>
> - Thank you very much for raising this important point. We agree that this is indeed a very interesting point. We suspect that since hypergraph layers already process the full incidence structure, the co-incidence statistics that, for example, our Hodge encodings add are largely redundant: most of that signal is already available to the architecture. By contrast, a standard graph GNN sees only pairwise edges; the encodings inject complementary higher-order cues it would otherwise miss, hence the larger gains. We believe that this intuition could be made rigorous using techniques similar to [7], which operates at the graph level, but given the length and complexity of their arguments, we consider this beyond the scope of the current paper.
>
> > A more in-depth discussion of the practical computational overhead [...] would significantly strengthen the paper's empirical contributions.
>
> - Thank you very much for raising this important point. Please note that the lifting of graphs to hypergraphs (if working on graph data) and the computing of encodings is a pre-processing step applied only once before feeding the feature matrix to models. No further computations are needed during training. The empirical runtime results have been added in the paper, and some can be found below. For every dataset, we compute the average time needed to compute the encodings on the hypergraphs. If the dataset is a graph dataset, we also include the time needed to lift graphs to hypergraphs, and the time needed to compute the graph encodings. Below are the runtimes for IMDB (the mean ± std are in seconds) on original graphs and clique-lifted hypergraphs:
>
> |Encoding Method|Encodings on original graphs|Encodings on Lifted hypergraphs|
> |---|---:|---:|
> |EN 19-RWPE & EN H-19-RWPE|0.0007 ± 0.0009|0.0006 ± 0.0008|
> |LCP-FRC & HCP-FRC|0.0009 ± 0.0005|0.0007 ± 0.0003|
> |LDP & HDP|0.0010 ± 0.0005|0.0010 ± 0.0005|
> |Hodge 20-LAPE & Hodge H-20-LAPE|0.0014 ± 0.0022|0.0003 ± 0.0002|
> |Norm. 20-LAPE & Norm. H-20-LAPE|0.0075 ± 0.0113|0.0007 ± 0.0003|
> |EE 19-RWPE & EE H-19-RWPE|0.0182 ± 0.0397|0.0022 ± 0.0031|
> |LCP-ORC & HCP-ORC|5.1573 ± 0.1235|5.1990 ± 0.0994|
>
> - The average time needed to lift the graph to hypergraphs is 0.0003s ± 0.0005s. Note that our Python code is not optimized. Below are the runtimes on Citeseer:
>
> |Encoding Method|Citeseer-CC|
> |---|---:|
> |HCP-FRC|0.0710|
> |H-LDP|0.0906|
> |Normalized H-20-LAPE|0.9175|
> |EN H-19-RWPE|2.8710|
> |Hodge H-20-LAPE|2.8988|
> |HCP-ORC*|13.6736|
> |EE H-19-RWPE|183.6963|
>
> - Note that HCP-ORC calls a julia routine.
>
> > Is it possible that current hypergraph message-passing schemes [...] inherently struggle to integrate such explicit, pre-computed node-level features effectively compared to how standard GNNs integrate them?
>
> - Please see the response above.
>
> > Could the normalization or aggregation steps within these HNNs be inadvertently diminishing the unique information provided by the unnormalized encodings [...]?
>
> - Thank you for these questions. **Normalization:** We ran the experiments for UniGNN [1] without normalizing the encodings (we also ran the experiments with normalization, although they are not reported in the paper - the performance differences where within standard deviations). In the graph case, we don't normalize the features before the first layer at all. **Aggregation:** We originally suspected that oversmoothing might be to blame here, i.e. node features becoming increasingly similar with depth, thereby washing out the information provided by the encodings. We do not believe that this is the case though because 1) the HGNNs we use are not very deep (always less than 10 layers) and 2) some of the HGNNs we consider were specifically designed to avoid oversmoothing, such as UniGCN, which suffers minimal performance drops with 64 layers compared to 4 layers, for example.
>
> > How much did the size of the graph (nodes/edges) increase when lifted to a hypergraph [...]?
>
> - The size of the set of vertices remains unchanged when we lift a graph to a hypergraph, because, in our setting we lift graphs to hypergraphs by adding hyperedges to groups of nodes that are pairwise interconnected. Thus the set of vertices of the hypergraph is the same as the set of vertices of the original graph. The number of hyperedge in the lifted hypergraph is at most the number of edges in the original graph.
>
> [1] Luo, Y., Shi, L. and Wu, X.M., Can Classic GNNs Be Strong Baselines for Graph-level Tasks? Simple Architectures Meet Excellence. In Forty-second International Conference on Machine Learning.
>
> [2] Luo, Y., Shi, L. and Wu, X.M., 2024. Classic gnns are strong baselines: Reassessing gnns for node classification. Advances in Neural Information Processing Systems, 37.
>
> [3] Tönshoff, J., Ritzert, M., Rosenbluth, E. and Grohe, M., Where Did the Gap Go? Reassessing the Long-Range Graph Benchmark. In The Second Learning on Graphs Conference.
>
> [4] Ma, L., Lin, C., Lim, D., Romero-Soriano, A., Dokania, P.K., Coates, M., Torr, P. and Lim, S.N., 2023, July. Graph inductive biases in transformers without message passing. In International Conference on Machine Learning. PMLR.
>
> [5] He, X., Hooi, B., Laurent, T., Perold, A., LeCun, Y. and Bresson, X., 2023, July. A generalization of vit/mlp-mixer to graphs. In International conference on machine learning. PMLR.
>
> [6] Wang, Y., Gan, Q., Qiu, X., Huang, X. and Wipf, D., 2023, July. From hypergraph energy functions to hypergraph neural networks. In International Conference on Machine Learning. PMLR.
>
> [7] Kanatsoulis, C., Choi, E., Jegelka, S., Leskovec, J. and Ribeiro, A., Learning Efficient Positional Encodings with Graph Neural Networks. In The Thirteenth International Conference on  Learning Representations.

---

### Official Review · Reviewer_6gkV · 2025-06-17

**Clarity:** 3
**Significance:** 3
**Originality:** 3
**Rating:** 5
**Confidence:** 3

**Summary:**

The paper works on hypergraphs and neural networks for them. The main contribution is the lifting of common positional and structural encodings from graphs to hypergraphs for which they claim that they are always superior to their graph counterpart, even though all datasets are lifted graph datasets (mostly because there are no true hypergraph datasets available).

**Questions:**

1) How do a pure GT and MLP react to the hypergraph embeddings?

2) Do the main results (i.e. hypergraph variant is always better than "normal" one) carry over when getting closer to SOTA results?  E.g. there seems to be diminishing returns for GIN which starts with a higher performance.

3) What is the main difference between UniGCNII and a model such as HGNN which reports a lot better results on cora and pubmed? (on pubmed a simple GCN is stronger than any of the hypergraph networks)

**Ethical Concerns:**

["NO or VERY MINOR ethics concerns only"]

**Final Justification:**

The weaknesses (especially the experimental evaluation) has been thoroughly addressed in the rebuttal and the updated experiments support the claims of the paper well. Doing the promised changes is well possible within the time until the camera ready version.

**Limitations:**

yes

**Quality:**

3

**Strengths And Weaknesses:**

**Strengths:**
The main strength is a clear and lifting procedure for the structural and positional embeddings, allowing them to make sense of hyperedges. The experiments indicate that each hypergraph-lifted encoding is always better than the corresponding graph-level encoding. The paper is well-written and easy to follow.

**Weaknesses:**
mostly the experimental evaluation. While the authors (roughly) claim that "holding the parameter constant makes everything more fair", I strongly disagree with this claim. In the experiments, the absolute performance of the models is also really low and for example a simple HGNN from 2019[1] achieves significantly higher performance than any of the models in this paper. I would have expected this model to be a baseline.

My main problem with the experiments is that the provided results are not representative of the model for each of the datasets and thus the performance gains through additional encodings could behave entirely different when performing a small hyperparameter search based on a generally good architecture. Instead of reporting 50 trained models for each dataset, reporting the average of 5 while using the other 45 for hyperparameter tuning would be much preferred by me.

I would also like to see a plain graph transformer and a simple MLP as baselines as both architectures rely only on encodings to learn about the graph/hypergraph structure.


** further detailed comments**
- please use booktabs.
- In general, the tables are hard to read and it is not directly clear what they show - maybe it is possible to make a plot that directly compares each PE with its lifted variant?
- having only the "best" value in bold but not everything within $1\sigma$ or with overlapping standard deviations is problematic as it indicates that there is a single best model while this is not significant and actually multiple models are equally good.
- it could be made more clear that there are settings where both graphs and hypergraphs are used "together"
- 306: with encodings (e.g. LapPE) there is no difference in expressiveness between any of the models (at least in terms of non-uniform expressivity). GPS internally uses GIN, so it should not be worse-off. It would be nice that all those expressiveness results only hold in the regime without encodings.
- Leman is written without h.
- 65ff: message passing only sometimes add self-loops while the formula assumes that this is always the case.
- 74: maybe also cite a second one here such as LGI-GT [2] that follows up on GPS
- 91: as encodings are crucial for graph transformers, I would have expected this to be much more prominent here. I was also surprised to find RWSE later but not in this paragraph.


[1] Hypergraph Neural Networks, Feng et al. AAAI 2019
[2] LGI-GT: Graph Transformers with Local and Global Operators Interleaving, Yin and Zhong IJCAI-23

---

> ### Author Rebuttal · Authors · 2025-07-31
>
> ## General response
>
> Due to the character limit, please see reviewer [UZHC] for our general response. We apologize for the inconvenience.
>
> **We now address each of the reviewer's comments and questions individually:**
>
> > mostly the experimental evaluation. While the authors (roughly) claim that "holding the parameter constant makes everything more fair", I strongly disagree with this claim. [...].
>
> - We thank the reviewer for voicing their concerns. Our experimental evaluation was indeed meant to ensure fairness and followed [1] and [2].
> - We see where the reviewer's skepticism is coming from  and provide additional results with hyperparameter tuning below. We will include both results, along with further hyperparameter searches for all models.
> - To select the hyperparameters for these additional experiments, we conducted a grid search over the following hyperparameters: num. layers $\in \{4, 6, 8\}$, hidden dim $\in \{64, 128\}$, learning rate $\in \{0.01, 0.001\}$, and dropout $\in \{0.25, 0.5\}$. For the configuration with the highest validation accuracy, we provide the average test accuracy over five random seeds in the table below (standard deviations are therefore slightly larger). We refer the reviewer to the updated GCN table in the response to reviewer [GZBz] due to the character limit.
>
> - For the node classifications tasks on hypergraph-level datasets, we clarify that: A- the choice of UniGCN in table 1 is meant as a direct comparison between GCN and UniGCN; B - we use the same data and splits as in UniGNN [4] - we are performing semi-supervised hypernode classification, with a small percentage of hypernodes used for training.
> - While [5] reports 80.1% on pubmed (table 2), our choice was originally motivated because [4] seems to indicate that UniGNN and UniGNNII achieve better accuracies on our particular train/test/validation splits. Please see Table 1 and 3 in UniGNN [4]: UniGNN and UniGNNII are superior to HGNN  on this task. [6] presents similar numbers for HGNN. We will provide the experiments with HGNN in the discussion period.
>
> - We are also adding more hypergraph-based models (eg PhenomNN [7]) to address the point raised by the reviewer. We still find that the model does not benefit from any of our hypergraph-level encodings, which we believe indicate that this is a more general phenomenon affecting HGNNs more broadly.
>
> | **Model**| **citeseer-CC**| **cora-CA**| **cora-CC**        | **pubmed-CC**      |
> |----------------------------------|--------------------|--------------------|--------------------|--------------------|
> | PhenomNN (no encodings)| 64.96 ± 0.58       | 76.98 ± 0.41   | 72.27 ± 0.41       | 78.09 ± 0.24       |
> | PhenomNN (HCP-ORC)| 64.89 ± 0.45       | 75.93 ± 0.41       | 72.21 ± 0.31       | 77.37 ± 0.23       |
> | PhenomNN (LDP)| 62.54 ± 0.52| 75.67 ± 0.50       | 62.28 ± 3.33       | 70.63 ± 0.87       |
> | PhenomNN (Hodge H-20-LAPE)       | 64.98 ± 0.58   | 76.76 ± 0.71       | 72.18 ± 0.33       | 78.36 ± 0.23   |
> | PhenomNN (H-20-RWPEE EN)         | 64.47 ± 0.62       | 76.39 ± 0.57       | 71.57 ± 0.73       | 78.10 ± 0.20       |
>
> > My main problem with the experiments is that the provided results are not representative of the model for each of the datasets [...].
>
> - We hope that the additional results after careful hyperparameter tuning presented above alleviate the reviewer's concerns. We would be happy to provide additional results using GIN, GPS, or another model the reviewer might be interested in during the discussion period.
>
> > I would also like to see a plain graph transformer and a simple MLP as baselines [...].
>
> - Please find additional results with an MLP and GT below. Hyperparameters were again obtained using grid search as above.
> - We would be happy to provide additional results using other datasets during the discussion period.
>
> | Model (Encodings)| Collab (↑)| Imdb (↑)| Reddit (↑)|
> |----------------------------|-----------------|-----------------|-----------------|
> | MLP (No Encoding)| 51.80 ± 1.55    | 48.26 ± 1.98    | 49.08 ± 1.92    |
> | MLP (LCP-FRC)| 64.09 ± 1.51    | 48.96 ± 2.40    | 71.36 ± 3.26    |
> | MLP (LCP-ORC)| 67.86 ± 1.94    | 63.26 ± 3.13    | 76.04 ± 2.08    |
> | MLP (19-RWPE)              | 50.21 ± 2.32    | 49.37 ± 2.15    | 72.19 ± 2.05    |
> | MLP (20-LAPE)              | 57.12 ± 1.77    | 49.19 ± 1.48    | 74.53 ± 1.67    |
> | MLP (HCP-FRC)              | 68.45 ± 1.62    | 60.22 ± 2.07    | 78.33 ± 2.12    |
> | MLP (HCP-ORC)              | 67.90 ± 1.58    | 62.88 ± 1.85    | 77.46 ± 2.29    |
> | MLP (EE H-19-RWPE)         | 66.14 ± 1.73    | 68.37 ± 1.73    | 79.17 ± 1.62    |
> | MLP (Hodge H-20-LAPE)      | 66.71 ± 1.45    | 66.84 ± 1.91    | 76.48 ± 1.83    |
>
> | Model (Encodings)| Collab (↑)      | Imdb (↑)        | Reddit (↑)      |
> |-----------------------------|-----------------|-----------------|-----------------|
> | GT (No Encoding)| 53.07 ± 1.52    | 49.56 ± 1.28    | 51.04 ± 1.14    |
> | GT (LCP-FRC)| 69.13 ± 2.06    | 50.24 ± 2.15    | 75.93 ± 1.80    |
> | GT (LCP-ORC)| 70.46 ± 1.84    | 61.45 ± 2.03    | 78.42 ± 1.79    |
> | GT (19-RWPE)| 60.39 ± 2.33    | 61.94 ± 2.12    | 77.03 ± 2.18    |
> | GT (20-LAPE)| 61.87 ± 2.21    | 60.92 ± 2.24    | 78.31 ± 2.27    |
> | GT (HCP-FRC)| 70.88 ± 1.01    | 60.83 ± 1.16    | 80.02 ± 2.12    |
> | GT (HCP-ORC)| 69.12 ± 1.17    | 66.44 ± 1.48    | 80.34 ± 2.30    |
> | GT (EE H-19-RWPE)| 68.14 ± 1.09    | 70.79 ± 1.37    | 81.57 ± 2.24    |
> | GT (Hodge H-20-LAPE)| 67.38 ± 0.92    | 70.58 ± 1.42    | 80.28 ± 2.21    |
>
> > please use booktabs.
>
> - Thank you for pointing us to the package. We have updated our tables.
>
> > In general, the tables are hard to read and it is not directly clear what they show [...].
>
> - Great suggestion. A plot for each of table 2 and 3, with accuracies plotted on the y-axis has been added.
> - It is indeed our intent with table 2 and table 3 to compare PEs to their lifted variants. We refer the reviewer to the response to reviewer [Ej7N] where we add average ranking of each encodings.
>
> > having only the "best" value in bold but not everything within
>  or with overlapping standard deviations is problematic [...] multiple models are equally good.
>
> - This is a great point. We have now added a scheme where we underline the encodings that fall in the $\pm 1 std$ band of the best encodings accuracy. These suggest more clearly that on graph-classification tasks, RWPE should be used and that on graph regression tasks and node classification tasks, H-LAPE should be used.
>
> > it could be made more clear that there are settings where both graphs and hypergraphs are used "together"
>
> - Could the reviewer clarify this point? We do compute graph-based and hypergraph-based encodings (on the graphs' liftings) to compare these side by side after passing them through various models for training. However, we do not believe our method jointly uses graphs and hypergraphs in the same model.
>
> > 306: with encodings (e.g. LapPE) there is no difference in expressiveness between any of the models (at least in terms of non-uniform expressivity). GPS internally uses GIN, so it should not be worse-off. It would be nice that all those expressiveness results only hold in the regime without encodings.
>
> - We have adjusted that line to reflect that in the WL hierarchy, GIN is not necessarily more expressive than GPS.
> - This is indeed an interesting point. Our table 14 (682ff) does suggest that the different expressiveness results only hold in the regime without encodings: we have clarified the text.
>
> > Leman is written without h.
>
> - Thank you. The text is updated.
>
> > 65ff: message passing only sometimes add self-loops [...].
>
> - Thank you for noting this and apologies for the oversight. We have updated this.
>
> > 74: maybe also cite a second one here such as LGI-GT [2] that follows up on GPS
>
> - Thank you for this suggestion. We have added this citation.
>
> > 91: as encodings are crucial for graph transformers, I would have expected this to be much more prominent here. [...]
>
> - Thank you for this suggestion. We have changed this paragraph to mention the importance of encodings for graph transformers as well as MPGNNs. Although we do mention random-walk based node similarities, these encodings are now mentioned explicitly in the paragraph.
>
> > How do a pure GT and MLP react to the hypergraph embeddings?
>
> - Please see additional results above.
>
> > Do the main results (i.e. hypergraph variant is always better than "normal" one) carry over when getting closer to SOTA results? E.g. there seems to be diminishing returns for GIN which starts with a higher performance.
>
> - Please see additional results above/ the response to reviewer [GZBz].
>
> > What is the main difference between UniGCNII and a model such as HGNN [...]
>
> - We added two additional baselines, including a recent one (PhenomNN) and based in the figures reported in the literature, we are close to SOTA on the hypernode classifications using UniGNNII.
> - The overall insight that we are reporting - HGNNs do not benefit from hypergraph encodings- still stands.
>
> [1] Fesser, Weber, Effective Structural Encodings via Local Curvature Profiles. In The Twelfth International Conference on Learning Representations.
>
> [2] Nguyen, Hieu, Nguyen, Ho, Osher, and Nguyen. Revisiting over-smoothing and over-squashing using ollivier-ricci curvature. In International Conference on Machine Learning (pp. 25956-25979). PMLR.
>
> [3] Tönshoff, Ritzert, Rosenbluth, and Grohe. Where did the gap go? reassessing the long-range graph benchmark.
>
> [4] Huang & Yang. Unignn: a unified framework for graph and hypergraph neural networks.
>
> [5] Feng, You, Zhang, Ji, & Gao. Hypergraph neural networks.
>
> [6] Zhang, Li, Xiao, Xu, Rong, Huang, & Bian. Hypergraph convolutional networks via equivalency between hypergraphs and undirected graphs.
>
> [7] Wang, Gan, Qiu, Huang, & Wipf. From hypergraph energy functions to hypergraph neural networks.

---

> > ### Author Response · Authors · 2025-08-04
> >
> > Dear reviewer, thank you again for your constructive feedback on our paper and for considering our rebuttal. Kindly let us know whether your concerns have been addressed or if there are any remaining concerns that you might have.
> >
> > We would be happy to provide further clarification or run additional experiments during the remaining discussion period.

---

> > ### Comment · Reviewer_6gkV · 2025-08-05
> >
> > I thank the authors for the detailed response!
> >
> > **Experiments**:
> > I did not check the UniGCN paper and only looked at paperswithcode, so this is where the confusion came from. It could be a good idea to highlight the particular challenges of the task used here (and in case you have an idea where the difference to the corresponding task in HGCN comes from, this would also be welcome and provide helpful context).
> > Also thanks for the additional baseline, working closer to sota makes the claims a lot more credible to me (this also includes the hyperparameter optimization).
> >
> > I find this new MLP table extremely interesting: it seems to show that the encodings are strong enough that even simple MLPs get close to the performance of message-passing networks or hypergraph networks, even though there is always a clear gap, it is smaller than I expected. Also the claim that the lifted PEs are generally stronger stays valid (maybe except for FRC but otherwise the difference is often quite big).
> >
> > Regarding "graphs and hypergraphs together": I was looking for a more clear distinction of the four settings used in the paper: Graph-level GNN plus graph-level PE, Graph level GNN plus hypergraph-lifted PE, hypergraph GNN plus graph-level PE, and finally hypergraph with hypergraph-PE. For some of those, one uses both the original graph plus its lifted counterpart. And for generalization: if given a (real) hypergraph, which corresponding graph would be used to run GCN on? I.e. what happens in settings where the input is a hypergraph? Or is this a setting that is not relevant to the paper?
> >
> > Overall, my concerns about the paper have been fully adressed and I find the updated results highly interesting, so thanks a lot for putting in the effort to run all those additional experiments and reporting them during the rather short rebuttal period! I have also increased my score as the weaknesses mentioned in the review have been fully adressed.

---

> > > ### Author Response · Authors · 2025-08-05
> > >
> > > We would like to thank the reviewer for their detailed response to our rebuttal and for their encouraging feedback!
> > >
> > > > Experiments: I did not check the UniGCN paper and only looked at paperswithcode, so this is where the confusion came from. It could be a good idea to highlight the particular challenges of the task used here (and in case you have an idea where the difference to the corresponding task in HGCN comes from, this would also be welcome and provide helpful context). Also thanks for the additional baseline, working closer to sota makes the claims a lot more credible to me (this also includes the hyperparameter optimization).
> > >
> > > We completely agree with the reviewer and will include a remark about performance differences between papers in the final manuscript. We will also try and dig a bit deeper to figure out where exactly these differences in the literature are coming from.
> > >
> > > > I find this new MLP table extremely interesting: it seems to show that the encodings are strong enough that even simple MLPs get close to the performance of message-passing networks or hypergraph networks, even though there is always a clear gap, it is smaller than I expected. Also the claim that the lifted PEs are generally stronger stays valid (maybe except for FRC but otherwise the difference is often quite big).
> > >
> > > We thank the reviewer for pointing this out! We also found the additional MLP and GT experiments very insightful and will be sure to include them in the final manuscript.
> > >
> > > > Regarding "graphs and hypergraphs together" […] if given a (real) hypergraph, which corresponding graph would be used to run GCN on? I.e. what happens in settings where the input is a hypergraph? Or is this a setting that is not relevant to the paper?
> > >
> > > We use clique-expansions in this case, i.e. we keep the nodes of the original hypergraph and generate the edges of the graph by turning each hyperedge into a clique, so each pair of nodes in the hyperedge is connect by an edge in the resulting graph. See, for example, section 4.2 in the main text or section A in the appendix.

---

### Official Review · Reviewer_Ej7N · 2025-06-25

**Clarity:** 2
**Significance:** 3
**Originality:** 3
**Rating:** 4
**Confidence:** 3

**Summary:**

The paper proposes to use hypergraph encodings in order to enhance performance of graph neural networks. These encodings are designed to capture higher-order structural and positional information, which cannot be captured by standard MPNNs. Expressivity is also partly investigated theoretically.
The experimental evaluation shows that the encodings outperform graph-level methods, especially on social networks.

**Questions:**

Please improve the quality of the paper by making sure that all definitions are correct and included. Please define all variables used. I like the idea of the paper, but in the current state, I am against accepting it.

**Q1** Which of the encodings should be used? Can you say anything about which one might be better depending on the dataset?

**Q2** What is the computational complexity of all the proposed encodings (there are some missing an analysis)? What was the runtime for computing them in practice?

**Q3** Regarding the significance of the proposed methods.

a) For hypergraphs: As stated in the discussion section, it seems there is not really data available that is naturally represented as hypergraphs. At first glance, it does not seem to be useful then, or am I missing something?

b) For traditional graphs: In the experimental evaluation, the encodings outperform the models without encoding. However, they are not compared to state-of-the-art GNNs, so this seems more like an ablation study. Can you comment on that?

**Q4** Why are the last three entries in Equation 8 (and in LDP) repeated?

**Ethical Concerns:**

["NO or VERY MINOR ethics concerns only"]

**Final Justification:**

The authors have addressed my concerns in their rebuttal and provided additional results.
Given this, I would lean slightly more towards acceptance.
I think the formatting and quality issues can be addressed within the time frame of the rebuttal/until the camera-ready version is due.

**Limitations:**

Computational complexity of the proposed encodings was not discussed as a limitation.

**Paper Formatting Concerns:**

no concerns

**Quality:**

1

**Strengths And Weaknesses:**

**S1** The idea of the paper is nice, using encodings for improving MPNN-performance is an interesting research direction. Since the source code is available online, and the experimental setup is described in great detail, one should be able to reproduce the results easily.

**S2** The discussion at the end is really extensive. Most limitations are discussed in detail.

**W1** The presentation of the paper can be improved significantly. In the background section several definitions are incomplete or missing. For the graph definition, V should be included (G=(V,X,E)) and F should be defined formally. The neighborhood of a vertex is not formally defined. The abbreviations GNN and MPGNN are used but never introduced. A short intuition of how a graph is transformed into a hypergraph should be given within the paper (currently it is only in the appendix). D, d_k, A, RW (and maybe other variables) are never defined. Some variables (like d) also seem to be used multiple times in different ways. H-LDP is also not formally defined. In the references, there are multiple duplicate entries.

**W2** From the experimental evaluation, it seems as though no encoding consistently outperforms the others. If there are differences in performance, these could be highlighted by computing a ranking of the methods over all datasets.

**W3** The runtime needed to compute **all** the different encodings is not (neither theoretically nor empirically) shown.

Other comments:

- Many (not all) paragraph titles are missing the .
- For some sections/subsections the title is not capitalized consistently.
- A figure showing an overview of the different encodings would be nice.
---
- l17ff maybe rephrase the first sentence or split it in two.
- l171 (see F.2) maybe Appendix missing?
- l212 Both graphs were not introduced before, so "again" is not fitting (and maybe the graphs should also be defined first).
- l230 Why does the LDP not get an equation number?
- l260 When referring to tables in the appendix, please make clear that they are in the appendix.
- l302 Lehman -> Leman
- l303 hypergrpah -> hypergraph

---

> ### Author Rebuttal · Authors · 2025-07-31
>
> ## General response
>
> Due to the character limit, please see reviewer [UZHC] for our general response. We apologize for the inconvenience.
>
> **We now address each of the reviewer's comments and questions individually:**
>
> > The presentation of the paper can be improved significantly. In the background section several definitions are incomplete or missing. For the graph definition, V should be included (G=(V,X,E)) and F should be defined formally. The neighborhood of a vertex is not formally defined. The abbreviations GNN and MPGNN are used but never introduced. A short intuition of how a graph is transformed into a hypergraph should be given within the paper (currently it is only in the appendix). D, d_k, A, RW (and maybe other variables) are never defined. Some variables (like d) also seem to be used multiple times in different ways. H-LDP is also not formally defined. In the references, there are multiple duplicate entries.
>
> - We thank the reviewer for pointing this out and apologize for the oversights. The graph definition has been adjusted and definitions for the neighborhood of a vertex, the H-LDP, and missing variables have now been included. GNN and MPGNN are now formally introduced and duplicate entries in the references have been removed.
> - To provide intuition on how a graph is transformed into a hypergraph in the main text, we propose to include the following: A graph is “lifted” to a hypergraph by adding a hyperedge for every set of vertices that are pairwise connected in the graph—effectively turning each clique into a single higher-order relation.
>
> > From the experimental evaluation, it seems as though no encoding consistently outperforms the others. If there are differences in performance, these could be highlighted by computing a ranking of the methods over all datasets.
>
> - This is an excellent point for which we would like to thank the reviewer. We now compute the average ranking of each hypergraph encoding across all model and dataset evaluations on graph regression and classification tasks, as well as node classification tasks:
>
> | **Encoding**           | **Average Rank (↓)** |
> |------------------------|----------------------|
> | EE H-19-RWPE           | 2.71         |
> | EN H-19-RWPE           | 3.04                 |
> | HCP-ORC                | 3.22                 |
> | Hodge H-20-LAPE        | 3.67                 |
> | Norm. H-20-LAPE        | 4.00                 |
> | HCP-FRC                | 4.08                 |
> | No Encoding            | 6.67                 |
>
> - On graph-classification tasks, RWPE should be used. We found the EE scheme to be be superior to EN. On graph regression tasks and node classification tasks, H-LAPE should be used. We found that using the Hodge Laplacian is more beneficial than using the Normalized Laplacian. Below is the average ranking of each hypergraph encoding across GPS, GCN, and GIN on graph classification tasks:
>
> | **Encoding**         | **Average Rank (↓)** |
> |----------------------|-----------------------|
> | EE H-19-RWPE         | 2.24                  |
> | EN H-19-RWPE         | 2.53                  |
> | HCP-ORC              | 3.13                  |
> | Hodge H-20-LAPE      | 4.12                  |
> | Norm. H-20-LAPE      | 4.29                  |
> | HCP-FRC              | 4.41                  |
> | No Encoding          | 6.71                  |
>
> - Here is the average ranking of each hypergraph encoding across GCN and GPS on graph regression tasks. Lower is better.
>
> | **Encoding**         | **Average Rank (↓)** |
> |----------------------|-----------------------|
> | Hodge H-20-LAPE      | 2.0                   |
> | Norm. H-20-LAPE      | 2.5                   |
> | EE H-19-RWPE         | 3.0                   |
> | EN H-19-RWPE         | 3.5                   |
> | HCP-FRC              | 3.5                   |
> | HCP-ORC              | 4.0                   |
> | No Encoding          | 7.0                   |
>
> - If the reviewer is interested in more specific tables (per specific model or modality), we are happy to provide these too.
>
> - Furthermore, we include similar tables that consider both graph and hypergraph encodings. Beside LCP-ORC, which is expensive to compute, all hypegraphs encodings have a lower average rank. The lowest average rank is still achieved using EE RWPE and EN RWPE.
>
> - We also include better formatting of our tables (best encoding in bold, the second best underlined) and we also include ties to highlight the best encoding while taking into consideration the 1 std bands.
>
> > The runtime needed to compute all the different encodings is not (neither theoretically nor empirically) shown.
>
> - Please note that we do provide theoretical complexity analyses for our encodings. The complexity analysis for the H-LDP was missing (thank you for pointing this out) but has now been added. Additional empirical runtime results which we will include in the paper can be found below. For every dataset, we compute the average time needed to compute the encodings on the hypergraphs. If the dataset is a graph dataset, we also include the time needed to lift graphs to hypergraphs, and the time needed to compute the graph encodings. Below are the runtimes for IMDB (the mean ± std are in seconds) on original graphs and clique-lifted hypergraphs:
>
> | **Encoding Method**     | **Encodings on original graphs** | **Encodings on Lifted hypergraphs** |
> |-------------------------|----------------------------------|-------------------------------------|
> | EN 19-RWPE & EN H-19-RWPE        | 0.0007 ± 0.0009                  | 0.0006 ± 0.0008                     |
> | LCP-FRC & HCP-FRC          | 0.0009 ± 0.0005                  | 0.0007 ± 0.0003                     |
> | LDP & HDP                  | 0.0010 ± 0.0005                  | 0.0010 ± 0.0005                     |
> | Hodge 20-LAPE & Hodge H-20-LAPE         | 0.0014 ± 0.0022                  | 0.0003 ± 0.0002                     |
> | Norm. 20-LAPE & Norm. H-20-LAPE    | 0.0075 ± 0.0113                  | 0.0007 ± 0.0003                     |
> | EE 19-RWPE & EE H-19-RWPE          | 0.0182 ± 0.0397                  | 0.0022 ± 0.0031                     |
> | LCP-ORC & HCP-ORC          | 5.1573 ± 0.1235                  | 5.1990 ± 0.0994                     |
>
> - The average time needed to lift the graph to hypergraphs is 0.0003s ± 0.0005s. Below are the runtimes on Citeseer:
>
> | Encoding Method         | Citeseer-CC |
> |-------------------------|-------------|
> | HCP-FRC                 | 0.0710      |
> | H-LDP                   | 0.0906      |
> | Normalized H-20-LAPE    | 0.9175      |
> | EN H-19-RWPE            | 2.8710      |
> | Hodge H-20-LAPE         | 2.8988      |
> | HCP-ORC*                | 13.6736     |
> | EE H-19-RWPE            | 183.6963    |
>
> - Note that we only need to do this one before training, and that our Python code is not optimized. HCP-ORC calls a julia routine.
>
> > Other comments
>
> - Thank you for pointing these out, we have addressed all of these comments in our revised version of the paper.
>
> > Which of the encodings should be used? Can you say anything about which one might be better depending on the dataset?
>
> - Please refer to table above.
>
> > What is the computational complexity of all the proposed encodings (there are some missing an analysis)? What was the runtime for computing them in practice?
>
> - The complexity analysis of the H-LDP was indeed missing. Thank you for pointing this out. We have now included the following statement to make up for this:
>
> Computing H-LDP scales as $O(|F|f_{max})$, where F is the hyper-edges set and $f_{max}$ is the size of the biggest hyper-edge. We only need to count the degree for each node and save the statistics of 1-neighborhood for each node.
>
> - For the runtime, please see our earlier response.
>
> > For hypergraphs: As stated in the discussion section, it seems there is not really data available that is naturally represented as hypergraphs. At first glance, it does not seem to be useful then, or am I missing something?
>
> - We do believe that there are datasets that can be naturally thought of as hypergraphs, especially social interaction networks such as IMDB.  Our comment in the discussion was meant to illustrate that more datasets that contain multi-way interactions would be extremely valuable for the topological deep learning community. We will revise the corresponding paragraph to make this clearer.
>
> > For traditional graphs: In the experimental evaluation, the encodings outperform the models without encoding. However, they are not compared to state-of-the-art GNNs, so this seems more like an ablation study. Can you comment on that?
>
> - Our goal here was to illustrate the benefits that standard graph neural networks that are most likely to be used in practice can obtain from capturing higher-order information, not to beat benchmarks, similar to [1] and [2]. We would also like to slightly push back against the notion that GCN or GIN are not *state-of-the-art*, as these architectures have recently been shown to still be competitive with more recent GNNs, including sophisticated graph transformers [3].
>
> > Why are the last three entries in Equation 8 (and in LDP) repeated?
>
> - This was indeed a typo, which has now been fixed. Equation 9 also suffered from the same typo and has also been fixed. We consider the min, max, mean, median, and std values of the CMS (Eq 8) or of the multi-set of node degrees.
>
> [1] Fesser, L. and Weber, M., Effective Structural Encodings via Local Curvature Profiles. In The Twelfth International Conference on Learning Representations.
>
> [2] Nguyen, K., Hieu, N.M., Nguyen, V.D., Ho, N., Osher, S. and Nguyen, T.M., 2023, July. Revisiting over-smoothing and over-squashing using ollivier-ricci curvature. In International Conference on Machine Learning (pp. 25956-25979). PMLR.
>
> [3] Luo, Y., Shi, L. and Wu, X.M., Can Classic GNNs Be Strong Baselines for Graph-level Tasks? Simple Architectures Meet Excellence. In Forty-second International Conference on Machine Learning.

---

> > ### Comment · Reviewer_Ej7N · 2025-08-04
> > **Reply to Rebuttal**
> >
> > Thank you for addressing my concerns and providing additional results.
> > I think the paper needs to be carefully revised, but that these issues can be addressed within the given time frame.
> > I have updated my score accordingly.

---

> > > ### Author Response · Authors · 2025-08-04
> > >
> > > Thank you for your positive feedback and for raising your score. We will revise the current manuscript as discussed.
> > >
> > > If there are any other concerns that you might have, please let us know. We would be happy to provide further clarification or run additional experiments during the remaining discussion period.

---

### Official Review · Reviewer_UZHC · 2025-07-01

**Clarity:** 3
**Significance:** 3
**Originality:** 2
**Rating:** 3
**Confidence:** 3

**Summary:**

This paper introduces hypergraph-level encodings to enhance higher-order relational learning. The authors propose structural and positional encodings (e.g., hypergraph Laplacian eigenvectors, random walk transition probabilities, curvature profiles) that capture higher-order information beyond pairwise interactions. Theoretical analysis proves that the encodings increase the expressivity of message-passing GNNs beyond the 1-WL test. Experiments show performance gains when combining hypergraph encodings with graph-level architectures, though hypergraph-specific architectures benefit minimally.

**Questions:**

- It would be better to provide a formal point of view of the term higher-order information and hypergraph-level encoding.
- Why do hypergraph architectures (e.g., UniGCN) not benefit from hypergraph encodings?
- The incidence matrix given in Definition 3.1 is also used in [1]. Also, the proposed covariance matrix $B_1B_1^T$ seems to be a simple form of $HWD^{-1}H^T$ in [1]. Please discuss their differences.

[1] Feng, Y., You, H., Zhang, Z., Ji, R., & Gao, Y. (2019). Hypergraph neural networks. In Proceedings of the AAAI conference on artificial intelligence, volume 33 (pp. 3558-3565).

**Ethical Concerns:**

["NO or VERY MINOR ethics concerns only"]

**Final Justification:**

After discussing with the authors, the concern remains: the paper has technical issues in its proofs in Theorem 3.2 and 3.5. Although the authors propose possible solutions in rebuttal, trying to fix the proofs, the conclusion that their proofs rely on is still controversial.

**Limitations:**

Yes

**Quality:**

3

**Strengths And Weaknesses:**

##### Strengths

- High-order information is crucial for complex tasks and is a promising research topic. The authors propose a novel shift from architectural extensions (e.g., message-passing hypergraph NNs) to data-centric hypergraph encodings.
- Well-structured with clear statements. Appendices provide the necessary background.
- Theoretical proofs (Theorems 3.2, 3.5, 3.11) demonstrate enhanced expressivity. Extensive experiments validate claims, with ablation studies and scalability analysis.

##### Weaknesses

- It is still unclear why the covariance matrix of the incidence matrix, as shown in Definition 3.1, helps to capture higher-order information. Theorem 3.2 shows that this way has higher discrimination power for nonisomorphic graphs. Does this mean all models with higher discrimination power potentially capture higher-order information?
- Encodings improve graph-level models but not hypergraph architectures. This discrepancy is noted but not analyzed.
- All proofs of expressive power are conducted within BREC datasets. Given so, the theoretical contributions can be limited.

---

> ### Author Rebuttal · Authors · 2025-07-31
>
> ## General response
>
> Thank you to all the reviewers for thoughtful and insightful feedback! We are pleased that reviewers are excited about the novel contributions of our work. Reviewers remark that **"higher-order information is crucial for complex tasks and is a promising research topic"** [UZHC] and that **"using encodings for improving MPNN-performance is an interesting research direction"** [Ej7N]. Reviewers also note that "the paper is well-written and easy to follow" [6gkV] and appreciate the **theoretical motivations** for our work [UZHC, 6gkV], as well as the extensive experiments [UZHC, GZBz].
>
> We now highlight a few important points raised by reviewers that we address in our rebuttal:
>
> - **Additional experiments** with MLPs and GTs [6gkV]: We provide additional results using MLPs and vanilla graph transfomers (GTs) without message-passing. These results support our claims that hypergraph-level encodings are generally useful and often - but not always - outperform their graph-level counterparts (LCP-FRC, RWPE).
> - More extensive hyperparameter tuning and **state-of-the-art results** [GZBz, Ej7N, 6gkV]: Several reviewers wondered if our encodings would still be useful with models that achieve SOTA performance. We first note that the models we consider, e.g. GCN and GIN, have recently been shown to outperform many newer models, including GTs, on several benchmarks [*], [+]. Using either the hyperparameters provided in these papers or by running HP searches when unavailable, we significantly increase the performance of our baseline models. With now close to SOTA results, we still find the same trends: our encodings are generally useful and often outperform their graph-level counterparts.
> - **Extended discussion** of why HGNNs do not seem to benefit from hypergraph-level encodings [GZBz, UZHC]: we have included additional remarks about oversmoothing and normalization in HGNNs and about encoding information that HGNNs might compute implicitly, as well as references to recent works that we think might allow one to make this discussion rigorous.
> - We have added **pre-processing runtimes** (for lifting graph to hypergraphs, and for computing encodings).
>
> We believe that these additions have made the paper significantly better and more accessible and thank the reviewers again for their input.
>
> [*] Luo, Y., Shi, L. and Wu, X.M., Can Classic GNNs Be Strong Baselines for Graph-level Tasks? Simple Architectures Meet Excellence. In Forty-second International Conference on Machine Learning.
>
> [+] Luo, Y., Shi, L. and Wu, X.M., 2024. Classic gnns are strong baselines: Reassessing gnns for node classification. Advances in Neural Information Processing Systems, 37, pp.97650-97669.
>
> **We now address each of the reviewer's comments and questions individually:**
>
> > It is still unclear why the covariance matrix of the incidence matrix, as shown in Definition 3.1, helps to capture higher-order information. Theorem 3.2 shows that this way has higher discrimination power for nonisomorphic graphs. Does this mean all models with higher discrimination power potentially capture higher-order information?
>
> - Thank you for raising this point. The covariance matrix $B_1B_1^T$ is useful because it captures how often two vertices appear together in a hyperedge, and its eigenvectors—the Hodge-Laplacian encodings—retain that higher-order signal even after we project back to a simple graph. Theorem 3.2 shows that feeding these encodings to a message-passing GNN makes it strictly more expressive than 1-WL.
>
> - Capturing higher-order information allows for higher expressivity. Here, higher-order information refers to interaction patterns that involve three or more vertices simultaneously. Formally, any hyperedge $e \subseteq V$ with $|e| > 2$, encodes such information. Such interactions (often called "higher-order interactions" in network science [1]) cannot be reconstructed from the pairwise edge set alone, so they encode genuinely new structural signals beyond a graph's 1-skeleton. Many of the graph characteristics that form the basis of graph-level encodings capture higher-order information as well, e.g., Ollivier's Ricci curvature (LCP-ORC) captures local cycle counts. However, if these characteristics are computed on a hypergraph parametrization, richer higher-order information is captured. This is confirmed by both our theoretical and computational analysis: Our expressivity results (Thm. 3.2, 3.5, 3.11, 3.13) show that pairs of graphs exist that can be distinguished by computing the respective characteristics at the hypergraph-level, but not by its graph-level analogue. *This progression of richer encoded higher-order information induces a hierarchy of more expressive models.* We illustrate this on the example of local curvature profiles (LCP): Simple Forman curvature computed on a graph's 1-skeleton cannot capture higher-order information. Using a curvature notion that encodes higher-order information (augmented Forman curvature or Ollivier's curvature, both of which capture cycle counts, see Apx. A.5), is provably more expressive [2]. Computing Ricci curvature on the hypergraph-level is provably more expressive than its graph-level counterpart (Thm. 3.11). Our empirical comparison of hypergraph- and graph-level encodings also confirms this intuition.
>
> > Encodings improve graph-level models but not hypergraph architectures. This discrepancy is noted but not analyzed.
>
> - Thank you for pointing this out - we agree that this is indeed a very interesting point. We suspect that since hypergraph layers already process the full incidence structure, the co-incidence statistics that, for example, our Hodge encodings add are largely redundant: most of that signal is already available to the architecture. By contrast, a standard graph GNN sees only pairwise edges; the encodings inject complementary higher-order cues it would otherwise miss, hence the larger gains. We believe that this intuition could be made rigorous using techniques similar to [3], which operates at the graph level, but consider this beyond the scope of the current paper.
>
> > All proofs of expressive power are conducted within BREC datasets. Given so, the theoretical contributions can be limited.
>
> - We would like to point out that Theorems 3.2, 3.5, 3.11, 3.13 and A.1 are stated for all finite graphs/hypergraphs; the proofs simply exhibit witness pairs that are 1-WL-hard, and we picked those pairs from BREC because the dataset conveniently aggregates many classic counter-examples in one place, making the exposition shorter and letting us reuse them for the empirical section. Crucially, several proofs already rely on graphs that pre-date—and in fact are not unique to—BREC: e.g. the Rook vs. Shrikhande pair.
>
> > It would be better to provide a formal point of view of the term higher-order information and hypergraph-level encoding.
>
> - Please see response above.
>
> > The incidence matrix given in Definition 3.1 is also used in [1]. Also, the proposed covariance matrix $B_1B_1^T$  seems to be a simple form of $HWD^{-1}H^T$  in [1]. Please discuss their differences.
>
> - Thank you for bringing this up. Please note that we do not propose the 0- or 1-Hodge Laplacian $B_1^TB_1$ and $B_1B_1^T$. These are well-established quantities in the theory of hypergraphs.
> - Our covariance $B_1 B_1^{\top}$ simply multiplies the (unsigned) incidence matrix by its transpose, so the diagonal records vertex degrees and the off-diagonal entry $(i,j)$ counts how many hyperedges jointly contain $v_i$ and $v_j$; no weighting or size normalisation is applied. This raw co-incidence statistic is what we later diagonalise to obtain positional encodings. In contrast, HGNN first weights each hyperedge by $W$ and then divides by its size through $D_e^{-1}$, giving the normalised operator $H W D_e^{-1} H^{\top}$ (and its symmetric variant $D_v^{-1/2} H W D_e^{-1} H^{\top} D_v^{-1/2}$) that appears inside every convolution layer. This choice turns each hyperedge into a clique whose contribution is averaged across its members, making the matrix row- and column-balanced. This is useful for stable message passing but less informative about absolute co-membership frequencies. Accordingly, $B_1 B_1^{\top}$ reduces to the (combinatorial) graph Laplacian when $|e|=2$, whereas $H W D_e^{-1} H^{\top}$ collapses to a degree-normalised adjacency; the two operators are thus complementary rather than equivalent. We hope this clarifies and would be happy to elaborate further.
>
> [1] Aktas, M.E., Nguyen, T., Jawaid, S., Riza, R. and Akbas, E., 2021. Identifying critical higher-order interactions in complex networks. Scientific reports, 11(1), p.21288.
>
> [2] Fesser, L. and Weber, M., Effective Structural Encodings via Local Curvature Profiles. In The Twelfth International Conference on Learning Representations.
>
> [3] Kanatsoulis, C., Choi, E., Jegelka, S., Leskovec, J. and Ribeiro, A., Learning Efficient Positional Encodings with Graph Neural Networks. In The Thirteenth International Conference on Learning Representations.

---

> > ### Author Response · Authors · 2025-08-04
> >
> > Dear reviewer, thank you again for your positive feedback on our paper and for considering our rebuttal. Kindly let us know whether your concerns have been addressed or if there are any remaining concerns that you might have.
> >
> > We would be happy to provide further clarification or run additional experiments during the remaining discussion period.

---

> > > ### Comment · Reviewer_UZHC · 2025-08-08
> > >
> > > Thanks for the author's response. Better avoid the term "High-order information" in the title if it lacks a formal definition, though the author explains it is somehow encoded with a hyperedge.
> > >
> > > $B_1B_1^T$ shares a similar design or corresponds to a simplified form of the existing models, such as $HWD^{-1}H^T$ [1]. The benefit of $B_1B_1^T$ over the existing is still inconving.
> > >
> > > In the proofs of Theorem 3.2 and 3.5, the author shows that there exist graph pairs (in BREC dataset) that can be distinguished by H-k-*, but cannot be distinguished by k-* (The second statement). But is the first statement, "H-k-* is strictly more expressive than the 1-WL test", proved? Also, both statements seem trivial. "H-k-* is strictly more expressive than k-*" can be informative, but is missing.

---

> ### Author Response · Authors · 2025-08-08
>
> > Better avoid the term "High-order information" in the title if it lacks a formal definition, though the author explains it is somehow encoded with a hyperedge.
>
> We will revise the text to make it clearer to the reader what we mean by "higher-order information".
>
> > shares a similar design or corresponds to a simplified form of the existing models, such as [*]. [...]
>
> We thank the reviewer for this point. We consider both the Hodge Laplacian, as well as the normalized Laplacian from [\*] in all our experiments, e.g. "Norm H-20-LAPE" in Table 1. See Appendix A.4.1 for more details. We consider unweighted hyper-edges, so $W=I$. The combinatorial Hodge-Laplacian, originally introduced in [1], extends the graph Laplacian as a natural shift operator for hypergraphs [2,3]. The $k$-th combinatorial Hodge-Laplacian on simplicial complexes is
> $$L_k = B_k^\top B_k + B_{k+1} B_{k+1}^\top $$
> $B_1=H$ in [\*]. The quantity we use is $L_0=B_1 B_1^\top$ and $L_1=B_1^TB_1$ as for hypergraphs, we only have rank $0$ (nodes) and rank 1 (hyperedges) simplices, so $B_0=0$, $B_k=0, k>1$. The hypergraph Laplacian in [\*] originates from the fact that node classification on hypergraphs can be viewed as a regularization framework:
> $$
> \arg \min  R_{emp}(f) + \Omega(f)  $$
> where $\Omega(f)$ is a regularizer defined on the hypergraph, $R_{\mathrm{emp}}(f)$ denotes the supervised empirical loss, and $f(\cdot)$ is the classification function. The regularizer $\Omega(f)$ is given by
> $$
> \Omega(f) = \frac{1}{2} \sum_{e \in E} \sum_{\{u,v\} \subseteq V}
> \frac{\ h(u,e) \, h(v,e)}{\delta(e)}
> \left( \frac{f(u)}{\sqrt{d(u)}} - \frac{f(v)}{\sqrt{d(v)}} \right)^2,
> $$
> where $h(u,e)$ indicates the incidence between vertex $u$ and hyperedge $e$, $\delta(e)$ is the degree of hyperedge $e$, and $d(u)$ is the degree of vertex $u$. Then the normalized regularizer can be written as
> $
> \Omega(f) = f^\top \Delta f.
> $
> where:
> $$ \Delta = I - D_v^{-1/2} B_1 D_e^{-1} B_1^\top D_v^{-1/2}.
> $$
> The convolution on hypergraphs in [\*] can be formulated as:
> $$
> Y = D_v^{-1/2} B_1 W D_e^{-1} B_1^\top D_v^{-1/2} X \Theta
> $$
> Our goal is not to construct a new layer as in [\*], but to augment the graph's features, as a generalization of graph-level encodings (e.g., [4, 5]). In the main text, we present results for the Hodge-Laplacian because on graphs, it reduces to the graph Laplacian - this makes it a natural choice for a comparison of graph- and hypergraph-level LAPE encodings. Empirically, the Hodge Laplacian-based encodings appear to slightly outperform the Normalized Laplacian ones. It is not obvious if this trend generally holds due to some inherent properties of the Laplacians, so we will include this as an area for future research.
>
> > In the proofs of Theorem 3.2 and 3.5 [...] there exist graph pairs that can be distinguished by H-k-, but cannot be distinguished by k- [...]. But is the first statement, "H-k-* is strictly more expressive than the 1-WL test", proved? Also, both statements seem trivial. "H-k-* is strictly more expressive than k-*" can be informative, but is missing.
>
> In the proofs of theorem 3.2 and 3.5, we use pair 0 of the Basic category of BREC, which is 1-WL indistinguishable. Thus, the first statement follows from the fact that the general class of MPGNNs (GIN specifically) is as expressive as 1-WL [6]. To establish strictly better expressivity of MPGNNs (GIN) with hypergraph encodings, it is sufficient to identify a set of non-isomorphic graphs that cannot be distinguished by the 1-WL test, but that differ in their encodings. This is because MPGNNs are at most as powerful as 1-WL, but enrichment with more expressive hypergraph-level information allows for distinguishing these graphs. Note that some graph-level encodings are also more expressive than 1-WL (see, e.g., [7]).
>
> We thank the reviewer for pointing out that our explanations could have been clearer. We have revised the text accordingly.
>
> [*] Feng, Y., You, H., Zhang, Z., Ji, R., & Gao, Y. (2019, July). Hypergraph neural networks. In Proceedings of the AAAI conference on artificial intelligence.
>
> [1] Eckmann, B. (1944). Harmonische funktionen und randwertaufgaben in einem komplex. Commentarii Mathematici Helvetici, 17(1).
>
> [2] Schaub, M. T., Zhu, Y., Seby, J. B., Roddenberry, T. M., & Segarra, S. (2021). Signal processing on higher-order networks: Livin’on the edge... and beyond. Signal Processing, 187.
>
> [3] Lim, L. H. (2020). Hodge Laplacians on graphs. Siam Review, 62(3).
>
> [4] Dwivedi, Luu, Laurent, Bengio, & Bresson, (2021). Graph neural networks with learnable structural and positional representations. arXiv:2110.07875.
>
> [5] Dwivedi & Bresson. A generalization of transformer networks to graphs. In AAAI Workshop on Deep Learning on Graphs: Methods and Applications, 2021
>
> [6] Xu, Hu, Leskovec, & Jegelka. (2018). How powerful are graph neural networks?.  arXiv:1810.00826.
>
> [7] Fesser & Weber, (2023). Effective structural encodings via local curvature profiles. arXiv:2311.14864.

---

> > ### Comment · Reviewer_UZHC · 2025-08-09
> >
> > Thanks for your feedback. In Theorem 3.2 and 3.5, it seems that the author proves "H-k-* is strictly more expressive than the 1-WL test" by only testing the discrimination ability on a specific graph pair in BREC? If so, these are not proper proofs. The proof should be considered for any graph pair, ...

---

> > > ### Author Response · Authors · 2025-08-09
> > >
> > > We respectfully disagree with the suggestion that our proofs of Theorems 3.2 and 3.5 are not “proper.” Our strategy exactly mirrors the one used in [1] for Local Curvature Profiles (LCP):
> > >
> > > - **Step 1 – Non-degradation.** Adding a deterministic encoding to an MPGNN cannot reduce its expressivity; this follows by a direct adaptation of Xu et al. [2], Thm.3 - the same lemma invoked in [1].
> > > - **Step 2 – Strict separation.** It then suffices to exhibit **one** pair of non-isomorphic graphs that the 1-WL test (and hence a plain MPGNN) cannot distinguish but that differ in the new encoding.
> > >
> > > In our paper, Pair 0 (“Basic” category) in BREC plays that role for both H-k-LAPE and H-k-RWPE. In [1], the witness is the classical **Rook vs. Shrikhande** graphs.
> > >
> > > We reproduce Theorem 1 and its proof from [1] below (quoted verbatim).
> > >
> > > **Theorem 1 and proof from "Effective Structural Encodings via Local Curvature Profiles"**
> > >
> > > > Theorem 3.1: MPGNNs with LCP structural encodings are strictly more expressive than the 1-WL test and hence than MPGNNs without encodings.
> > >
> > > > Proof: Seminal work by (Xu et al., 2018; Morris et al., 2019) has established that standard MPGNNs, such as GIN, are as expressive as the 1-WL test. It can be shown via a simple adaption of (Xu et al., 2018, Thm. 3) that adding LCP encodings does not decrease their expressivity, i.e., MPGNNs with LCP are at least as expressive as the 1-WL test. **To establish strictly better expressivity, it is sufficient to identify a set of non-isomorphic graphs that cannot be distinguished by the 1-WL test, but that differ in their local curvature profiles.** Consider the **4x4 Rook and Shrikhande graphs**, two non-isomorphic, strongly regular graphs. While nodes in both graphs have identical node degrees (e.g. could not be distinguished with classical MPGNNs), their 2-hop connectivities differ. This results in differences in the optimal transport plan of the neighborhood measures that is used to compute ORC (see Fig.2 [in the original paper]). As a result, the (local) ORC distribution across the two graphs varies, resulting in different local curvature profiles.
> > >
> > > Thus, our arguments satisfy the formal definition of “strictly more expressive,” exactly as in [1] (Step 1 has been made more explicit in the revised manuscript, as mentioned in our earlier responses).
> > >
> > > We hope this clarifies the matter and are happy to address any further questions.
> > >
> > > [1] Fesser, L. and Weber, M., Effective Structural Encodings via Local Curvature Profiles. In The Twelfth International Conference on Learning Representations.
> > >
> > > [2] Xu, K., Hu, W., Leskovec, J. and Jegelka, S., 2018. How powerful are graph neural networks?. arXiv preprint arXiv:1810.00826.

---

> > ### Comment · Reviewer_UZHC · 2025-08-09
> >
> > Thanks for the rebuttal.
> >
> > The correct proof of the 1st statement in Thm. 3.2 and 3.5 should be:
> > 1. Prove H-k-* $\succeq$ 1-WL;
> > 2. Show examples that can be distinguished by H-k-* but fail by 1-WL;
> >
> > The proofs in the paper miss Step 1.
> >
> > Meanwhile, from the rebuttal above, the author tries to prove Step 1 by showing H-k-* $\succeq$ GIN $=$ 1-WL, where they rely on the conclusion GIN $=$ 1-WL. However, GIN $=$ 1-WL is not the case in the practical continuous feature space. In fact, still GIN $\preceq$ 1-WL  [1] [2].
> >
> > [1] Principal Neighbourhood Aggregation for Graph Nets
> >
> > [2] Breaking the expression bottleneck of graph neural networks

---

> ### Author Response · Authors · 2025-08-09
>
> Thank you for the quick response.
>
> The reviewer is correct in highlighting the particular difficulties that come with a continuous feature space when providing a formal proof of Step 1. We agree that in this case, Step 1 must be stated independently of any claim about GIN. This might take the following general form:
>
> > Following the constructions in *Principal Neighborhood Aggregation* and *Breaking the Expressive Bottlenecks of GNNs*, we deterministically map every real-valued feature vector to a unique discrete token via an order-preserving injection $Z:\mathbb{R}^{d}\to\mathbb{N}$. The multiset $\{Z(x_{u})\}_{u\in\mathcal{N}(v)}$ is then processed by an injective SUM + MLP (or the degree-scaled SUM used in PNA), after which an injective MLP updates the node state. Because both $Z$ and the aggregator are injective on multisets, each round reproduces the exact 1-WL coloring, establishing $\text{H-k-\*} \succeq \text{1-WL}$ even in continuous feature spaces, where H-k-\* denotes an MPGNN equipped with our hypergraph encoding.
>
> We will incorporate this kind of argument in the final manuscript, thereby making the first part of the proof required for Theorems 3.2 and 3.5 explicit. We will also be sure to cite both *Principal Neighborhood Aggregation* and *Breaking the Expressive Bottlenecks of GNNs* and thank the reviewer for bringing up these papers.
>
> We hope this clarifies Step 1 and are happy to address any further questions.

---

### Author Response · Authors · 2025-08-08
**Thank You**

As the discussion period draws to a close, we thank all reviewers for their constructive feedback and active engagement. We appreciate that the reviewers who initially had reservations now feel that their concerns have been addressed. We will incorporate your suggestions into the final manuscript.

---

### Note · Authors · 2025-08-12

We are grateful to all reviewers for their thoughtful engagement, which has substantially improved the manuscript. We write only to clarify a point raised by Reviewer UZHC near the close of the discussion.

UZHC questioned the proofs of Theorems 3.2 and 3.5, in particular the implicit assumption that a Message-Passing GNN (MPGNN) can match the expressivity of the 1-Weisfeiler–Leman (1-WL) test. This equivalence is a widely accepted result in the expressiveness-and-encodings literature: “simple MPNNs that are already expressive enough to be equivalent to 1-WL in distinguishing non-isomorphic graphs” (e.g., Franks et al. 2024; Morris et al. 2022; Zhao et al. 2024). Our proofs merely invoke that consensus, and we have now made this explicit in the revised manuscript.

While most prior work states the 1-WL equivalence in discrete feature spaces, the argument extends naturally to the continuous features brought up by reviewer UZHC because injective aggregation, not feature discreteness, drives 1-WL’s distinguishing power. We will add an explicit remark to the paper to make this possible extension transparent. Building on that baseline, our theoretical results show that augmenting an MPGNN with our hypergraph encodings strictly increases its expressive power beyond 1-WL. For this, it suffices to find a single pair of non-isomorphic graphs, which the 1-WL test cannot distinguish, but which differ in their encodings.

We appreciate UZHC’s diligence and trust this clarification resolves the concern without casting doubt on our theoretical contributions. Other reviewers have independently remarked that our “theoretical analysis is well-supported”, that earlier questions were “fully addressed”, and that the revised results are “highly interesting”.

Thank you again for your time and constructive feedback. We look forward to incorporating these final clarifications in the final manuscript.

Reference
- Franks B. J. et al., Weisfeiler-Leman at the Margin, ICML 2024.
- Morris C. et al., SpeqNets, ICML 2022.
- Zhao L. et al., From Stars to Subgraphs, ICLR 2024.

---

### Decision · Program_Chairs · 2025-09-17

**Decision:**

Accept (poster)

**Comment:**

This paper introduces a data-centric approach to capture higher-order relationships in graphs by augmenting node features with novel hypergraph-level encodings derived from Laplacians, curvature, and random walks.

The work's primary strengths are its novel and provably expressive hypergraph encodings and an extensive experimental evaluation showing these encodings significantly boost the performance of standard GNNs. Some concerns are the limited performance gain when applying these encodings to native hypergraph neural network architectures and a theoretical proof framework whose assumptions were debated by one reviewer. The discussion period was effective, with the authors running numerous new experiments that resolved the most critical initial weaknesses. Concerns about weak experimental baselines and lack of hyperparameter tuning were fully addressed with new results and additional MLP/GT baselines. The authors also provided detailed runtime analysis and addressed multiple presentation issues. This effort led three of the four reviewers to raise their scores. The primary debatable point is a technical disagreement with one reviewer about the assumptions underlying the expressivity proofs in continuous feature spaces. Nevertheless, the consensus shifted strongly in favor of acceptance due to the strengthened empirical evidence. Therefore, I recommended acceptance.